# Distributionally Robust Optimization via Ball Oracle Acceleration

**Yair Carmon**
Tel Aviv University
ycarmon@tauex.tau.ac.il

**Danielle Hausler**
Tel Aviv University
hausler@mail.tau.ac.il

## Abstract

We develop and analyze algorithms for distributionally robust optimization (DRO) of convex losses. In particular, we consider group-structured and bounded $f$-divergence uncertainty sets. Our approach relies on an accelerated method that queries a ball optimization oracle, i.e., a subroutine that minimizes the objective within a small ball around the query point. Our main contribution is efficient implementations of this oracle for DRO objectives. For DRO with $N$ non-smooth loss functions, the resulting algorithms find an $\epsilon$-accurate solution with $\widetilde{O}\left(N\epsilon^{-2/3} + \epsilon^{-2}\right)$ first-order oracle queries to individual loss functions. Compared to existing algorithms for this problem, we improve complexity by a factor of up to $\epsilon^{-4/3}$.

## 1 Introduction

The increasing use of machine learning models in high-stakes applications highlights the importance of reliable performance across changing domains and populations [11, 46, 35]. An emerging body of research addresses this challenge by replacing Empirical Risk Minimization (ERM) with Distributionally Robust Optimization (DRO) [6, 50, 49, 22, 40], with applications in natural language processing [47, 61, 35], reinforcement learning [17, 54] and algorithmic fairness [27, 56]. While ERM minimizes the average training loss, DRO minimizes the worst-case expected loss over all probability distributions in an *uncertainty set $\mathcal{U}$*, that is, it minimizes

$$\mathcal{L}_{\text{DRO}}(x) := \sup_{Q \in \mathcal{U}} \mathbb{E}_{S \sim Q}[\ell_S(x)], \tag{1}$$

where $\ell_S(x)$ is the loss a model $x \in \mathcal{X}$ incurs on a sample $S$ and $\mathcal{X}$ is a closed convex set with bounded Euclidean diameter. This work develops new algorithms for DRO, focusing on formulations where $\mathcal{U}$ contains distributions supported on $N$ training points, where $N$ is potentially large. We consider two well-studied DRO variants: (1) Group DRO [57, 31, 49], and (2) $f$-divergence DRO [19, 6, 22].

**Group DRO** Machine learning models may rely on spurious correlations (that hold for most training examples but are wrongly linked to the target) and therefore suffer high loss on minority groups where these correlations do not hold [30, 27, 11]. To obtain high performance across all groups, Group DRO minimizes the worst-case loss over groups. Given a set $\mathcal{U} = \{w_1, \dots, w_M\}$ of $M$ distributions over $[N]$, the Group DRO objective is[1]

$$\mathcal{L}_{\text{g-DRO}}(x) := \max_{i \in [M]} \mathbb{E}_{j \sim w_i} \ell_j(x) = \max_{i \in [M]} \sum_{j=1}^{N} w_{ij} \ell_j(x). \tag{2}$$

---

[1]Typically, each $w_i$ is uniform over a subset ("group") of the $N$ training points. However, most approaches (and ours included) extend to the setting of arbitrary $w_i$'s, which was previously considered in [57].

36th Conference on Neural Information Processing Systems (NeurIPS 2022).

| Smoothness | Method | Group DRO (2) | $f$-divergence DRO (3) |
|---|---|---|---|
| None ($L = \infty$) | Subgradient method [45] | $N\epsilon^{-2}$ | $N\epsilon^{-2}$ |
| | Stoch. primal-dual [43] * | $M\epsilon^{-2}$ | $N\epsilon^{-2}$ |
| | MLMC stoch. gradient [39] | - | $\rho\epsilon^{-3}$ or $\alpha^{-1}\epsilon^{-2}$ † |
| | Ours | $N\epsilon^{-2/3} + \epsilon^{-2}$ | $N\epsilon^{-2/3} + \epsilon^{-2}$ |
| Weak ($L \approx 1/\epsilon$) | AGD on softmax [44] | $N\epsilon^{-1}$ | $N\epsilon^{-1}$ |
| | Ours | $N\epsilon^{-2/3} + N^{3/4}\epsilon^{-1}$ | $N\epsilon^{-2/3} + \sqrt{N}\epsilon^{-1}$ |

**Table 1.** Number of $\nabla\ell_i$ and $\ell_i$ evaluations to obtain $\mathbb{E}[\mathcal{L}_{\text{DRO}}(x)] - \min_{x_\star \in \mathcal{X}} \mathcal{L}_{\text{DRO}}(x_\star) \leq \epsilon$, where $N$ is the number of training points and (for Group DRO) $M$ is the number of groups. The stated rates omit constant and polylogarithmic factors. * Requires an additional uniform bound on losses (see Appendix A.1). † These rates hold only for specific $f$-divergences: CVaR at level $\alpha$ or $\chi^2$-divergence with size $\rho$, respectively.

If we define the loss of group $i$ as $\mathcal{L}_i(x) := \sum_{j=1}^{N} w_{ij}\ell_j(x)$ then objective (2) is equivalent to $\max_{q \in \Delta^M} \sum_{i \in [M]} q_i \mathcal{L}_i(x)$ with $\Delta^M := \{q \in \mathbb{R}_{\geq 0}^M \mid \vec{1}^T q = 1\}$. Note that, unlike ERM, Group DRO requires additional supervision in the form of subgroup identities encoded by $\{w_i\}$.

**DRO with $f$-divergence** Another approach to DRO, which requires only as much supervision as ERM, takes $\mathcal{U}$ to be an $f$-divergence ball around the empirical (training) distribution. For every convex function $f : \mathbb{R}_+ \to \mathbb{R} \cup \{+\infty\}$ such that $f(1) = 0$, $f(0/0) = 0$ and the $f$-divergence between distributions $q$ and $p$ over $[N]$ is $D_f(q, p) := \sum_{i \in [N]} p_i f(q_i/p_i)$. The $f$-divergence DRO problem corresponds to the uncertainty set $\mathcal{U} = \{q \in \Delta^N : D_f(q, \frac{1}{N}\mathbf{1})) \leq 1\}$, i.e.,

$$\mathcal{L}_{f\text{-div}}(x) := \max_{q \in \Delta^N : \frac{1}{N}\sum_{i \in [N]} f(Nq_i) \leq 1} \sum_{i \in [N]} q_i \ell_i(x). \tag{3}$$

Several well-studied instances of DRO are a special case of this formulation, with the two most notable examples being conditional value at risk (CVaR) and $\chi^2$ uncertainty sets. CVaR at level $\alpha$ corresponds to $f(x) = \mathbb{1}_{\{x < \frac{1}{\alpha}\}}$ such that $\mathcal{U} = \{q \in \Delta^N \text{ s.t } \|q\|_\infty \leq 1/(\alpha N)\}$, and has many applications in finance such as portfolio optimization and credit risk evaluation [48, 37] as well as in machine learning [47, 39, 20, 60, 17, 54]. The $\chi^2$ uncertainty set with size $\rho > 0$ corresponds to $f(x) := \frac{1}{2\rho}(x - 1)^2$ and the resulting DRO problem is closely linked to variance regularization [21] and has been extensively studied in statistics and machine learning [42, 27, 21, 39, 61].

**Complexity notion** In this paper, we design improved-complexity methods for solving the convex problems (2) and (3) under the assumption that the loss $\ell_i$ is convex and Lipschitz for all $i$. We measure complexity by the (expected) required number of $\ell_i(x)$ and $\nabla\ell_i(x)$ evaluations to obtain $\epsilon$-suboptimal solution, i.e., return $x$ such that $\mathcal{L}_{\text{DRO}}(x) - \min_{x_\star \in \mathcal{X}} \mathcal{L}_{\text{DRO}}(x_\star) \leq \epsilon$ with constant probability. Table 1 summarizes our complexity bounds and compares them to prior art. Throughout the introduction we assume (for simplicity) a unit domain size ($\|x - y\| \leq 1$ for all $x, y \in \mathcal{X}$) and that each loss is 1-Lipschitz.

**Prior art** Let us review existing methods that solve Group DRO and $f$-divergence DRO problems (see Section 1.1 for extended discussion). For a dataset with $N$ training points, the subgradient method [45] finds an $\epsilon$ approximate solution in $\epsilon^{-2}$ iterations. Computing a single subgradient costs $N$ functions evaluations (since we need to find the maximizing $q$). Therefore, the complexity of this method is $O(N\epsilon^{-2})$.

DRO can also be viewed as a game between a minimizing $x$-player and a maximizing $q$-player, which makes it amenable to primal-dual methods [43, 42, 49]. If we further assume that the losses are bounded then, for $q \in \Delta^m$, stochastic mirror descent with local norms obtains a regret bound of $O(\sqrt{m\log(m)/T})$ (see Appendix A.1). As a consequence, for Group DRO (where $m = M$) the complexity is $\widetilde{O}(M\epsilon^{-2})$, and for $f$-divergence DRO ($m = N$) the complexity is $\widetilde{O}(N\epsilon^{-2})$.

Levy et al. [39] studied $\chi^2$-divergence and CVaR DRO problems, and proposed using standard gradient methods with a gradient estimator based on multilevel Monte Carlo (MLMC) [9]. For $\chi^2$-divergence with ball of size $\rho$ they proved a complexity bound of $\widetilde{O}(\rho\epsilon^{-3})$, and for CVaR at

level $\alpha$ they established complexity $\widetilde{O}(\alpha^{-1}\epsilon^{-2})$. However, for large uncertainty sets (when $\rho$ or $\alpha^{-1}$ approach $N$) their method does not improve over the subgradient method.

Stronger complexity bounds are available under the weak smoothness assumption that each $\ell_i$ has $O(\epsilon^{-1})$-Lipschitz gradient. Note that this is a weak assumption since if a function $\ell$ is not continuously differentiable, it is possible to approximate $\ell$ with additive error at most $\epsilon/2$, by its Moreau envelope $\widetilde{\ell}(x) = \min_{y \in \mathcal{X}} \left\{ \ell(y) + \frac{G^2}{2\epsilon}\|x - y\|^2 \right\}$ (see [15, Appendix A.1] for more details). In particular, we can apply Nesterov's accelerated gradient descent method [44] on an entropy-regularized version of our objective to solve the problem with complexity $\widetilde{O}(N\epsilon^{-1})$; see Appendix A.2 for more details.

**Our contribution** We propose algorithms that solve the problems (2) and (3) with complexity $\widetilde{O}(N\epsilon^{-2/3} + \epsilon^{-2})$. Compared to previous works, we obtain better convergence rates for DRO with general $f$-divergence when $N \gg 1$ and for Group DRO when $M \gg N\epsilon^{4/3}$. When the losses have $O(\epsilon^{-1})$-Lipschitz gradient, we solve $f$-divergence DRO with complexity $\widetilde{O}(N\epsilon^{-2/3} + \sqrt{N}\epsilon^{-1})$, and, under an even weaker mean-square smoothness assumption ($\mathbb{E}_{j \sim w_i}\|\nabla \ell_j(x) - \nabla \ell_j(y)\|^2 \leq O(\epsilon^{-2})\|x - y\|^2$ for all $x, y$ and $i$), we solve Group DRO with complexity $\widetilde{O}(N\epsilon^{-2/3} + N^{3/4}\epsilon^{-1})$.

Our complexity bounds are independent of the structure of $f$ and $\{w_i\}$, allowing us to consider arbitrarily $f$-divergence balls and support a large number of (potentially overlapping) groups. Our rates are optimal up to logarithmic factors for the special case of minimizing $\max_{i \in [N]} \ell_i(x)$, which corresponds to Group DRO with $N$ distinct groups and $f$-divergence DRO with $f = 0$ [58, 62, 15].

**Our approach** Our algorithms are based on a technique for acceleration with a *ball optimization oracle*, introduced by Carmon et al. [13] and further developed in [15, 3]. Given a function $F$ and a query point $x$, the ball optimization oracle returns an approximate minimizer of $F$ inside a ball around $x$ with radius $r$; the works [13, 15, 3] show how to minimize $F$ using $\widetilde{O}(r^{-2/3})$ oracle calls. Our development consists of efficiently implementing ball oracles with radius $r = \widetilde{O}(\epsilon)$ for the DRO problems (2) and (3), leveraging the small ball constraint to apply stochastic gradient estimators that would have exponential variance and/or cost without it.

Carmon et al. [15] previously executed this strategy for minimizing the maximum loss, i.e., $\max_{q \in \Delta^N} \sum_i q_i \ell_i(x)$, which is a special case of both Group DRO and $f$-divergence DRO. However, the ball-oracle implementations of [15] do not directly apply to the DRO problems that we consider; our oracle implementations differ significantly and intimately rely on the Group DRO and $f$-divergence problem structure. We now briefly review the main differences between our approach and [15], highlighting our key technical innovations along the way.

Since the Group DRO objective is $\max_{q \in \Delta^M} \sum_i q_i \mathcal{L}_i(x)$ for $\mathcal{L}_i(x) = \sum_{j \in [N]} w_{ij} \ell_j(x)$, one may naively apply the technique of [15] with $\mathcal{L}_i$ replacing $\ell_i$. However, every step of such a method would involve computing quantities of the form $e^{\mathcal{L}_i(x)/\epsilon'}$ (for some $\epsilon' = \widetilde{\Theta}(\epsilon)$), which can be up to $N$ times more expensive than computing $e^{\ell_j(x)/\epsilon'}$ for a single $j$. To avoid such expensive computation we use MLMC [9] to obtain an unbiased estimate of $e^{\mathcal{L}_i(x)/\epsilon}$ with complexity $O(1)$ and appropriately bounded variance. In the weakly-smooth case we also adapt our estimator to facilitate variance reduction [33, 2].

For $f$-divergence we consider the well-known dual form [6, 50]

$$\max_{q \in \Delta^N} \sum_{i \in [N]} \{q_i \ell_i(x) - \psi(q_i)\} = \min_{y \in \mathbb{R}} \left\{ \Upsilon(x, y) \coloneqq \sum_{i \in [N]} \psi^*(\ell_i(x) - y) + y \right\}$$

where $\psi^*(v) = \max_{t \geq 0}\{vt - \psi(t)\}$ is the Fenchel dual of $\psi$. Stochastic gradient methods applied directly on the dual formulation are notoriously unstable (see e.g., [42]). This is due to the fact that the Fenchel dual $\psi^*$ can be very badly behaved even for standard $f$-divergences. We solve this problem by (a) considering a small ball, (b) entropy-regularizing $\psi$. Our techniques rely on two technical observations: (i) $\log(\psi_\epsilon^{*\prime}(\cdot))$ is $1/\epsilon'$-Lipschitz for all convex $\psi$, and (ii) for 1-Lipschitz losses, $y^*(x) = \operatorname{argmin}_{y \in \mathbb{R}} \Upsilon(x, y)$ satisfies $|y^*(x) - y^*(x')| \leq \|x - x'\|$ for all $x, x'$. To the best of our knowledge, these observations are new and potentially of independent interest.

**Paper organization.** Section 2 provides notation and a concise summary of the ball acceleration framework (largely taken from prior work) on which we build our algorithms. Sections 3 and 4

present our main contributions in the Group and $f$-divergence DRO settings, respectively. Finally, Section 5 concludes with discussion on the limitations and possible extensions of our work.

## 1.1 Additional related work

**MLMC estimators** The multilevel Monte Carlo (MLMC) technique was introduced by Giles [26] and Heinrich [29] in order to reduce the computational cost of Monte Carlo estimation of integrals. Blanchet and Glynn [9] extended this technique to estimating functions of expectation and proposed several applications, including stochastic optimization [10]. In this work we use their estimator for two distinct purposes: (1) obtaining unbiased Moreau envelope gradient estimates for ball oracle acceleration as proposed by Asi et al. [3], and (2) estimating the exponential of an expectation for Group DRO. Levy et al. [39] also rely on MLMC for DRO, but quite differently than we do: they directly estimate the DRO objective gradient via MLMC, while we estimate different quantities.

**Other DRO methods** Several additional works proposed algorithms with theoretical guarantees for $f$-divergence DRO. Jin et al. [32] considered non-convex and smooth losses. Song et al. [53] proposed an algorithm for linear models with complexity comparable to the "AGD on softmax" approach (Appendix A.2). Namkoong and Duchi [42] proposed a primal-dual algorithm that is suitable for small uncertainty $\chi^2$ sets (with size $\rho \ll \frac{1}{N}$) and Curi et al. [20] proposed a primal-dual algorithm specialized for CVaR. Other works consider DRO with uncertainty sets defined using the Wasserstein distance [24, 23, 51, 34]. Another relevant line of works proposes refinements for DRO that address some of the challenges in applying it to learning problems [60, 59, 55].

## 2 Preliminaries

**Notation** We write $\|\cdot\|$ for the Euclidean norm. We denote by $\mathbb{B}_r(x_0)$ the Euclidean ball of radius $r$ around $x_0$. We let $\Delta^n := \{q \in \mathbb{R}^n_{\geq 0} \mid \mathbf{1}^T q = 1\}$ denote the probability simplex in $\mathbb{R}^n$. For the sequence $z_m, \ldots, z_n$ we use the shorthand $z_m^n$. Using $F$ as a generic placeholder (typically for a loss function $\ell_i$), we make frequent use of the following assumption.

**Assumption 1.** *The function $F : \mathcal{X} \to \mathbb{R}$ is convex and $G$-Lipschitz, i.e., for all $x, y \in \mathcal{X}$ we have $|F(x) - F(y)| \leq G\|x - y\|$. In addition, the domain $\mathcal{X}$ is a closed and convex set, and it has Euclidean diameter at most $R$.*

Throughout, $N$ denotes the number of losses and, in Section 3, $M$ denotes the number of groups. We use $\epsilon$ for our target accuracy and $r_\epsilon := \epsilon'/G$ for the ball radius, where $\epsilon' = \epsilon/(2\log M)$ for Group-DRO (Section 3) and $\epsilon' = \epsilon/(2\log N)$ for $f$-divergence DRO (Section 4).

**Complexity model** We measure an algorithm's complexity by its *expected* number of $\ell_i$ and $\nabla \ell_i$ evaluations; bounds on expected evaluation number can be readily converted to more standard probability 1 bounds [see 3, Appendix A.3]. Moreover if $\mathcal{X} \subset \mathbb{R}^d$, $d = \Omega(\log N)$,[2] and the time to evaluate $\ell_i$ and $\nabla \ell_i$ is $O(d)$, the expected runtime of all the algorithms we consider is at most $d$ times the evaluation complexity.

### 2.1 Ball oracle acceleration

We now briefly summarize the complexity bounds given by the framework of [13, 15, 3] for accelerated minimization using queries to (inexact) ball optimization oracles, defined as follows.

**Definition 1.** *An algorithm is a Ball Regularized Optimization Oracle of radius $r$ (r-BROO) for function $F : \mathcal{X} \to \mathbb{R}$ if for query point $\bar{x} \in \mathcal{X}$, regularization parameter $\lambda > 0$ and desired accuracy $\delta > 0$ it returns $\mathcal{O}_{\lambda,\delta}(\bar{x}) \in \mathcal{X}$ satisfying*

$$\mathbb{E}\left[F(\mathcal{O}_{\lambda,\delta}(\bar{x})) + \frac{\lambda}{2}\|\mathcal{O}_{\lambda,\delta}(\bar{x}) - \bar{x}\|^2\right] \leq \min_{x \in \mathbb{B}_r(\bar{x}) \cap \mathcal{X}} \left\{F(x) + \frac{\lambda}{2}\|x - \bar{x}\|^2\right\} + \frac{\lambda}{2}\delta^2. \quad (4)$$

**Proposition 1.** *Let $F$ satisfy Assumption 1, let $\mathcal{C}_F$ be the complexity of evaluating $F$ exactly, and let $\mathcal{C}_\lambda(\delta)$ bound the complexity of an r-BROO query with $\delta, \lambda$. Assume that $\mathcal{C}_\lambda(\delta)$ is non-increasing in $\lambda$ and at most polynomial in $1/\delta$. For any $\epsilon > 0$, Algorithm 1 returns $x$ such that*

---

[2]The assumption $d = \Omega(\log N)$ is only necessary for our results on $f$-divergence DRO (Section 4), where the runtime of computing $\operatorname{argmin}_{y \in \mathbb{R}} \Upsilon(x, y)$ is $O(Nd + N\log N)$ due to the need to sort the losses.

$F(x) - \min_{x_\star \in \mathcal{X}} F(x_\star) \leq \epsilon$ with probability at least $\frac{1}{2}$. For $m_\epsilon = O\big(\log \frac{GR^2}{\epsilon r}\big)$ and $\lambda_{\mathrm{m}} = O\big(\frac{m_\epsilon^2 \epsilon}{r^{4/3} R^{2/3}}\big)$, the complexity of the algorithm is

$$O\left( \left(\frac{R}{r}\right)^{2/3} \left[ \left( \sum_{j=0}^{m_\epsilon} \frac{1}{2^j} \mathcal{C}_{\lambda_{\mathrm{m}}} \left( \frac{r}{2^{j/2} m_\epsilon^2} \right) \right) m_\epsilon + (\mathcal{C}_{\lambda_{\mathrm{m}}}(r) + \mathcal{C}_F) m_\epsilon^3, \right] \right). \qquad (5)$$

Informally, the proposition shows that $\widetilde{O}((R/r)^{2/3})$ BROO calls with $\lambda = \widetilde{\Omega}(\epsilon/(r^{3/4} R^{2/3}))$ and accuracy $\delta = \widetilde{O}(r)$ suffice to find an $\epsilon$-accurate solution. As we show in the sequel, for $\mathcal{C}_\lambda(\delta) = \widetilde{O}\big(N + (\frac{G}{\lambda \delta})^2\big)$ the resulting complexity bound is $\widetilde{O}\big(N(\frac{GR}{\epsilon})^{2/3} + (\frac{GR}{\epsilon})^2\big)$. The summation over $j$ in bound (5) stems from the use of MLMC to de-bias the BROO output (i.e., make it exact in expectation): compared to the original proposal of Asi et al. [3], our version of the procedure in Appendix B slightly alters this MLMC scheme by de-biasing one accurate BROO call instead of averaging many inaccurate de-biased calls, improving our bounds by logarithmic factors.

## 3 Group DRO

In this section we develop our BROO implementations for the Group DRO objective (2). In Section 3.1 we describe an "exponentiated group-softmax" function that approximates $\mathcal{L}_{\text{g-DRO}}$ with additive error at most $\epsilon/2$. We then apply stochastic gradient methods on this function to obtain BROO implementations that yield improved rates for Group DRO via Proposition 1: we first consider the non-smooth case in Section 3.2 and then the weakly-smooth case in Section 3.3.

### 3.1 Exponentiated group-softmax

Given a cheap and unbiased stochastic gradient estimator of $\nabla \mathcal{L}_{\text{g-DRO}}$, we could use a variant of SGD and minimize $\mathcal{L}_{\text{g-DRO}}$ to $\epsilon$-suboptimal solution using $O(\epsilon^{-2})$ steps. However, obtaining an unbiased estimator is challenging due to the maximum operator in $\mathcal{L}_{\text{g-DRO}}$. As a first step we use entropy smoothing [7, 8, 5, 4] to replace the maximum in $\mathcal{L}_{\text{g-DRO}}$ with the softmax operation. More specifically, we use the trick from [15] and minimize the "exponentiated softmax" (that has the form of a weighted finite sum) within a small ball. For target accuracy $\epsilon$, regularization parameter $\lambda \geq 0$, center point $\bar{x} \in \mathcal{X}$ and $\epsilon' = \epsilon/(2 \log M) > 0$, the (regularized) group-softmax function is

$$\mathcal{L}_{\text{smax},\epsilon,\lambda}(x) := \epsilon' \log \left( \sum_{i \in [M]} e^{\frac{\mathcal{L}_i(x)}{\epsilon'}} \right) + \frac{\lambda}{2} \|x - \bar{x}\|^2 \text{ where } \mathcal{L}_i(x) = \sum_{j \in [N]} w_{ij} \ell_j(x). \qquad (6)$$

We will implement a BROO for $\mathcal{L}_{\text{smax},\epsilon} := \mathcal{L}_{\text{smax},\epsilon,0}$, which is a uniform approximation of $\mathcal{L}_{\text{g-DRO}}$: $|\mathcal{L}_{\text{g-DRO}}(x) - \mathcal{L}_{\text{smax},\epsilon}(x)| \leq \epsilon/2$ for all $x \in \mathcal{X}$; see Appendix C.1 for details.

The (regularized) exponentiated group-softmax is

$$\Gamma_{\epsilon,\lambda}(x) := \sum_{i \in [M]} \bar{p}_i \gamma_i(x) \text{ where } \gamma_i(x) = \epsilon' e^{\frac{\mathcal{L}_i(x) - \mathcal{L}_i(\bar{x}) + \frac{\lambda}{2}\|x - \bar{x}\|^2}{\epsilon'}} \text{ and } \bar{p}_i = \frac{e^{\frac{\mathcal{L}_i(\bar{x})}{\epsilon'}}}{\sum_{i \in [M]} e^{\frac{\mathcal{L}_i(\bar{x})}{\epsilon'}}}. \qquad (7)$$

In the following lemma we (easily) extend Carmon et al. [15, Lemma 1] to exponentiated group-softmax, showing that $\Gamma_{\epsilon,\lambda}$ is well-behaved inside a ball of (appropriately small) radius $r$ around $\bar{x}$ and facilitates minimizing $\mathcal{L}_{\text{smax},\epsilon,\lambda}$ in that ball; see Appendix C.1 for the proof.

**Lemma 1.** *Let each $\ell_i$ satisfy Assumption 1, and consider the restriction of $\mathcal{L}_{\text{smax},\epsilon,\lambda}$ (6) and $\Gamma_{\epsilon,\lambda}$ (7) to $\mathbb{B}_r(\bar{x})$. Then the functions have the same minimizer $x_\star \in \mathbb{B}_r(\bar{x})$ and, if $\lambda \leq O(G/r)$ and $r \leq O(\epsilon'/G)$, then (a) $\Gamma_{\epsilon,\lambda}$ is $\Omega(\lambda)$-strongly convex, (b) each $\gamma_i$ is $O(G)$-Lipschitz and (c) for every $x \in \mathbb{B}_r(\bar{x})$ we have $\mathcal{L}_{\text{smax},\epsilon,\lambda}(x) - \mathcal{L}_{\text{smax},\epsilon,\lambda}(x_\star) \leq O(\Gamma_{\epsilon,\lambda}(x) - \Gamma_{\epsilon,\lambda}(x_\star))$.*

### 3.2 BROO implementation for Group DRO non-smooth losses

To motivate our BROO implementation, let us review how [15] use the exponentiated softmax in the special case of size-1 groups, i.e., $\mathcal{L}_i = \ell_i$, and explain the difficulty that their approach faces when the group structure is introduced. The BROO implementation in [15] is based on SGD variant with the stochastic gradient estimator $\hat{g}(x) = e^{(\mathcal{L}_i(x) - \mathcal{L}_i(\bar{x}))/\epsilon'} \nabla \mathcal{L}_i(x)$ where $i \sim \bar{p}_i$. However, for

Group DRO where $\mathcal{L}_i = \sum_{j \in [N]} w_{ij} \ell_j$, the estimator $\hat{g}(x)$ can be up to $N$ times more expensive to compute. Approximating $\hat{g}(x)$ by drawing $j, j' \sim w_i$ and taking $e^{(\ell_j(x) - \ell_j(\bar{x}))/\epsilon'} \nabla \ell_{j'}(x)$ will result in a biased estimator since $\mathbb{E}_{j \sim w_i} e^{(\ell_j(x) - \ell_j(\bar{x}))/\epsilon'} \neq e^{(\mathcal{L}_i(x) - \mathcal{L}_i(\bar{x}))/\epsilon'}$. To address this challenge we propose a new gradient estimator based on the multilevel Monte Carlo (MLMC) method [9].

The MLMC unbiased estimator for $\gamma_i(x) = \epsilon' e^{(\mathcal{L}_i(x) - \mathcal{L}_i(\bar{x}))/\epsilon'}$, which we denote by $\widehat{\mathcal{M}}[\gamma_i(x)]$, is defined as follows:

$$\text{Draw } J \sim \text{Geom}\left(1 - \tfrac{1}{\sqrt{8}}\right), S_1, \ldots, S_n \overset{\text{iid}}{\sim} w_i \text{ and let } \widehat{\mathcal{M}}[\gamma_i(x)] := \widehat{\gamma}(x; S_1) + \frac{\widehat{\mathcal{D}}_{2^J}}{p_J},$$

where $p_j := \mathbb{P}(J = j) = \left(1/\sqrt{8}\right)^j \left(1 - \tfrac{1}{\sqrt{8}}\right)$ and, for $n \in 2\mathbb{N}$, we define

$$\widehat{\mathcal{D}}_n := \widehat{\gamma}(x; S_1^n) - \frac{\widehat{\gamma}\left(x; S_1^{n/2}\right) + \widehat{\gamma}\left(x; S_{n/2+1}^n\right)}{2} \text{ and } \widehat{\gamma}(x; S_1^n) := \epsilon' e^{\frac{1}{n} \sum_{j=1}^{n} \frac{\ell_{S_j}(x) - \ell_{S_j}(\bar{x}) + \frac{\lambda}{2} \|x - \bar{x}\|^2}{\epsilon'}}.$$

With the MLMC estimator for $\gamma_i$ in hand, we estimate the gradient of $\Gamma_{\epsilon, \lambda}$ as follows:

$$\text{Draw } i \sim p(\bar{x}), \, j \sim w_i \text{ and set } \hat{g}(x) = \frac{1}{\epsilon'} \widehat{\mathcal{M}}[\gamma_i(x)](\nabla \ell_j(x) + \lambda(x - \bar{x})). \tag{8}$$

In the following lemma we summarize the important properties of the MLMC and gradient estimators; see Appendix C.2 for the proof.

**Lemma 2.** *Let each $\ell_i$ satisfy Assumption 1, and let $r \leq \frac{\epsilon'}{G}$, $\lambda \leq \frac{G}{r}$ and $x \in \mathbb{B}_r(\bar{x})$. Then $\widehat{\mathcal{M}}[\gamma_i(x)]$ and $\hat{g}(x)$ are unbiased for $\gamma_i(x)$ and $\nabla \Gamma_{\epsilon, \lambda}(x)$, respectively, and have bounded second moments: $\mathbb{E}\left[\widehat{\mathcal{M}}[\gamma_i(x)]\right]^2 \leq O\left(\frac{G^4 \|x - \bar{x}\|^4}{\epsilon'^2} + \epsilon'^2\right)$ and $\mathbb{E}\|\hat{g}(x)\|^2 \leq O(G^2)$. In addition, the complexity of computing $\widehat{\mathcal{M}}[\gamma_i(x)]$ and $\hat{g}(x)$ is $O(1)$.*

Due to Lemma 2 and since $\Gamma_{\epsilon, \lambda}$ is $\Omega(\lambda)$-strongly convex, we can use the Epoch-SGD algorithm of Hazan and Kale [28] with our gradient estimator (8). This algorithm has rate of convergence $O(G^2/(\lambda T))$ and our gradient estimator requires additional $N$ function evaluations for precomputing the sampling probabilities $\{\bar{p}_i\}$. We thus arrive at the following complexity bound.

**Theorem 1.** *Let each $\ell_j$ satisfy Assumption 1, let $\epsilon, \delta, \lambda > 0$ and let $r_\epsilon = \epsilon/(2G \log M)$. For any query point $\bar{x} \in \mathbb{R}^d$, regularization strength $\lambda \leq O(G/r_\epsilon)$ and accuracy $\delta$, EpochSGD [28, Algorithm 1]) with the gradient estimator (8) outputs a valid $r_\epsilon$-BROO response and has complexity $\mathcal{C}_\lambda(\delta) = O\left(N + \frac{G^2}{\lambda^2 \delta^2}\right)$. Consequently, the complexity of finding an $\epsilon$-suboptimal minimizer of $\mathcal{L}_{\text{g-DRO}}$ (2) with probability at least $\frac{1}{2}$ is*

$$O\left(N\left(\frac{GR}{\epsilon}\right)^{2/3} \log^{11/3} H + \left(\frac{GR}{\epsilon}\right)^2 \log^2 H\right) \text{ where } H := M \frac{GR}{\epsilon}.$$

We provide the proof for Theorem 1 in Appendix C.3; the final complexity bound follows from straightforward calculations which we now briefly outline. According to Proposition 1, finding an $\frac{\epsilon}{2}$-suboptimal solution for $\mathcal{L}_{\text{smax}, \epsilon}$ (and consequently an $\epsilon$-suboptimal solution for $\mathcal{L}_{\text{g-DRO}}$) involves $\widetilde{O}\left((R/r_\epsilon)^{2/3}\right)$ BROO calls with accuracy $\delta = \widetilde{\Omega}\left(r_\epsilon 2^{-J/2}\right)$ and regularization strength $\lambda \geq \lambda_{\text{m}}$, where $J = \min\{\text{Geom}(\frac{1}{2}), m\}$. We may therefore bound the complexity of each such call by

$$\sum_{j=0}^{m} 2^{-j} \mathcal{C}_{\lambda_{\text{m}}}\left(r_\epsilon 2^{-j/2}\right) = \sum_{j=0}^{m} 2^{-j} \widetilde{O}\left(N + \frac{2^j G^2}{\lambda_{\text{m}}^2 r_\epsilon^2}\right) \overset{(\star)}{=} \widetilde{O}\left(N + \left(\frac{GR}{\epsilon}\right)^2 \left(\frac{r_\epsilon}{R}\right)^{2/3}\right),$$

where $(\star)$ follows from substituting $\lambda_{\text{m}} = \widetilde{\Omega}\left(\epsilon r_\epsilon^{-4/3} R^{-2/3}\right)$ and $m = \widetilde{O}(1)$. Multiplying this bound by $\widetilde{O}\left((R/r_\epsilon)^{2/3}\right)$ yields (up to polylogarithmic factors) the conclusion of Theorem 1.

## 3.3 Accelerated variance reduction for mean-square smooth losses

In this section we provide an algorithm with an improved rate of convergence under the following mean-square smoothness assumption.

**Assumption 2.** *For all $x, x' \in \mathbb{B}_r(\bar{x})$ and $i \in [M]$, $\mathbb{E}_{j \sim w_i} \|\nabla \ell_j(x) - \nabla \ell_j(x')\|^2 \leq L^2 \|x - x'\|^2$.*

Note that assuming $L$-Lipschitz gradient for each $\ell_i$ implies Assumption 2, but not the other way around. To take advantage of Assumption 2, we first rewrite the function $\Gamma_{\epsilon,\lambda}(x)$ in a way that is more amenable to variance reduction:

$$\Gamma_{\epsilon,\lambda}(x) := \sum_{i \in [M]} c_{x',\bar{x}} p_i(x') \gamma_i(x, x'), \quad \text{where} \quad \gamma_i(x, x') := \epsilon' e^{\frac{\mathcal{L}_i(x) - \mathcal{L}_i(x') + \frac{\lambda}{2}\|x - \bar{x}\|^2}{\epsilon'}},$$

$$c_{x',\bar{x}} = \left( \frac{\sum_{j \in [M]} e^{\frac{\mathcal{L}_j(x')}{\epsilon'}}}{\sum_{j \in [M]} e^{\frac{\mathcal{L}_j(\bar{x})}{\epsilon'}}} \right) \quad \text{and} \quad p_i(x') := \frac{e^{\frac{\mathcal{L}_i(x')}{\epsilon'}}}{\sum_{j \in [M]} e^{\frac{\mathcal{L}_j(x')}{\epsilon'}}}.$$

(Note that $\gamma_i(x, \bar{x}) = \gamma_i(x)$). Given a reference point $x'$, to compute a reduced-variance estimator of $\nabla \Gamma_{\epsilon,\lambda}(x)$, we draw $i \sim p_i(x')$ and $j \sim w_i$, and set:

$$\hat{g}_{x'}(x) := \nabla \Gamma_{\epsilon,\lambda}(x') + \frac{c_{x',\bar{x}}}{\epsilon'} \left[ \widehat{\mathcal{M}}[\gamma_i(x, x')] \nabla \ell_j^\lambda(x) - \gamma_i(x', x') \nabla \ell_j^\lambda(x') \right] \quad (9)$$

where $\nabla \ell_j^\lambda(x) := \nabla \ell_j(x) + \lambda(x - \bar{x})$ and $\widehat{\mathcal{M}}[\gamma_i(x, x')]$ is an MLMC estimator for $\gamma_i(x, x')$ defined analogously to $\widehat{\mathcal{M}}[\gamma_i(x)]$ (see details in Appendix C.4). The estimator (9) is not precisely standard SVRG [33] since we use $\widehat{\mathcal{M}}[\gamma_i(x, x')]$ as an estimator for $\gamma_i(x, x')$. Simple calculations show that $\mathbb{E}\hat{g}_{x'}(x) = \nabla \Gamma_{\epsilon,\lambda}(x)$ and the following lemma shows that $\hat{g}$ satisfies a type of variance bound conducive to variance-reduction schemes; see Appendix C.4 for the proof.

**Lemma 3.** *Let each $\ell_j$ satisfy Assumptions 1 and 2. For any $\lambda \leq \frac{G}{r}$, $r = \frac{\epsilon'}{G}$ and $x, x' \in \mathbb{B}_r(\bar{x})$, the variance of $\hat{g}_{x'}(x)$ is bounded by $\mathrm{Var}(\hat{g}_{x'}(x)) \leq O\left( \left( L + \lambda + \frac{G^2}{\epsilon'} \right)^2 \|x - x'\|^2 \right).$*

Accelerated variance reduction methods for convex functions typically require a stronger variance bound of the form $\mathrm{Var}(\hat{g}_{x'}(x)) \leq 2L(F(x') - F(x) - \langle \nabla F(x), x' - x \rangle)$ for every $x$ [cf. 1, Lemma 2.4]. The guarantee of Lemma 3 is weaker, but still allows for certain accelerated rates via, e.g., the Katyusha X algorithm [2]. With it, we obtain the following guarantee.

**Theorem 2.** *Let each $\ell_j$ satisfy Assumptions 1 and 2. Let $\epsilon > 0$, $\epsilon' = \epsilon/(2 \log M)$ and $r_\epsilon = \epsilon'/G$. For any query point $\bar{x} \in \mathbb{R}^d$, regularization strength $\lambda \leq O(G/r_\epsilon)$ and accuracy $\delta$, KatyushaX$^\mathrm{s}$ [2, Algorithm 2] with the gradient estimator (9) outputs a valid $r_\epsilon$-BROO response and has complexity $\mathcal{C}_\lambda(\delta) = O\left( \left( N + \frac{N^{3/4}(G + \sqrt{\epsilon' L})}{\sqrt{\lambda \epsilon'}} \right) \log\left( \frac{Gr_\epsilon}{\lambda \delta^2} \right) \right)$. Consequently, the complexity of finding an $\epsilon$-suboptimal minimizer of $\mathcal{L}_{\text{g-DRO}}$ (2) with probability at least $\frac{1}{2}$ is*

$$O\left( N \left( \frac{GR}{\epsilon} \right)^{2/3} \log^{14/3} H + N^{3/4} \left( \frac{GR}{\epsilon} + \sqrt{\frac{LR^2}{\epsilon}} \right) \log^{7/2} H \right) \quad \text{where} \quad H := M \frac{GR}{\epsilon}.$$

We provide the proof of Theorem 2 in Appendix C.5. For the special case of Group DRO with a single group satisfying Assumption 2 with $L = \Theta(G^2/\epsilon)$, i.e. minimizing the average loss, we have the lower bound $\widetilde{\Omega}(N + N^{3/4} \frac{GR}{\epsilon})$ [62] and for the case of $N$ distinct groups, i.e. minimizing the maximal loss, we have the lower bound $\widetilde{\Omega}(N\epsilon^{-2/3})$ [15]. This implies that in the weakly mean-square smooth setting the term scaling as $N^{3/4}\epsilon^{-1}$ and the term scaling as $N\epsilon^{-2/3}$ are unimprovable.

## 4 DRO with $f$-divergence

In this section we develop our BROO implementation for the $f$-divergence objective (3). In Section 4.1 we reduce the original DRO problem to a regularized form using Lagrange multipliers. Next, in Section 4.2 we show that adding negative entropy regularization to the objective produces the stability properties necessary for efficient ball optimization. In Section 4.3 we describe a BROO implementation for the non-smooth case using a variant of Epoch-SGD [28], and in Section 4.4 we implement the BROO under a weak-smoothness assumption by carefully restarting an accelerated variance reduction method [1].

## 4.1 The dual problem

We first note that (due to Slater's condition), by Lagrange duality, the objective (3) is equivalent to

$$\mathcal{L}_{f\text{-div}}(x) := \max_{q \in \Delta^N : \sum_{i \in [N]} \frac{f(Nq_i)}{N} \leq 1} \sum_{i \in [N]} q_i \ell_i(x) = \min_{\nu \geq 0} \left\{ \nu + \max_{q \in \Delta^N} \sum_{i \in [N]} \left( q_i \ell_i(x) - \frac{\nu}{N} f(Nq_i) \right) \right\}.$$

Writing $\psi(s) := \frac{\nu}{N} f(Ns)$ for $\psi : \mathbb{R}_+ \to \mathbb{R}$, we therefore consider objectives of the form

$$\mathcal{L}_\psi(x) := \max_{q \in \Delta^N} \sum_{i \in [N]} (q_i \ell_i(x) - \psi(q_i)) = \min_{y \in \mathbb{R}} \left\{ \Upsilon(x, y) := \sum_{i \in [N]} \psi^*(\ell_i(x) - Gy) + Gy \right\} \quad (10)$$

where $G$ is the Lipschitz constant of each loss $\ell_i$ and for the last equality we use Lagrange duality, with $\psi^*(v) := \max_{t \in \text{dom}(\psi)} \{vt - \psi(t)\}$ the Fenchel dual of $\psi$ (for more details see Appendix D.1). We show that under weak assumptions (introducing logarithmic dependence on bounds on $f$ and the losses) we can solve the constrained problem (3) to accuracy $\epsilon$ by computing a polylogarithmic number of $O(\epsilon)$-accurate minimizers of (10); see Appendix D.2 for details. Since the complexity of solving (10) holds for any $\nu > 0$, and we have a lower bound for the required $\nu$, for the remainder of this section we focus on minimizing $\mathcal{L}_\psi$ for *arbitrary* convex $\psi$.

## 4.2 Stabilizing the gradient estimator

While minimizing (10) can be viewed as ERM (over $x$ and $y$), straightforward application of SGD does not solve it efficiently. To see this, consider the standard gradient estimator formed by sampling $i \sim \text{Unif}([N])$ and taking $\hat{g}^{\text{x}} = N\psi^{*'}(\ell_i(x) - Gy)\nabla \ell_i(x)$ and $\hat{g}^{\text{y}} = G(1 - N\psi^{*'}(\ell_i(x) - Gy))$. For general $\psi$, this estimator will have unbounded second moments, and therefore SGD using them would lack a convergence guarantee. As an extreme example, consider $\psi = 0$ (corresponding to minimizing the maximum loss) whose conjugate function $\psi^*(v)$ is 0 for $v \leq 0$ and $\infty$ for $v > 0$, leading to meaningless stochastic gradients.

We obtain bounded gradient estimates in two steps. First, we find a better distribution for $i$ using a reference point $\bar{x} \in \mathcal{X}$ with corresponding $\bar{y} = \text{argmin}_{y \in \mathbb{R}} \Upsilon(\bar{x}, y)$. Namely, we note that the optimality condition for $\bar{y}$ implies that $\psi^{*'}(\ell_i(\bar{x}) - G\bar{y})$ is a pmf over $[N]$. Therefore, we may sample $i \sim \psi^{*'}(\ell_i(\bar{x}) - G\bar{y})$ and estimate the gradient of $\Upsilon$ at $(x, y)$ using $\hat{g}^{\text{x}} = \rho_i(x, y)\nabla \ell_i(x)$ and $\hat{g}^{\text{y}} = G(1 - \rho_i(x, y))$, where $\rho_i(x, y) = \frac{\psi^{*'}(\ell_i(x) - Gy)}{\psi^{*'}(\ell_i(\bar{x}) - G\bar{y})}$. However, for general $\psi$ (and $\psi = 0$ in particular), the ratio $\rho_i(x, y)$ can be unbounded even when $x, y$ are arbitrarily close to $\bar{x}, \bar{y}$.

Our second step ensures that $\rho_i(x, y)$ is bounded around $\bar{x}, \bar{y}$ by adding a small negative entropy term to $\psi$, defining

$$\psi_\epsilon(q) := \psi(q) + \epsilon' q \log q \quad \text{where} \quad \epsilon' := \frac{\epsilon}{2 \log N}, \quad (11)$$

and

$$\mathcal{L}_{\psi, \epsilon}(x) = \min_{y \in \mathbb{R}} \Upsilon_\epsilon(x, y) \quad \text{with} \quad \Upsilon_\epsilon(x, y) := \sum_{i \in [N]} \psi_\epsilon^*(\ell_i(x) - Gy) + Gy. \quad (12)$$

Due to our choice of $\epsilon'$, we have $|\mathcal{L}_\psi(x) - \mathcal{L}_{\psi,\epsilon}(x)| \leq \epsilon/2$ for all $x \in \mathbb{R}^d$, and consequently an $\epsilon/2$-accurate minimizer of $\mathcal{L}_{\psi,\epsilon}$ is also an $\epsilon$-accurate for $\mathcal{L}_\psi$ (see Lemma 18 in Appendix D.3). When $\psi = 0$ we have $\psi_\epsilon^*(v) = e^{(v-1)/\epsilon'}$ and therefore the corresponding $\rho_i(x, y) = e^{(\ell_i(x) - \ell_i(\bar{x}) - G(y - \bar{y}))/\epsilon'}$. The following lemma, which might be of independent interest, shows that the same conclusion holds for any convex $\psi$.

**Lemma 4.** *For any convex $\psi : \mathbb{R}_+ \to \mathbb{R}$ and $\psi_\epsilon$ defined in (10), $\log(\psi_\epsilon^{*'}(\cdot))$ is $\frac{1}{\epsilon'}$-Lipschitz.*

See proof in Appendix D.4. Thus, $\psi_\epsilon^{*'}(v)/\psi_\epsilon^{*'}(\bar{v}) = e^{\log \psi_\epsilon^{*'}(v) - \log \psi_\epsilon^{*'}(\bar{v})} \leq e^{(v-\bar{v})/\epsilon'}$ and $\rho_i(x, y) \leq e^{(\ell_i(x) - \ell_i(\bar{x}) - G(y - \bar{y}))/\epsilon'}$ continues to hold. Therefore, if $|y - \bar{y}| \leq \epsilon'/G = r_\epsilon$ and $x \in \mathbb{B}_{r_\epsilon}(\bar{x})$ (so that $|\ell_i(x) - \ell_i(\bar{x})| \leq \epsilon'$ if $\ell_i$ satisfies Assumption 1), we have the bound $\rho_i(x, y) \leq e^2$.

It remains to show that we may indeed restrict $y$ to be within distance $r_\epsilon$ from $\bar{y}$. To this end, we make the following observation which plays a key part in our analysis and might also be of independent interest (see proof in Appendix D.4).

**Lemma 5.** *For $G > 0$, $\ell(x) = (\ell_1(x), \ldots, \ell_N(x))$ and $y^\star(x) = \operatorname{argmin}_{y \in \mathbb{R}} \Upsilon_\epsilon(x, y)$, we have $|y^\star(x) - y^\star(x')| \leq \frac{1}{G} \|\ell(x) - \ell(x')\|_\infty$ for all $x, x' \in \mathcal{X}$. Moreover, if each $\ell_i$ is $G$-Lipschitz, we have $|y^\star(x) - y^\star(x')| \leq \|x - x'\|$.*

Lemma 5 implies that $x^\star, y^\star = \operatorname{argmin}_{x \in \mathbb{B}_{r_\epsilon}(\bar{x}), y \in \mathbb{R}} \Upsilon_\epsilon(x, y)$ satisfy $|y^\star - \bar{y}| \leq \|x^\star - \bar{x}\| \leq r_\epsilon$. Therefore, when minimizing $\Upsilon_\epsilon$ (or any regularized version of it) inside the ball $\mathbb{B}_{r_\epsilon}(\bar{x})$, we may restrict $y$ to $[\bar{y} - r_\epsilon, \bar{y} + r_\epsilon]$ without loss of generality. We also note that Lemma 5 holds for all values of $\epsilon$ and is therefore valid even without entropy regularization (as long as $\psi^*$ is strongly convex $y^\star$ is unique, and if $y^\star$ is not unique then we can still choose $y^\star$ such that the bound of this lemma holds).

### 4.3 BROO implementation for $f$-divergence DRO with non-smooth losses

By the discussion above, to implement a BROO for $\mathcal{L}_{\psi, \epsilon}(x)$ (with radius $r_\epsilon = \epsilon'/G$, regularization $\lambda$, and query $\bar{x} \in \mathcal{X}$) it suffices to minimize $\Upsilon_{\epsilon, \lambda}(x, y) := \Upsilon_\epsilon(x, y) + \frac{\lambda}{2} \|x - \bar{x}\|^2$ over $x \in \mathbb{B}_{r_\epsilon}(\bar{x})$ and $y \in [\bar{y} - r_\epsilon, \bar{y} + r_\epsilon]$, where $\bar{y} = \operatorname{argmin}_{y \in \mathbb{R}} \Upsilon_\epsilon(\bar{x}, y)$. To that end we estimate the gradient of $\Upsilon_{\epsilon, \lambda}(x, y)$ as follows. Letting $\bar{p}_i = \psi_\epsilon^{*\prime}(\ell_i(\bar{x}) - G\bar{y})$ (making $\bar{p}$ a pmf by optimality of $\bar{y}$), we sample $i \sim \bar{p}$ and set

$$\hat{g}^{\mathrm{x}}(x, y) = \frac{\psi_\epsilon^{*\prime}(\ell_i(x) - Gy)}{\bar{p}_i} \nabla \ell_i(x, y) \ \text{ and } \ \hat{g}^{\mathrm{y}}(x, y) = G\left(1 - \frac{\psi_\epsilon^{*\prime}(\ell_i(x) - Gy)}{\bar{p}_i}\right). \tag{13}$$

Lemma 4 implies the following bounds on our gradient estimator; see proof in Appendix D.5.

**Lemma 6.** *Let each $\ell_i$ be $G$-Lipschitz, let $\bar{x} \in \mathcal{X}$ and $\bar{y} = \operatorname{argmin}_{y \in \mathbb{R}} \Upsilon_{\epsilon, \lambda}(\bar{x}, y)$. Let $r_\epsilon = \frac{\epsilon'}{G}$, then for all $x \in \mathbb{B}_{r_\epsilon}(\bar{x})$ and $y \in [\bar{y} - r_\epsilon, \bar{y} + r_\epsilon]$, the gradient estimators $\hat{g}^{\mathrm{x}}$ and $\hat{g}^{\mathrm{y}}$ satisfy the following:*

*1. $\mathbb{E}_{i \sim \bar{p}_i}[\hat{g}^{\mathrm{x}}(x, y)] = \nabla_x \Upsilon_\epsilon(x, y) \ \text{ and } \ \mathbb{E}_{i \sim \bar{p}_i}[\hat{g}^{\mathrm{y}}(x, y)] = \nabla_y \Upsilon_\epsilon(x, y)$.*

*2. $\mathbb{E}_{i \sim \bar{p}_i}\|\hat{g}^{\mathrm{x}}(x, y)\|^2 \leq e^4 G^2 \ \text{ and } \ \mathbb{E}_{i \sim \bar{p}_i}|\hat{g}^{\mathrm{y}}(x, y)|^2 \leq e^4 G^2$.*

To implement the BROO using our gradient estimator we develop a variant of the Epoch-SGD algorithm of Hazan and Kale [28] (Algorithm 3 in Appendix D.5). Similarly to Epoch-SGD, we apply standard SGD on $\Upsilon_{\epsilon, \lambda}$ (with gradient estimator (13)) in "epochs" whose length doubles in every repetition. Our algorithm differs slightly in how each epoch is initialized. Standard Epoch-SGD initializes with the average of the previous epoch's iterates, and strong convexity shows that the suboptimality and distance to the optimum shrink by a constant factor after every epoch. However, since $\Upsilon_{\epsilon, \lambda}$ is strongly convex only in $x$ and not in $y$, we cannot directly use this scheme. Instead, we set the initial $y$ variable to be $\operatorname{argmin}_y \Upsilon_{\epsilon, \lambda}(x', y)$, where $x'$ is the initial $x$ variable still defined as the previous epoch's average; this initialization has complexity $N$, but we only preform it a logarithmic number of times. Using our initialization scheme and Lemma 5, we recover the original Epoch-SGD contraction argument, yielding the following complexity bound (see proof in Appendix D.5).

**Theorem 3.** *Let each $\ell_i$ satisfy Assumption 1. Let $\epsilon, \lambda, \delta > 0$, and $r_\epsilon = \epsilon/(2G \log N)$. For any query point $\bar{x} \in \mathbb{R}^d$, regularization strength $\lambda \leq O(G/r_\epsilon)$ and accuracy $\delta < r_\epsilon/2$, Algorithm 3 outputs a valid $r_\epsilon$-BROO response for $\mathcal{L}_{\psi, \epsilon}$ and has complexity $\mathcal{C}_\lambda(\delta) = O\left(\frac{G^2}{\lambda^2 \delta^2} + N \log\left(\frac{r_\epsilon}{\delta}\right)\right)$. Consequently, the complexity of finding an $\epsilon$-suboptimal minimizer of $\mathcal{L}_\psi$ (10) with probability at least $\frac{1}{2}$ is*

$$O\left(N\left(\frac{GR}{\epsilon}\right)^{2/3} \log^{11/3} H + \left(\frac{GR}{\epsilon}\right)^2 \log^2 H\right) \ \text{ where } \ H := N\frac{GR}{\epsilon}.$$

### 4.4 Accelerated variance reduction for smooth losses

In this section, we take advantage of the following smoothness assumption.

**Assumption 3.** *For every $i \in [N]$ the loss $\ell_i$ is $L$-smooth, i.e., has $L$-Lipschitz gradient.*

Let us rewrite $\Upsilon_{\epsilon, \lambda}$ in a form that is more amenable to variance reduction techniques:

$$\Upsilon_{\epsilon, \lambda}(x, y) = \sum_{i \in [N]} \bar{p}_i \upsilon_i(x, y) \ \text{ where } \ \upsilon_i(x, y) := \frac{\psi_\epsilon^*(\ell_i(x) - Gy)}{\bar{p}_i} + Gy + \frac{\lambda}{2} \|x - \bar{x}\|^2$$

and, as before $\bar{p}_i = \psi_\epsilon^*(\ell_i(\bar{x}) - G\bar{y})$ for some ball center $\bar{x} \in \mathcal{X}$ and $\bar{y} = \operatorname{argmin}_{y \in \mathbb{R}} \Upsilon_\epsilon(\bar{x}, y)$. In the following lemma, we bound the smoothness of the functions $\upsilon_i$, deferring the proof to Appendix D.6.

**Lemma 7.** *For any $i \in [N]$, let $\ell_i$ be G-Lipschitz and L -smooth, let $r_\epsilon = \frac{\epsilon'}{G}$ and $\lambda = O\left(\frac{G}{r_\epsilon}\right)$. The restriction of $\upsilon_i$ to $x \in \mathbb{B}_{r_\epsilon}(\bar{x})$ and $y \in [\bar{y} - r_\epsilon, \bar{y} + r_\epsilon]$ is $O(G)$-Lipschitz and $O\left(L + \frac{G^2}{\epsilon'}\right)$-smooth.*

Since $\Upsilon_{\epsilon,\lambda}$ is a finite sum of smooth functions, we can obtain reduced-variance gradient estimates by the standard SVRG technique [33]. For any reference point $x', y'$, the estimator is

$$\hat{g}_{x',y'}(x,y) = \nabla \Upsilon_\epsilon(x', y') + \nabla \upsilon_i(x,y) - \nabla \upsilon_i(x', y'), \tag{14}$$

where $\nabla$ is with respect to the vector $[x, y]$. Similar to non-smooth case, obtaining an efficient BROO implementation is complicated by the fact that $\Upsilon_{\epsilon,\lambda}$ is strongly-convex in $x$ but not in $y$. Our solution is also similar: we propose a restart scheme and minimize over $y$ exactly between restarts (Algorithm 4 in Appendix D.6), that gives the following complexity bound (see proof in Appendix D.6).

**Theorem 4.** *Let each $\ell_i$ satisfy Assumptions 1 and 3, let $\epsilon, \lambda, \delta > 0$, and $r_\epsilon = \frac{\epsilon}{2G \log N}$. For any query point $\bar{x} \in \mathbb{R}^d$, regularization strength $\lambda \leq O(\frac{G}{r_\epsilon})$ and accuracy $\delta$, Algorithm 4 outputs a valid $r_\epsilon$-BROO response for $\mathcal{L}_{\psi,\epsilon}$ and has complexity $\mathcal{C}_\lambda(\delta) = O\left(\left(N + \frac{\sqrt{N}(G + \sqrt{\epsilon' L})}{\sqrt{\lambda \epsilon'}}\right) \log \frac{Gr_\epsilon}{\lambda \delta^2}\right)$. Consequently, the complexity of finding an $\epsilon$-suboptimal minimizer of $\mathcal{L}_\psi$ (10) with probability at least $\frac{1}{2}$ is*

$$O\left(N\left(\frac{GR}{\epsilon}\right)^{2/3} \log^{14/3} H + \sqrt{N}\left(\frac{GR}{\epsilon} + \sqrt{\frac{LR^2}{\epsilon}}\right) \log^{5/2} H\right) \quad \text{where } H := N\frac{GR}{\epsilon}.$$

## 5   Discussion

**Limitations**  While our work indicates that the ball optimization approach offers significant complexity gains for DRO, we note that turning the algorithms we propose into practical DRO methods faces several challenges. A main challenge is the costly bisection procedure common to all Monteiro-Svaiter-type acceleration schemes [41, 25, 13, 52]. Fortunately, very recently, two works [16, 36] (the former partially motivated by our paper) have shown how to remove the bisection from Monteiro-Svaiter schemes, significantly improving the practical potential of the methods we propose. However, another practical limitation of our approach is the need to tune many parameters that are not known in advance, such as those relating to the ball radius $r_\epsilon$ and smoothing level $\epsilon'$, as well as step sizes and number of iterations of oracle implementations; a more adaptive setting for these parameters is likely important.

**Extensions**  First, it would be interesting to extend our approach to DRO objectives $\max_{q \in \mathcal{U}} \sum_{i \in [N]} q_i \ell_i(x)$ with uncertainty set $\mathcal{U}$ that is an arbitrary subset of the simplex. While the subgradient method, the primal-dual method (Appendix A.1), and "AGD on softmax" (Appendix A.2) all apply to any $\mathcal{U} \subseteq \Delta^N$, our methods strongly rely on the structure of $\mathcal{U}$ induced by Group- $f$-divergence DRO, and extending them to unstructured $\mathcal{U}$'s seems challenging.

Second, it would be interesting to generalize our results in the "opposite" direction of getting better complexity bounds for problems with additional structure. For Group-DRO our bounds are essentially optimal when the number of groups $M = \Omega(N)$, but are suboptimal when $M = O(1)$. We leave it as a question for further research if it is possible to obtain a stronger bound such as $\widetilde{O}(N + M\epsilon^{-2/3} + \epsilon^{-2})$, which recovers our result for $M = N$ but improves on it for smaller values of $M$. Taking CVaR at level $\alpha$ as a special case of $f$-divergence DRO, our bounds are optimal when $\alpha$ is close to $1/N$ but suboptimal for larger value of $\alpha$; it would be interesting to obtain bounds such as $\widetilde{O}(N + \alpha^{-1}\epsilon^{-2/3} + \epsilon^{-2})$.

A third possible extension of our research is DRO in the non-convex setting. For this purpose, it might be possible to use the technique of Carmon et al. [12] for turning accelerated convex optimization algorithms to improved-complexity methods for smooth non-convex optimization.

## Acknowledgments

We thank Marc Teboulle for detailed and helpful feedback. This work was supported by Israeli Science Foundation (ISF) grant no. 2486/21, the Len Blavatnik and the Blavatnik Family foundation and the Adelis Foundation.

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
