## A  Alternative algorithms for solving DRO problems

### A.1  Primal-dual stochastic mirror descent

In this section, we present a primal-dual method capable of solving all the DRO problems our paper considers, under an additional assumption of bounded losses: for every $j$ and $x$ we assume $|\ell_j(x)| \leq B_\ell$. Consider the primal-dual problem

$$\underset{x \in \mathcal{X}}{\text{minimize}} \max_{q \in \mathcal{U}} \left\{ \mathcal{L}_{\text{pd}}(x, q) \coloneqq \sum_{i \in [m]} q_i \mathcal{L}_i(x) \right\} \tag{15}$$

where $\mathcal{X} \subseteq B_R(x_0)$ is a closed convex set as before and $\mathcal{U}$ is now an arbitrary closed convex subset of the simplex $\Delta^m$ and $\mathcal{L}_i(x) = \sum_{j \in [N]} w_{ij} \ell_j(x)$ are "group losses" with $w_i \in \Delta^N$ for every $i \in [m]$. This formulation subsumes both Group DRO (where $m = M$ and $\mathcal{U} = \Delta^M$) and $f$-divergence DRO (where $m = N$, $\mathcal{L}_i(x) = \ell_i(x)$, and $\mathcal{U}$ is an $f$-divergence ball).

As discussed in the introduction, several works have proposed primal-dual methods for DRO, but we could not find in these works the precise rate we prove here (in Proposition 2 below) in its full generality. Our proof is a straightforward specialization of the more general results of Carmon et al. [14].

The particular algorithm we consider is primal-dual stochastic mirror descent, with distances generated by the squared Euclidean norm on $\mathcal{X}$ and entropy on $\mathcal{U}$ and gradient clipping for the $\mathcal{U}$ iterates, corresponding to the following recursion:

$$x_{t+1} = \underset{x \in \mathcal{X}}{\arg\min} \left\{ \langle \eta \hat{g}^{\text{x}}(x_t, q_t), x \rangle + \frac{\log m}{R^2} \|x - x_t\|^2 \right\} \text{ and}$$

$$q_{t+1} = \underset{x \in \mathcal{X}}{\arg\max} \left\{ \langle \Pi_{[-1,1]^m}(\eta \hat{g}^{\text{q}}(x_t, q_t)), q \rangle + \sum_{i \in [m]} [q]_i \log \frac{[q]_i}{[q_t]_i} \right\}, \tag{16}$$

where $\eta$ is a step-size parameter, $\Pi_{[-1,1]^m}$ is the Euclidean projection to the unit box (i.e., entry-wise clipping to $[-1, 1]$), and $\hat{g}^{\text{x}}$ and $\hat{g}^{\text{q}}$ are unbiased estimators for $\nabla_x \mathcal{L}_{\text{pd}}$ and $\nabla_q \mathcal{L}_{\text{pd}}$, respectively, given by

$$\hat{g}^{\text{x}}(z) \coloneqq \nabla \ell_j(z^{\text{x}}) \text{ with } i \sim z^q \text{ and } j \sim w_i \tag{17}$$
$$\hat{g}^{\text{q}}(z) \coloneqq m \ell_j(z^{\text{x}}) e_i \text{ with } i \sim \text{Unif}([m]) \text{ and } j \sim w_i,$$

with $e_i \in \mathbb{R}^m$ being the $i$th standard basis vector in $\mathbb{R}^m$.

This method yields the following convergence guarantees.

**Proposition 2.** *Assume that each $\ell_j$ convex and $G$-Lipschitz and satisfies $|\ell_j(x)| \leq B_\ell$ for every $x \in \mathcal{X}$. For $T \in \mathbb{N}$ let $\bar{x}_T = \frac{1}{T} \sum_{t=0}^{T} x_t$ and $\bar{q}_T = \frac{1}{T} \sum_{t=0}^{T} q_t$, where $\{x_t, q_t\}$ are the iterates defined in (16), with $\eta = O\left( \frac{\epsilon \log m}{G^2 R^2 + m B_\ell^2} \right)$. Then, for any $\epsilon > 0$, if $T \geq O\left( \frac{G^2 R^2 + B_\ell^2 m \log m}{\epsilon^2} \right)$ we have that*

$$\mathbb{E} \mathcal{L}_{\text{DRO}}(\bar{x}_T) - \min_{x_\star \in \mathcal{X}} \mathcal{L}_{\text{DRO}}(x_\star) \leq \mathbb{E} \max_{q \in \mathcal{U}} \mathcal{L}_{\text{pd}}(\bar{x}_T, q) - \mathbb{E} \min_{x \in \mathcal{X}} \mathcal{L}_{\text{pd}}(x, \bar{q}_T) \leq \epsilon,$$

*where $\mathcal{L}_{\text{DRO}}(x) = \max_{q \in \mathcal{U}} \mathcal{L}_{\text{pd}}(x, q)$.*

*Proof.* The proposition is a direct corollary of a more general result by Carmon et al. [14]. To show this, we rewrite the iterations (16) using "local norm setup" notation of [14]. In particular, let $\mathcal{U} = \mathcal{X} \times \mathcal{U}$ and for every $z = (z^{\text{x}}, z^{\text{q}}) \in \mathcal{U}$ define the local norm of $\delta \in \mathcal{U}^*$ at $z$ as

$$\|\delta\|_z \coloneqq \sqrt{\frac{R^2}{\log m} \|\delta^{\text{x}}\|_2^2 + \sum_{i \in [m]} [z^q]_i [\delta^q]_i^2}.$$

In addition, we define the generating distance function

$$r(z) = r(z^{\text{x}}, z^{\text{q}}) \coloneqq \frac{\log m}{R^2} \|z^{\text{x}}\|_2^2 + \sum_{i \in [m]} [z^q]_i \log[z^q]_i$$

and write its associated Bregman divergence as $V_z(z') = r(z') - r(z) - \langle \nabla r(z), z' - z \rangle$. Next, we let $\Theta := \max_{z,z' \in \mathcal{Z}} \{r(z) - r(z')\}$ and observe that since $z^{\mathrm{x}} \in \mathcal{X} \subset \mathbb{B}_R(x_0)$ and $z^{\mathrm{q}} \in \mathcal{U} \subset \Delta^m$, then $\Theta = 2 \log m$. Last, we define the function clip: $\mathcal{Z}^* \to \mathcal{Z}^*$ as follows:

$$\mathrm{clip}(\delta^{\mathrm{x}}, \delta^{\mathrm{q}}) := (\delta^{\mathrm{x}}, \Pi_{[-1,1]^m}(\delta^{\mathrm{q}})),$$

where $\Pi_{[-1,1]^m}$ denotes entry-wise clipping to $[-1, 1]$. By an argument directly analogous to [14, Proposition 1], the quintuplet $(\mathcal{Z}, \|\cdot\|., r, \Theta, \mathrm{clip})$ forms a valid local norm setup [14, Definition 1].

With this notation, the iterations (16) have the concise form

$$z_{t+1} = \underset{w \in \mathcal{Z}}{\mathrm{argmin}} \{\langle \mathrm{clip}(\eta \hat{g}(z_t)), w \rangle + V_{z_t}(w)\}, \tag{18}$$

where $\hat{g}(z) := (\hat{g}^{\mathrm{x}}, -\hat{g}^{\mathrm{q}})$, with $\hat{g}^{\mathrm{x}}$ and $\hat{g}^{\mathrm{q}}$ as defined in (17) above. It is then straight-forward to verify that $\mathbb{E}\hat{g}(z) = (\nabla_x \mathcal{L}_{\mathrm{pd}}(z), -\nabla_q \mathcal{L}_{\mathrm{pd}}(z))$ for every $z \in \mathcal{U}$, and that

$$\mathbb{E}[\|\hat{g}(z)\|_w^2] = \mathbb{E}_{i \sim z^{\mathrm{q}}, j \sim w_i} \left[ \frac{R^2}{\log m} \|\nabla \ell_j(z^{\mathrm{x}})\|^2 \right] + \mathbb{E}_{i \sim \mathrm{Unif}([m]), j \sim w_i} \left[ [z^{\mathrm{q}}]_i m^2 \ell_j^2(z^{\mathrm{x}}) \right] \leq \frac{G^2 R^2}{\log m} + m B_\ell^2$$

for every $z, w \in \mathcal{U}$. Therefore, $\hat{g}$ is an $L$-local estimator [14, Definition 3] with $L^2 = G^2 R^2 / (\log m) + m B_\ell^2$ so that $L^2 \Theta = 2G^2 R^2 + 2B_\ell^2 m \log m$. Proposition 2 now follows immediately from [14, Proposition 2]. $\qquad\square$

## A.2   AGD on the softmax: complexity bound

In this appendix we briefly develop the complexity guarantees of the "AGD on softmax" approach mentioned in the introduction. While the idea is well known, we could not find in the literature an analysis of the method for the general DRO setting (i.e., maximization over $q$ in arbitrary subsets of the simplex), so we provide it here.

We wish to minimize, over $x \in \mathcal{X}$,

$$\mathcal{L}_{\mathrm{DRO}}(x) := \max_{q \in \mathcal{U}} \sum_{i \in [N]} q_i \ell_i(x)$$

where each $\ell_i$ is convex, $G$-Lipschitz and $L$-smooth and $\mathcal{U}$ is an arbitrary closed convex subset of the simplex $\Delta^N$; note that this includes both Group DRO and $f$-divergence DRO as special cases. We define the approximation

$$\widetilde{\mathcal{L}}_{\mathrm{DRO}}(x) := \max_{q \in \mathcal{U} \subset \Delta^N} \left\{ \sum_{i \in [N]} q_i \ell_i(x) - \epsilon' q_i \log q_i \right\}$$

with $\epsilon' = \frac{\epsilon}{2 \log N}$. In addition, since $\sum_{i \in [N]} q_i \log q_i \in [-\log N, 0]$, for $q \in \Delta^N$ we have that

$$\left| \mathcal{L}_{\mathrm{DRO}}(x) - \widetilde{\mathcal{L}}_{\mathrm{DRO}}(x) \right| = \left| \max_{q \in \mathcal{U} \subset \Delta^N} \left\{ \sum_{i \in [N]} q_i \ell_i(x) \right\} - \max_{q \in \mathcal{U} \subset \Delta^N} \left\{ \sum_{i \in [N]} q_i \ell_i(x) - \epsilon' q_i \log q_i \right\} \right|$$

$$\leq \left| \sum_{i \in [N]} \epsilon' q_i \log q_i \right| \leq \epsilon/2.$$

Thus for $x$ satisfying $\widetilde{\mathcal{L}}_{\mathrm{DRO}}(x) - \min_{x \in \mathcal{X}} \widetilde{\mathcal{L}}_{\mathrm{DRO}}(x) \leq \epsilon/2$ we have that $\mathcal{L}_{\mathrm{DRO}}(x) - \min_{x_\star \in \mathcal{X}} \mathcal{L}_{\mathrm{DRO}}(x_\star) \leq \epsilon$ as well.

Next, we show that $\widetilde{\mathcal{L}}_{\mathrm{DRO}}$ is $\widetilde{O}(1/\epsilon)$-smooth when each $\ell_i$ is $O(1/\epsilon)$-smooth. For $q \in \mathcal{U}$, the function $\Psi(q) = \sum_{i \in [N]} \epsilon' q_i \log(q_i)$ is $\epsilon'$-strongly convex w.r.t to the $\|\cdot\|_1$ norm, therefore the conjugate function $\Psi^*(\cdot)$ is $\frac{1}{\epsilon'}$-smooth w.r.t the dual norm $\|\cdot\|_\infty$, such that

$$\|\nabla \Psi^*(v) - \nabla \Psi^*(v')\|_1 \leq \frac{1}{\epsilon'} \|v - v'\|_\infty \tag{19}$$

**Algorithm 1:** Stochastic accelerated proximal point method

---

**Input:** BROO $\mathcal{O}_{\lambda,\delta}(\cdot)$, $T_{\max}$, initialization $x_0 = v_0$ and $A_0 \geq 0$

**Parameters:** Approximation parameters $\{\delta_k, \beta_k, \sigma_k\}$, stopping parameters $A_{\max}$ and $K_{\max}$

**1 for** $k = 0, 1, 2, \cdots$ **do**

**2**      $\lambda_{k+1} = \lambda\text{-BISECTION}(x_k, v_k, A_k)$

**3**      $a_{k+1} = \frac{1}{2\lambda_{k+1}}\sqrt{1 + 4\lambda_{k+1}A_k}$ and $A_{k+1} = A_k + a_{k+1}$

**4**      $y_k = \frac{A_k}{A_{k+1}}x_k + \frac{a_{k+1}}{A_{k+1}}v_k$

**5**      $x_{k+1} = \mathcal{O}_{\lambda_{k+1}, \delta_{k+1}}(y_k)$

**6**      $g_{k+1} = \text{MORGRADEST}\left(\mathcal{O}_{\lambda,\delta}(\cdot), y_k, \lambda_k, \frac{\beta_{k+1}}{\lambda_{k+1}}, \frac{\sigma_k^2}{\lambda_{k+1}}\right)$

**7**      $v_{k+1} = \text{Proj}_{\mathcal{X}}\left(v_k - \frac{1}{2}a_{k+1}g_{k+1}\right)$

**8**      **if** $A_{k+1} \geq A_{\max}$ **or** $k + 1 = K_{\max}$ **then**

**9**          **return** $x_{k+1}$

---

In addition, let $q^\star(\ell(x)) = \nabla\Psi^*(\ell(x)) = \text{argmax}_{q \in \mathcal{U}}\{\ell(x) - \Psi(q)\}$ and note that $\widetilde{\mathcal{L}}_{\text{DRO}}(x) = \Psi^*(\ell(x))$. Using this for every $x, y \in \mathcal{X}$ we have

$$
\left\|\nabla\widetilde{\mathcal{L}}_{\text{DRO}}(x) - \nabla\widetilde{\mathcal{L}}_{\text{DRO}}(y)\right\| = \left\|\sum_{i \in [N]} \nabla\ell_i(x)q_i^\star(\ell(x)) - \nabla\ell_i(y)q_i^\star(\ell(y))\right\|
$$

$$
\leq \left\|\sum_{i \in [N]} \nabla\ell_i(x)[q_i^\star(\ell(x)) - q_i^\star(\ell(y))]\right\| + \left\|\sum_{i \in [N]} q_i^\star(\ell(y))[\nabla\ell_i(x) - \nabla\ell_i(y)]\right\|
$$

$$
\overset{(i)}{\leq} G\|q^\star(\ell(x)) - q^\star(\ell(y))\|_1 + L\|x - y\|
$$

$$
\overset{(ii)}{\leq} \frac{G}{\epsilon'}\|\ell(x) - \ell(y)\|_\infty + L\|x - y\|
$$

$$
\overset{(iii)}{\leq} \left(\frac{G^2}{\epsilon'} + L\right)\|x - y\|
$$

where $(i)$ follows since every $\ell_i$ is $G$ Lipschitz and $L$ smooth, in addition for every $\ell_i(x) \in \mathbb{R}$ we have that $q^\star(\ell(x)) \in \mathcal{U} \subset \Delta^N$, therefore $\sum_{i \in [N]} q_i^\star(\ell(x)) = 1$, $(ii)$ follows from the inequality in (19) and $(iii)$ follows since each $\ell_i$ is $G$-Lipschitz.

Since $\widetilde{\mathcal{L}}_{\text{DRO}}$ is $\widetilde{L} = L + \frac{G^2}{\epsilon'}$-smooth, Nesterov's accelerated gradient descent [44] method is efficient for minimizing it. This method finds $x$ such that $\widetilde{\mathcal{L}}_{\text{DRO}}(x) - \min_x \widetilde{\mathcal{L}}_{\text{DRO}}(x) \leq \epsilon/2$ with $O\left(\sqrt{\widetilde{L}R^2/\epsilon}\right) = \widetilde{O}\left(\frac{GR}{\epsilon}\right)$ iterations when $L = O(G^2/\epsilon)$. Note that in every iteration we need to compute the full gradient of $\widetilde{\mathcal{L}}_{\text{DRO}}$, which requires to evaluate each $\nabla\ell_i$ and $\ell_i$. Therefore, in the weakly smooth setting the complexity of this method is $\widetilde{O}(\frac{NGR}{\epsilon})$.

## B Proof of Proposition 1

In this section we provide the proof for Proposition 1 that follows from the analysis of Algorithm 1; this section closely follows Asi et al. [3], and we refer the readers to that paper for a more detailed exposition.

We begin with a short description of Algorithm 1. This algorithm iteratively computes a $\frac{\lambda\delta^2}{2}$-approximate minimizer of $F_\lambda(x) = F(x) + \frac{\lambda}{2}\|x - y\|^2$ within a small ball of radius $r$ around $y$. To keep the ball constraint inactive it uses a bisection procedure that outputs the regularization strength value $\lambda$, such that for the minimizer $\hat{x} = \text{argmin}_{x \in \mathcal{X}} F_\lambda(x)$ with high probability we have $\|\hat{x} - y\| \leq r$. It then computes a (nearly) unbiased gradient estimator of the Moreau envelope $M_\lambda(y) = \min_{x \in \mathcal{X}} F_\lambda(x)$ and uses a momentum-like scheme to compute the next ball center.

---

**Algorithm 2:** MORGRADEST

---

**Input:** BROO $\mathcal{O}_{\lambda,\delta}(\cdot)$, query point $y$, regularization $\lambda$, bias $\beta$, and variance $\sigma^2$.

**1** Set $T_{\max} = \frac{2G^2}{\lambda^2 \min\left\{\beta^2, \frac{1}{2}\sigma^2\right\}}$, $T_0 = \frac{14G^2 \log T_{\max}}{\sigma^2}$, and $\delta_0 = \frac{G}{\lambda\sqrt{T_0}}$

**2** $x_0 = \mathcal{O}_{\lambda,\delta_0}(y_k)$

**3** Sample $J \sim \text{Geom}\left(\frac{1}{2}\right)$

**4** **if** $2^J \leq T_{\max}$ **then**

**5** $\quad\big|\quad \delta_J = \frac{G}{\lambda\sqrt{2^J T_0}}$

**6** $\quad\big|\quad x_J, x_{J-1} = \mathcal{O}_{\lambda,\delta_J}(y), \mathcal{O}_{\lambda,\delta_{J-1}}(y)$

**7** $\quad\big|\quad \hat{x} = x_0 + 2^J(x_J - x_{J-1})$

**8** **else**

**9** $\quad\big|\quad \hat{x} = x_0$

**10** **return** $\lambda(y - \hat{x})$

---

Algorithm 1 is a variant of the accelerated proximal point method in [3, Algorithm 4], and we now describe the differences between the two. The main difference is that the bias reduction scheme of Asi et al. [3] averages $\widetilde{O}\left(\frac{G^2}{\sigma^2} + 1\right)$ calls to an estimator with accuracy $\delta'_J = O\left(\frac{G}{\lambda 2^{J/2}}\right)$ where $J \sim \text{Geom}\left(\frac{1}{2}, T_{\max}\right)$ and $G$ is the Lipschitz constant of $F$. In contrast, we use a single call to an estimator with higher accuracy $\delta_J = \widetilde{O}\left(\delta'_J / \sqrt{G^2/\sigma^2}\right)$; cf. the implementations of MORGRADEST subroutine in each algorithm. There are additional differences between the error tolerance settings of our algorithms, as described below.

The guarantees of [3, Proposition 2] require the following choice of approximation parameters:

$$\varphi_k = \frac{\lambda_k \delta_k^2}{2} = \frac{\epsilon}{60\lambda_k a_k} \,,\, \beta_k = \frac{\epsilon}{120R} \text{ and } \sigma_k^2 = \frac{\epsilon}{60a_k}$$

which implies $\max_{k \leq K_{\max}}\left\{\lambda_k a_k \varphi_k + a_k \sigma_k^2 + 2R\beta_k\right\} \leq \frac{\epsilon}{20}$ (where $\beta_k$ in our notation is $\delta_k$ in the notation of [3]). These parameters were chosen so that together with [3, Lemma 6] they give the following bound

$$\mathbb{E}\left[A_K(F(x_K) - F(x_\star)) + \frac{1}{6}\sum_{i \leq K}\lambda_i A_i \|\hat{x}_i - y_{i-1}\|^2\right] \leq A_0(F(x_0) - F(x_\star)) + \frac{\epsilon}{20}\mathbb{E}A_K + R^2,$$

(20)

which is a key component in the proof of [3, Proposition 2]. However, for improved efficiency we set different parameters:

$$\varphi_k = \frac{\lambda_k \delta_k^2}{2} = \frac{\lambda_k r^2}{900 \log^3\left(\frac{GR^2}{\epsilon r}\right)} \text{ and } \sigma_k^2 = \frac{\lambda_k^2 r^2}{900 \log^3\left(\frac{GR^2}{\epsilon r}\right)}.$$

To obtain the guarantees of [3, Proposition 2] for our implementation, in the following lemma we reprove [3, Lemma 6] with our parameters and show the same bound as in (20) (with a slightly different constant factor).

**Lemma 8** (modification of [3, Lemma 6]). *Let $F$ satisfy Assumption 1 with a minimizer $x_\star$. Let*

$$\varphi_k = \frac{\lambda_k r^2}{900 \log^3\left(\frac{GR^2}{r\epsilon}\right)} \,,\, \sigma_k^2 = \frac{\lambda_k^2 r^2}{900 \log^3\left(\frac{GR^2}{r\epsilon}\right)} \,,\, \beta_k = \frac{\epsilon}{120R} \,,\, A_0 = \frac{R}{G} \text{ and } A_{\max} = \frac{9R^2}{\epsilon}.$$

*Define $\hat{x}_k := \text{argmin}_{x \in \mathcal{X}}\left\{F(x) + \frac{\lambda}{2}\|x - y_k\|^2\right\}$ and assume that for each $k$ we have $\|\hat{x}_k - y_{k-1}\| \leq r$ and that one of the following must occur*

1. *$\lambda_k < 2\lambda_m = \frac{2\epsilon}{r^{4/3}R^{2/3}}\log^2\left(\frac{GR^2}{r\epsilon}\right)$, or*

2. *$\|\hat{x}_k - y_{k-1}\| \geq \frac{3}{4}r$.*

*Then we have that*

$$\mathbb{E}\left[A_K\left(F(x_K) - F(x_\star) - \frac{\epsilon}{20}\right) + \frac{1}{12}\sum_{i \leq K} \lambda_i A_i \|\hat{x}_i - y_{i-1}\|^2\right] \leq A_0(F(x_0) - F(x_\star)) + R^2.$$

*Proof.* Define the filtration

$$\mathcal{F}_k = \sigma(x_1, v_1, A_1, \zeta_1, \ldots, x_k, v_k, A_k, \zeta_k)$$

where $\zeta_i$ is the internal randomness of $\lambda$-BISECTION$(x_k, v_k, A_k)$ and note that $A_{k+1}, y_k, \hat{x}_{k+1}$ are deterministic when conditioned on $x_k, v_k, A_k, \zeta_k$. Following the proof of [3, Lemma 6], we define

$$M_k = A_k\left(F(x_k) - F(x_\star) - \frac{\epsilon}{20}\right) + \frac{1}{12}\sum_{i \leq k} \lambda_i A_i \|\hat{x}_i - y_{i-1}\|^2 + \|v_k - x_\star\|^2$$

and show it is a supermartingale adapted to filteration $\mathcal{F}_k$. From [3, Lemma 5] we have

$$\mathbb{E}[M_{k+1}|\mathcal{F}_k] \leq A_k(F(x_k) - F(x_\star)) + \|v_k - x_\star\|^2 - \frac{1}{6}\lambda_{k+1}A_{k+1}\|\hat{x}_{k+1} - y_k\|^2$$

$$+ \mu_{k+1} - A_{k+1}\frac{\epsilon}{c} + \frac{1}{12}\sum_{i \leq k+1} \lambda_i A_i \|\hat{x}_i - y_{i-1}\|^2 \tag{21}$$

where

$$\mu_{k+1} := \lambda_{k+1}a_{k+1}^2\varphi_{k+1} + a_{k+1}^2\sigma_{k+1}^2 + 2Ra_{k+1}\beta_{k+1}.$$

Substituting the values of $\varphi_{k+1}, \sigma_{k+1}^2, \beta_{k+1}$ into the definition of $\mu_{k+1}$ gives

$$\mu_{k+1} = \frac{2\lambda_{k+1}^2 a_{k+1}^2 r^2}{900 \log^3\left(\frac{GR^2}{r\epsilon}\right)} + a_{k+1}\frac{\epsilon}{60}.$$

Recall that $A_{k+1} = a_{k+1}^2\lambda_{k+1}$, therefore

$$\mu_{k+1} = \frac{2\lambda_{k+1}A_{k+1}r^2}{900 \log^3\left(\frac{GR^2}{r\epsilon}\right)} + a_{k+1}\frac{\epsilon}{60} = \frac{2a_{k+1}\lambda_{k+1}^{3/2}\sqrt{A_{k+1}}r^2}{900 \log^3\left(\frac{GR^2}{r\epsilon}\right)} + a_{k+1}\frac{\epsilon}{60}.$$

If $\|\hat{x}_k - y_{k-1}\| \geq \frac{3}{4}r$, we have that

$$\mu_{k+1} \leq \frac{1}{12}\lambda_{k+1}A_{k+1}\|\hat{x}_{k+1} - y_k\|^2 + a_{k+1}\frac{\epsilon}{60}.$$

Else, if $\lambda_k < 2\lambda_{\mathrm{m}} = \frac{2\epsilon}{r^{4/3}R^{2/3}}\log^2\left(\frac{GR^2}{r\epsilon}\right)$, we have

$$\mu_{k+1} < \frac{2a_{k+1}2\lambda_{\mathrm{m}}^{3/2}\sqrt{A_{k+1}}r^2}{900 \log^3\left(\frac{GR^2}{r\epsilon}\right)} + a_{k+1}\frac{\epsilon}{60} = \frac{4a_{k+1}\epsilon^{3/2}\sqrt{A_{k+1}}}{900R} + a_{k+1}\frac{\epsilon}{60}.$$

Now, note that $\lambda_{\mathrm{m}} \geq \widetilde{O}\left(\frac{\epsilon}{R^2}\right) \geq \frac{1}{A_{\max}} = \frac{\epsilon}{9R^2}$ and therefore $a_K = \sqrt{\frac{1}{\lambda_K^2} + \frac{4A_{K-1}}{\lambda_K}} \leq 1.2A_{\max}$. From the definition of $A_k$ we have that $A_{K_{\max}-1} \leq A_{\max}$, therefore for every $k \leq K$

$$A_k \leq A_K = a_K + A_{K-1} \leq 2.2A_{\max}.$$

Thus, when $\lambda_k < 2\lambda_{\mathrm{m}}$ the bound on $\mu_{k+1}$ becomes

$$\mu_{k+1} < \frac{4a_{k+1}\epsilon^{3/2}\sqrt{2.2\frac{R^2}{\epsilon}}}{900R} + a_{k+1}\frac{\epsilon}{60} \leq a_{k+1}\frac{\epsilon}{20} \leq \frac{1}{12}\lambda_{k+1}A_{k+1}\|\hat{x}_{k+1} - y_k\|^2 + a_{k+1}\frac{\epsilon}{20}$$

where the last inequality follows since $A_k \geq 0$ and $\lambda_k \geq 0$. Therefore, for every $k \leq K$ we have that

$$\mu_{k+1} \leq \frac{1}{12}\lambda_{k+1}A_{k+1}\|\hat{x}_{k+1} - y_k\|^2 + a_{k+1}\frac{\epsilon}{20}.$$

Noting that $\mathbb{E}|M_k| < \infty$ and substituting the bound on $\mu_{k+1}$ into (21) we get

$$\mathbb{E}[M_{k+1}|\mathcal{F}_k] \leq A_k\Big(F(x_k) - F(x_\star) - \frac{\epsilon}{20}\Big) + \|v_k - x_\star\|^2 + \frac{1}{12}\sum_{i \leq k}\lambda_i A_i\|\hat{x}_i - y_{i-1}\|^2 = M_k.$$

Therefore $M_k$ is a supermartingle adapted to filtration $\mathcal{F}_k$. Since $K$ is a stopping time adapted to filtration $\mathcal{F}_k$, by the optional stopping theorem for supermartingles we have

$$\mathbb{E}M_K \leq M_0 = A_0\Big(F(x_0) - F(x_\star) - \frac{\epsilon}{20}\Big) + \|v_0 - x_\star\|^2 \leq A_0(F(x_0) - F(x_\star)) + R^2.$$

$\square$

For line 2 of Algorithm 1, we use the same $\lambda$-BISECTION implementation of [15]. This implementation requires calling to a *high-probability Ball Regularized Optimization Oracle* (high-probability BROO) and we give the definition of it bellow.

**Definition 2.** *An algorithm is a probability* $1 - p$ *Ball Regularized Optimization Oracle of radius* $r$ *($r$-BROO) for function* $F : \mathcal{X} \to \mathbb{R}$ *if for query point* $\bar{x} \in \mathcal{X}$,*probability* $p$, *regularization parameter* $\lambda > 0$ *and desired accuracy* $\delta > 0$ *it returns* $\mathcal{O}_{\lambda,\delta}(\bar{x}) \in \mathcal{X}$ *that with probability at least* $1 - p$ *satisfies*

$$F(\mathcal{O}_{\lambda,\delta}(\bar{x})) + \frac{\lambda}{2}\|\mathcal{O}_{\lambda,\delta}(\bar{x}) - \bar{x}\|^2 \leq \min_{x \in \mathbb{B}_r(\bar{x}) \cap \mathcal{X}}\Big\{F(x) + \frac{\lambda}{2}\|x - \bar{x}\|^2\Big\} + \frac{\lambda}{2}\delta^2. \qquad (22)$$

In the following lemma we give the complexity guarantee for a high probability $r$-BROO.

**Lemma 9.** *Let* $\mathcal{C}_F$ *be the complexity of evaluating* $F$ *exactly, and* $\mathcal{C}_\lambda(\delta)$ *be an* $r$-BROO *implementation complexity. Then the complexity of implementing a probability* $1 - p$ $r$-BROO *of Definition 2 is*

$$\log\Big(\frac{1}{p}\Big)\Big[\mathcal{C}_\lambda\Big(\frac{\delta}{\sqrt{2}}\Big) + \mathcal{C}_F\Big].$$

*Proof.* To obtain a high-probability $r$-BROO we run $\log_2\big(\frac{1}{p}\big)$ copies of $r$-BROO with query point $\bar{x}$, regularization strength $\lambda$ and accuracy $\delta/\sqrt{2}$ and take the best output, i.e., the output with the minimal value of $F$. Applying Markov's inequality to a single run of $r$-BROO with output $x$, gives

$$\mathbb{P}\Big(F(x) + \frac{\lambda}{2}\|x - \bar{x}\|^2 - \min_{x \in \mathbb{B}_r(\bar{x}) \cap \mathcal{X}}\Big\{F(x) + \frac{\lambda}{2}\|x - \bar{x}\|^2\Big\} \geq \frac{\lambda}{2}\delta^2\Big) \leq \frac{1}{2}$$

therefore with probability at least $1 - \big(\frac{1}{2}\big)^{\log_2(\frac{1}{p})} = 1 - p$, the best output $x'$ satisfies

$$F(x') + \frac{\lambda}{2}\|x' - \bar{x}\|^2 - \min_{x \in \mathbb{B}_r(\bar{x}) \cap \mathcal{X}}\Big\{F(x) + \frac{\lambda}{2}\|x - \bar{x}\|^2\Big\} \leq \frac{\lambda}{2}\delta^2.$$

Procedure require $\log(1/p)$ BROO calls and the same number of exact function evaluation (to choose the best BROO output), resulting in the claimed complexity bound $\log\big(\frac{1}{p}\big)\big[\mathcal{C}_\lambda\big(\frac{\delta}{\sqrt{2}}\big) + \mathcal{C}_F\big]$. $\square$

To compute the gradient estimator in line 6 of Algorithm 1, we use Algorithm 2. Our implementation is slightly different than [3] and in the following lemma we show it produces an estimator with the same bias and variance guarantees of [3].

**Lemma 10.** *Let* $F : \mathcal{X} \to \mathbb{R}$ *satisfy Assumption 1 and for query point* $y \in \mathcal{X}$ *and regularization strength* $\lambda > 0$ *define* $x' = \arg\min_{x \in \mathcal{X}}\{F(x) + \frac{\lambda}{2}\|x - y\|^2\}$ *and* $g = \lambda(y - x')$. *Then, for any bias and variance parameters* $\beta, \sigma > 0$ *Algorithm 2 outputs* $\hat{g} = \lambda(y - \hat{x})$ *satisfying*

$$\|\mathbb{E}\hat{g} - g\| \leq \beta \text{ and } \mathbb{E}\|\hat{g} - \mathbb{E}\hat{g}\|^2 \leq \sigma^2.$$

*Proof.* First note that if $x = \mathcal{O}_{\lambda,\delta}(\bar{x})$ is the output of an $r$-BROO with accuracy $\delta$ and if $x' = \arg\min_{x \in \mathbb{B}_r(\bar{x})} F(x)$, from Definition 1 and strong convexity (of $F(x) + \frac{\lambda}{2}\|x - \bar{x}\|^2$) we have:

$$\frac{\lambda}{2}\mathbb{E}\|x - x'\|^2 \leq \mathbb{E}\Big[F(x) + \frac{\lambda}{2}\|x - \bar{x}\|^2\Big] - \Big[F(x') + \frac{\lambda}{2}\|x' - \bar{x}\|^2\Big] \leq \frac{\lambda\delta^2}{2}$$

giving

$$\mathbb{E}\|x - x'\|^2 \leq \delta^2. \tag{23}$$

Let $j_{\max} = \lfloor \log_2 T_{\max} \rfloor$, then from the definition of $\hat{x}$ in Algorithm 2 we have that

$$\mathbb{E}\hat{x} = \mathbb{E}x_0 + \sum_{j=1}^{j_{\max}} \mathbb{P}(J = j)2^j(\mathbb{E}x_j - \mathbb{E}x_{j-1}) = \mathbb{E}x_{j_{\max}}.$$

Therefore

$$\|\mathbb{E}\hat{g} - g\| = \lambda\|\mathbb{E}x_{j_{\max}} - x'\| \overset{(i)}{\leq} \lambda\sqrt{\mathbb{E}\|x_{j_{\max}} - x'\|^2} \overset{(ii)}{\leq} \lambda\delta_{j_{\max}} = \lambda\frac{G\sqrt{\min\{\beta^2, \frac{1}{2}\sigma^2\}}}{\lambda\sqrt{2G^2T_0}} \leq \min\left\{\beta, \frac{1}{2}\sigma\right\}$$

with $(i)$ following from Jensen inequality and $(ii)$ following from the guarantee in (23). To bound the variance of $\hat{g}$ note that

$$\mathbb{E}\|\hat{g} - \mathbb{E}\hat{g}\|^2 = \lambda^2\mathbb{E}\|\hat{x} - \mathbb{E}\hat{x}\|^2 \leq \lambda^2\mathbb{E}\|\hat{x} - x'\|^2 \leq \lambda^2\left(2\mathbb{E}\|\hat{x} - x_0\|^2 + 2\mathbb{E}\|x_0 - x'\|^2\right)$$

where the last inequality follows from $\|a + b\|^2 \leq 2\|a\|^2 + 2\|b\|^2$. The definition of $\hat{x}$ gives

$$\mathbb{E}\|\hat{x} - x_0\|^2 = \sum_{j=1}^{j_{\max}} 2^j\mathbb{E}\|x_j - x_{j-1}\|^2$$

and from the guarantee in (23) we get

$$\mathbb{E}\|x_j - x_{j-1}\|^2 \leq 2\mathbb{E}\|x_j - x'\|^2 + 2\mathbb{E}\|x_{j-1} - x'\|^2 \leq 6\delta_j^2 = \frac{6G^2}{\lambda^2T_02^j}$$

thus

$$\mathbb{E}\|\hat{x} - x_0\|^2 \leq j_{\max}\frac{6G^2}{\lambda^2T_0}.$$

In addition we have $\mathbb{E}\|x_0 - x'\|^2 \leq \delta_0^2 = \frac{G^2}{\lambda^2T_0}$ and substituting back we get

$$\mathbb{E}\|\hat{g} - \mathbb{E}\hat{g}\|^2 = \lambda^2\mathbb{E}\|\hat{x} - \mathbb{E}\hat{x}\|^2 \leq \lambda^2\left(12j_{\max}\frac{G^2}{\lambda^2T_0} + 2\frac{G^2}{\lambda^2T_0}\right) \leq 14j_{\max}\frac{G^2}{T_0} \leq \sigma^2.$$

$$\square$$

Combining the previous statements, we prove our main proposition.

**Proposition 1.** *Let $F$ satisfy Assumption 1, let $\mathcal{C}_F$ be the complexity of evaluating $F$ exactly, and let $\mathcal{C}_\lambda(\delta)$ bound the complexity of an $r$-BROO query with $\delta, \lambda$. Assume that $\mathcal{C}_\lambda(\delta)$ is non-increasing in $\lambda$ and at most polynomial in $1/\delta$. For any $\epsilon > 0$, Algorithm 1 returns $x$ such that $F(x) - \min_{x_\star \in \mathcal{X}} F(x_\star) \leq \epsilon$ with probability at least $\frac{1}{2}$. For $m_\epsilon = O\left(\log\frac{GR^2}{\epsilon r}\right)$ and $\lambda_{\mathrm{m}} = O\left(\frac{m_\epsilon^2\epsilon}{r^{4/3}R^{2/3}}\right)$, the complexity of the algorithm is*

$$O\left(\left(\frac{R}{r}\right)^{2/3}\left[\left(\sum_{j=0}^{m_\epsilon}\frac{1}{2^j}\mathcal{C}_{\lambda_{\mathrm{m}}}\left(\frac{r}{2^{j/2}m_\epsilon^2}\right)\right)m_\epsilon + (\mathcal{C}_{\lambda_{\mathrm{m}}}(r) + \mathcal{C}_F)m_\epsilon^3,\right]\right). \tag{5}$$

*Proof.* We divide the proof into a correctness argument and a complexity calculation.

**Correctness.** To prove the correctness of Proposition 1 we first need to show that the guarantees of [3, Proposition 2] still hold for our implementation that includes different parameters $\varphi_k$ and $\sigma_k^2$:

$$\varphi_k = \frac{\lambda_k r^2}{900 \log^3\left(\frac{GR^2}{\epsilon r}\right)} \quad \text{and} \quad \sigma_k^2 = \frac{\lambda_k^2 r^2}{900 \log^3\left(\frac{GR^2}{\epsilon r}\right)}$$

and different implementation of lines 5 and 6.

Following Lemma 8, the guarantees of [3, Proposition 2] are still valid with our different choice of $\varphi_k$ and $\sigma_k^2$. In addition, following Lemma 10, our implementation of line 6 is valid since it produces the same guarantees that the implementation in [3] gives. Last, if the implementation of line 5 is valid it needs to satisfy

$$\mathbb{E}\left[F(x_{k+1}) + \frac{\lambda_{k+1}}{2}\|x_{k+1} - y_k\|^2\right] - \min_{x \in \mathcal{X}}\left\{F(x) + \frac{\lambda_{k+1}}{2}\|x - y_k\|^2\right\} \le \varphi_{k+1}.$$

Note that $x_{k+1}$ in our implementation is the output of $r$-BROO with accuracy $\delta_k \le \frac{r}{\sqrt{14 \cdot 900} \log^{3/2}\left(\frac{GR^2}{\epsilon r}\right)}$, therefore by Definition 1 it satisfies

$$\mathbb{E}\left[F(x_{k+1}) + \frac{\lambda_{k+1}}{2}\|x_{k+1} - y_k\|^2\right] - \min_{x \in \mathbb{B}_r(y_k)}\left\{F(x) + \frac{\lambda_{k+1}}{2}\|x - y_k\|^2\right\} \le \frac{\lambda_{k+1}\delta_{k+1}^2}{2}$$

$$\le \frac{\lambda_{k+1} r^2}{900 \log^3\left(\frac{GR^2}{\epsilon r}\right)} = \varphi_{k+1}$$

and for valid output of $\lambda$-BISECTION we have

$$\min_{x \in \mathbb{B}_r(y_k)}\left\{F(x) + \frac{\lambda_{k+1}}{2}\|x - y_k\|^2\right\} = \min_{x \in \mathcal{X}}\left\{F(x) + \frac{\lambda_{k+1}}{2}\|x - y_k\|^2\right\}$$

implying that line 5 is valid. Now let $p_{\text{BROO}}$ be the probability that all calls to $r$-BROO result in a valid output. Following [3, Proposition 2], for $K_{\max} = O\left(\left(\frac{R}{r}\right)^{2/3} m_\epsilon\right)$ with probability at least $1 - \left(1 - \frac{2}{3}\right) - (1 - p_{\text{BROO}}) = p_{\text{BROO}} - \frac{1}{3}$ the algorithm outputs $x$ that satisfies

$$F(x) - F(\hat{x}) \le \epsilon/2.$$

Let $p$ be the probability that a single BROO implementation produce invalid output and let $K_{\text{bisect-max}}$ be the maximal number of calls to high-probability $r$-BROO within line 2. Then, $p_{\text{BROO}} \ge 1 - K_{\max} K_{\text{bisect-max}} p$ and for

$$p \le \frac{1}{6 K_{\max} K_{\text{bisect-max}}}$$

with probability at least $\frac{1}{2}$ Algorithm 1 outputs $\frac{\epsilon}{2}$-suboptimal minimizer of $F$.

**Complexity.** To bound the complexity of Algorithm 1 we first bound the complexity of line 2 and the complexity of line 6 in the $k$-th iteration of Algorithm 1. Note that, for $x_{k+1}$ in line 5 we can use $x_0$ from Algorithm 2, and therefore the complexity of line 6 already includes the complexity of line 5.

Following [15, Proposition 2], $\lambda$-BISECTION in line 2 requires $m_\epsilon = O\left(\log\left(\frac{GR^2}{\epsilon r}\right)\right)$ calls to a high-probability $r$-BROO with accuracy $\delta = \frac{r}{30}$. From Lemma 9 the complexity of a single call to a probability $1 - p$ $r$-BROO is $O\left(\log\left(\frac{1}{p}\right)[\mathcal{C}_\lambda(r) + \mathcal{C}_F]\right)$. We set $p = \frac{1}{6 K_{\max} K_{\text{bisect-max}}}$, and since $K_{\max} = O\left(\left(\frac{R}{r}\right)^{2/3} m_\epsilon\right)$ and $K_{\text{bisect-max}} = m_\epsilon$ we get $\log\left(\frac{1}{p}\right) = O\left(\log\left(\frac{R}{r} m_\epsilon^2\right)\right) \le m_\epsilon$. Therefore, the total complexity of $\lambda$-BISECTION is $O\left(m_\epsilon^2[\mathcal{C}_\lambda(r) + \mathcal{C}_F]\right)$.

For the complexity of Line 6 note that Algorithm 2 calls to an $r$-BROO with accuracy $\delta_J \ge \frac{r}{30\sqrt{142^{J/2} m_\epsilon^2}}$ where $J \sim \text{Geom}\left(\frac{1}{2}, j_{\max}\right)$ and $j_{\max} \le m_\epsilon$. Therefore, we can bound the complexity of Algorithm 2 by

$$O\left(\sum_{j=0}^{m_\epsilon} \frac{1}{2^j} \mathcal{C}_{\lambda_m}\left(\frac{r}{2^{j/2} m_\epsilon^2}\right)\right).$$

The complexity of the entire algorithm is at most $K_{\max}$ times the complexity of a single iteration. Using $K_{\max} = O\left(\left(\frac{R}{r}\right)^{2/3} m_\epsilon\right)$, the total complexity becomes

$$O\left(\left(\frac{R}{r}\right)^{2/3}\left(m_\epsilon \sum_{j=0}^{m_\epsilon} \frac{1}{2^j}\mathcal{C}_{\lambda_m}\left(\frac{r}{2^{j/2}m_\epsilon^2}\right) + m_\epsilon^3[\mathcal{C}_\lambda(r) + \mathcal{C}_F]\right)\right).$$

$\square$

## C  Group DRO

In this section we provide the proofs for the results of Section 3. In Appendix C.1 we first prove that the group-softmax is a uniform approximation of $\mathcal{L}_{\text{g-DRO}}$, then, through extension of [15], we show that we can approximate $\mathcal{L}_{\text{g-DRO}}$ using the group-exponentiated softmax instead. Next, in Appendix C.2 we prove Lemma 2 and bound the moments of the MLMC and gradient estimators. We then prove the complexity guarantees of Theorem 1 in Appendix C.3. Last, in Appendix C.4 under the mean-square smoothness assumption, we provide the properties of the gradient estimator in (9) and in Appendix C.5 we prove the complexity guarantees of Theorem 2.

### C.1  Exponentiated group-softmax

Recall the definition of the (regularized) group-softmax

$$\mathcal{L}_{\text{smax},\epsilon,\lambda}(x) := \epsilon' \log\left(\sum_{i\in[M]} e^{\frac{\mathcal{L}_i(x)}{\epsilon'}}\right) + \frac{\lambda}{2}\|x - \bar{x}\|^2 \ \text{ where } \ \mathcal{L}_i(x) = \sum_{j\in[N]} w_{ij}\ell_j(x)$$

with $\mathcal{L}_{\text{smax},\epsilon,0}(x) = \mathcal{L}_{\text{smax},\epsilon}(x)$. In addition, recall the definition of the (regularized) group-exponentiated softmax

$$\Gamma_{\epsilon,\lambda}(x) := \sum_{i\in[M]} \bar{p}_i\gamma_i(x) \ \text{ where } \ \gamma_i(x) = \epsilon' e^{\frac{\mathcal{L}_i(x) - \mathcal{L}_i(\bar{x}) + \frac{\lambda}{2}\|x-\bar{x}\|^2}{\epsilon'}} \ \text{ and } \ \bar{p}_i = \frac{e^{\frac{\mathcal{L}_i(\bar{x})}{\epsilon'}}}{\sum_{i\in[M]} e^{\frac{\mathcal{L}_i(\bar{x})}{\epsilon'}}}.$$

**Lemma 11.** *Let $\mathcal{L}_{\text{smax},\epsilon}$ be the group-softmax defined in eq. (6) and $\mathcal{L}_{\text{g-DRO}}$ be the Group DRO objective defined in (2). Let $\epsilon > 0$ and $\epsilon' = \epsilon/(2\log M) > 0$. Then for all $x \in \mathcal{X}$ we have that*

$$|\mathcal{L}_{\text{g-DRO}}(x) - \mathcal{L}_{\text{smax},\epsilon}(x)| \leq \epsilon/2$$

*Proof.* First note that

$$\mathcal{L}_{\text{g-DRO}}(x) := \max_{i\in[M]} \sum_{j=1}^{N} w_{ij}\ell_j(x) = \max_{q\in\Delta^M} \sum_{i\in[M]} q_i\mathcal{L}_i(x)$$

and

$$\mathcal{L}_{\text{smax},\epsilon}(x) = \epsilon' \log\left(\sum_{i\in[M]} e^{\frac{\mathcal{L}_i(x)}{\epsilon'}}\right) = \max_{q\in\Delta^M}\left\{\sum_{i\in[M]} q_i\mathcal{L}_i(x) - \epsilon' q_i\log q_i\right\}.$$

In addition, for $q \in \Delta^M$ we have that $\sum_{i\in[M]} q_i\log qi \in [-\log M, 0]$. Combining these facts gives

$$|\mathcal{L}_{\text{smax},\epsilon}(x) - \mathcal{L}_{\text{g-DRO}}(x)| = \left|\max_{q\in\Delta^M}\left\{\sum_{i\in[M]} q_i\mathcal{L}_i(x) - \epsilon' q_i\log q_i\right\} - \max_{q\in\Delta^M} \sum_{i\in[M]} q_i\mathcal{L}_i(x)\right|$$

$$\leq \left|\epsilon' \sum_{i\in[M]} q_i\log q_i\right| \leq \epsilon'\log M = \epsilon/2$$

$\square$

**Lemma 1.** *Let each $\ell_i$ satisfy Assumption 1, and consider the restriction of $\mathcal{L}_{\mathrm{smax},\epsilon,\lambda}$ (6) and $\Gamma_{\epsilon,\lambda}$ (7) to $\mathbb{B}_r(\bar{x})$. Then the functions have the same minimizer $x_\star \in \mathbb{B}_r(\bar{x})$ and, if $\lambda \leq O(G/r)$ and $r \leq O(\epsilon'/G)$, then (a) $\Gamma_{\epsilon,\lambda}$ is $\Omega(\lambda)$-strongly convex, (b) each $\gamma_i$ is $O(G)$-Lipschitz and (c) for every $x \in \mathbb{B}_r(\bar{x})$ we have $\mathcal{L}_{\mathrm{smax},\epsilon,\lambda}(x) - \mathcal{L}_{\mathrm{smax},\epsilon,\lambda}(x_\star) \leq O(\Gamma_{\epsilon,\lambda}(x) - \Gamma_{\epsilon,\lambda}(x_\star))$.*

*Proof.* This lemma is a simple extension of [15, Lemma 1], that considers the exponentiated-softmax:

$$\sum_{i \in M} \epsilon' \frac{e^{l_i(\bar{x})/\epsilon'}}{\sum_{j \in M} e^{f_j(\bar{x})/\epsilon'}} e^{\frac{l_i(x) - l_i(\bar{x}) + \lambda \|x - \bar{x}\|}{\epsilon'}},$$

for some $l_1, \ldots, l_N$. The only assumption that [15] have on $l_i$ (for the guarantees we state in Lemma 1) is that each $l_i$ is $G$-Lipschitz. Note that each $\mathcal{L}_i$ is $G$-Lipschitz since it is a weighted average of $G$-Lipschitz functions. Therefore, we can replace $l_i$ in [15, Lemma 1] with the group average $\mathcal{L}_i$ and obtain Lemma 1. □

### C.2  MLMC estimator moment bounds

To make the MLMC estimator suitable for both Epoch-SGD and variance reduction methods, we rewrite its definition using more general notation. Specifically, for every $x, x' \in \mathcal{X}$ and $S_1^n \in [N]^n$, let

$$\widehat{\gamma}(x, x'; S_1^n) := \epsilon' e^{\frac{1}{n} \sum_{j=1}^n \frac{\ell_{S_j}(x) - \ell_{S_j}(x') + \frac{\lambda}{2}\|x - \bar{x}\|^2}{\epsilon'}} \tag{24}$$

so that $\widehat{\gamma}(x, \bar{x}; S_1^n) = \widehat{\gamma}(x; S_1^n)$. The MLMC estimator is

Draw $J \sim \mathrm{Geom}\left(1 - \frac{1}{\sqrt{8}}\right)$, $S_1, \ldots, S_n \overset{\mathrm{iid}}{\sim} w_i$ and let $\widehat{\mathcal{M}}[\gamma_i(x)] := \widehat{\gamma}(x, \bar{x}; S_1) + \frac{\widehat{\mathcal{D}}_{2^J}}{p_J}$,

where $p_j := \mathbb{P}(J = j) = \left(1/\sqrt{8}\right)^j \left(1 - \frac{1}{\sqrt{8}}\right)$ and, for $n \in 2\mathbb{N}$ we define

$$\widehat{\mathcal{D}}_n := \widehat{\gamma}(x, x'; S_1^n) - \frac{\widehat{\gamma}\left(x, x'; S_1^{\frac{n}{2}}\right) + \widehat{\gamma}\left(x, x'; S_{\frac{n}{2}+1}^n\right)}{2}. \tag{25}$$

**Lemma 12.** *Let each $\ell_i$ satisfy Assumption 1, and let $r \leq \frac{\epsilon'}{G}$, $\lambda \leq \frac{G}{r}$, $\|x - x'\| \leq 2r$ and $\|x - \bar{x}\| \leq r$. For $\widehat{\mathcal{D}}_n$ defined in (25) we have $\mathbb{E}\left|\widehat{\mathcal{D}}_n\right|^2 \leq O\left(\frac{G^4 \|x - x'\|^4}{n^2 \epsilon'^2}\right)$.*

*Proof.* For abbreviation let $M = \frac{1}{n}\sum_{j \in [n]} \frac{\hat{\ell}_{S_j}(x) + \frac{\lambda}{2}\|x - \bar{x}\|^2}{\epsilon'}$ and $\delta = \frac{1}{n}\sum_{j \in [n/2]}\left(\frac{\hat{\ell}_{S_j}(x) - \hat{\ell}_{S_{j+n/2}}(x)}{\epsilon'}\right)$ where $\hat{\ell}_{S_j}(x) = \ell_{S_j}(x) - \ell_{S_j}(x')$. We have the following bound on $|M|$:

$$|M| \overset{(i)}{\leq} \frac{1}{n}\sum_{j \in [n]} \frac{G\|x - x'\| + \frac{G}{2r}\|x - \bar{x}\|^2}{\epsilon'} \overset{(ii)}{\leq} \frac{2Gr + Gr/2}{\epsilon'} \overset{(iii)}{\leq} 2.5 \tag{26}$$

with $(i)$ following since each $\ell_j$ is $G$-Lipschitz and $\lambda \leq G/r$, $(ii)$ follows since $\|x - x'\| \leq 2r$ and $\|x - \bar{x}\| \leq r$, and $(iii)$ since $r \leq \epsilon'/G$. For $|\delta|$ we have the bound,

$$|\delta| \leq \left|\frac{1}{n}\sum_{j \in [n/2]}\frac{\hat{\ell}_{S_j}(x)}{\epsilon'}\right| + \left|\frac{1}{n}\sum_{j \in [n/2]}\frac{\hat{\ell}_{S_{j+n/2}}(x)}{\epsilon'}\right| \leq \frac{G\|x - x'\|}{\epsilon'} \leq 2, \tag{27}$$

with the second inequality following since each $\ell_j$ is $G$-Lipschitz and the last inequality since $\|x - x'\| \leq 2r$ and $r \leq \epsilon'/G$. We bound $\left|\widehat{\mathcal{D}}_n\right|$ using the previous guarantees on $|M|$ and $|\delta|$:

$$\left|\widehat{\mathcal{D}}_n\right| = \epsilon'\left|e^M - \frac{e^{M+\delta} + e^{M-\delta}}{2}\right| \leq \epsilon' e^M \left(\frac{e^\delta + e^{-\delta}}{2} - 1\right) \overset{(i)}{\leq} \epsilon' e^{2.5}\left(\frac{e^\delta + e^{-\delta}}{2} - 1\right) \overset{(ii)}{\leq} 2e^{2.5}\epsilon'\delta^2$$

where $(i)$ follows from (26) and $(ii)$ from (27) and the inequality $e^x \leq 1 + x + 2x^2$ for all $x \leq 3$ with $x = \delta$. Therefore, we have that

$$\mathbb{E}\left|\widehat{\mathcal{D}}_n\right|^2 \leq 4e^5\epsilon'^2\mathbb{E}\left[\delta^4\right].$$

Let $Y_i = \frac{\hat{\ell}_{S_i}(x) - \hat{\ell}_{S_{i+n/2}}(x)}{n\epsilon'}$ and note that the Lipschitz property of each $\ell_j$ gives $|Y_i| \leq 2\frac{G\|x-x'\|}{n\epsilon'}$, in addition, since the samples $S_1, \ldots, S_n$ are i.i.d we have $\mathbb{E}\frac{\hat{\ell}_{S_i}(x)}{n\epsilon'} = \mathbb{E}\frac{\hat{\ell}_{S_{i+n/2}}(x)}{n\epsilon'}$ and therefore $\mathbb{E}Y_i = 0$. Thus, we can use Lemma 16 below, with $Y_i = \frac{\hat{\ell}_{S_i}(x) - \hat{\ell}_{S_{i+n/2}}(x)}{n\epsilon'}$ and $c = 2\frac{G\|x-x'\|}{n\epsilon'}$, to obtain the bound $\mathbb{E}[\delta]^4 \leq O\left(\frac{G^4\|x-x'\|^4}{n^2\epsilon'^4}\right)$. Therefore,

$$\mathbb{E}\left|\widehat{\mathcal{D}}_n\right|^2 \leq O\left(\frac{G^4\|x-x'\|^4}{n^2\epsilon'^2}\right).$$

$\square$

**Lemma 2.** *Let each $\ell_i$ satisfy Assumption 1, and let $r \leq \frac{\epsilon'}{G}$, $\lambda \leq \frac{G}{r}$ and $x \in \mathbb{B}_r(\bar{x})$. Then $\widehat{\mathcal{M}}[\gamma_i(x)]$ and $\hat{g}(x)$ are unbiased for $\gamma_i(x)$ and $\nabla\Gamma_{\epsilon,\lambda}(x)$, respectively, and have bounded second moments: $\mathbb{E}\left[\widehat{\mathcal{M}}[\gamma_i(x)]\right]^2 \leq O\left(\frac{G^4\|x-\bar{x}\|^4}{\epsilon'^2} + \epsilon'^2\right)$ and $\mathbb{E}\|\hat{g}(x)\|^2 \leq O(G^2)$. In addition, the complexity of computing $\widehat{\mathcal{M}}[\gamma_i(x)]$ and $\hat{g}(x)$ is $O(1)$.*

*Proof.* We first prove the bias and moment bounds, and then address complexity.

**Properties of the MLMC estimator.** We first show that the MLMC estimator is unbiased. For every $n \in 2\mathbb{N}$ we have that $\mathbb{E}\hat{\gamma}(x; S_1^{n/2}) = \mathbb{E}\hat{\gamma}(x; S_{n/2+1}^n)$, therefore $\mathbb{E}\widehat{\mathcal{D}}_n = \mathbb{E}\hat{\gamma}(x; S_1^n) - \mathbb{E}\hat{\gamma}(x; S_1^{n/2})$ and we get

$$\mathbb{E}\left[\widehat{\mathcal{M}}[\gamma_i(x)]\right] = \mathbb{E}\hat{\gamma}(x; S_1) + \sum_{j=1}^{\infty}\left(\mathbb{E}\hat{\gamma}(x; S_1^{2^j}) - \mathbb{E}\hat{\gamma}(x; S_1^{2^{j-1}})\right) = \mathbb{E}\hat{\gamma}(x; S_1^\infty) = \gamma_i(x). \quad (28)$$

To bound the second moment of the estimator we use the inequality $(a+b)^2 \leq 2a^2 + 2b^2$, yielding

$$\mathbb{E}\left|\widehat{\mathcal{M}}[\gamma_i(x)]\right|^2 \leq 2\mathbb{E}|\hat{\gamma}(x; S_1)|^2 + 2\sum_{j=1}^{\infty}\frac{1}{p_j}\mathbb{E}\left|\widehat{\mathcal{D}}_{2^j}\right|^2. \quad (29)$$

Lemma 12 with $x' = \bar{x}$ gives $\mathbb{E}\left|\widehat{\mathcal{D}}_n\right|^2 \leq O\left(\frac{G^4\|x-\bar{x}\|^4}{n^2\epsilon'^2}\right)$ and substituting this bound into (29) while noting that $\mathbb{E}|\hat{\gamma}(x; S_1)|^2 \leq \epsilon'^2 e^3$ gives

$$\mathbb{E}\left|\widehat{\mathcal{M}}[\gamma_i(x)]\right|^2 \leq O\left(\epsilon'^2 + \frac{G^4\|x-\bar{x}\|^4}{\epsilon'^2}\sum_{j=1}^{\infty}\left(1 - \frac{1}{\sqrt{8}}\right)\frac{2^{1.5j}}{2^{2j}}\right) = O\left(\frac{G^4\|x-\bar{x}\|^4}{\epsilon'^2} + \epsilon'^2\right). \quad (30)$$

**Properties of the gradient estimator.** We use the fact that $\widehat{\mathcal{M}}[\gamma_i(x)]$ is unbiased for $\gamma_i(x)$ (shown in eq. (28) above) to argue that gradient estimator is also unbiased:

$$\mathbb{E}[\hat{g}(x)] = \mathbb{E}\left[\frac{1}{\epsilon'}\widehat{\mathcal{M}}[\gamma_i(x)](\nabla\ell_j(x) + \lambda(x-\bar{x}))\right] = \frac{1}{\epsilon'}\sum_{i\in[M]}\sum_{j\in[N]}\bar{p}_i w_{ij}\mathbb{E}\widehat{\mathcal{M}}[\gamma_i(x)](\nabla\ell_j(x) + \lambda(x-\bar{x}))$$

$$= \frac{1}{\epsilon'}\sum_{i\in[M]}\bar{p}_i\gamma_i(x)(\nabla\mathcal{L}_i(x) + \lambda(x-\bar{x})) = \sum_{i\in[M]}\bar{p}_i\nabla\gamma_i(x) = \nabla\Gamma_{\epsilon,\lambda}(x).$$

Next we bound the second moment of the gradient estimator

$$
\mathbb{E}\|\hat{g}(x)\|^2 = \frac{1}{\epsilon'^2} \sum_{i \in [M]} \bar{p}_i \mathbb{E}\Big(\widehat{\mathcal{M}}[\gamma_i(x)]\Big)^2 \sum_{j \in [N]} w_{ij} \|\nabla \ell_j(x) + \lambda(x - \bar{x})\|^2
$$

$$
\overset{(i)}{\leq} O\left( \left( \frac{G^4 \|x - \bar{x}\|^4}{\epsilon'^4} + 1 \right) \sum_{i \in [M]} \bar{p}_i \sum_{j \in [N]} w_{ij} \|\nabla \ell_j(x) + \lambda(x - \bar{x})\|^2 \right)
$$

$$
\overset{(ii)}{\leq} O\left( \left( \frac{G^4 \|x - \bar{x}\|^4}{\epsilon'^4} + 1 \right) G^2 \right) \overset{(iii)}{\leq} O(G^2)
$$

where $(i)$ follows from (30), $(ii)$ follows since each $\ell_j$ is $G$-Lipschitz, $\lambda \leq \frac{G}{r}$ and $\|x - \bar{x}\| \leq r$ and $(iii)$ since $\frac{G^4 \|x - \bar{x}\|^4}{\epsilon'^4} \leq \frac{G^4 r^4}{\epsilon'^4} \leq 1$ .

**Complexity of the MLMC and gradient estimators.** $J \sim \text{Geom}(1 - \frac{1}{\sqrt{8}})$, therefore

$$
\mathbb{E}[2^J] = \sum_{j=1}^{\infty} \frac{1}{1 - \frac{1}{\sqrt{8}}} \left( \frac{1}{\sqrt{8}} \right)^j 2^j = O(1).
$$

Note that the estimator $\widehat{\mathcal{M}}[\gamma_i(x)] = \widehat{\gamma}(x, S_1) + \frac{1}{P_J} \widehat{\mathcal{D}}_{2^J}$ requires a single function evaluation for $\widehat{\gamma}(x, S_1)$ and $2^J$ function evaluations for the term $\widehat{\mathcal{D}}_{2^J}$. As a consequence the computation of $\widehat{\mathcal{M}}[\gamma_i(x)]$ requires only $O(1)$ function evaluations in expectation. To compute $\hat{g}(x)$ we need to compute $\widehat{\mathcal{M}}[\gamma_i(x)]$ and a single sub-gradient, hence, the complexity of computing $\hat{g}(x)$ is also $O(1)$ in expectation. $\qquad \square$

### C.3 Epoch-SGD BROO implementation

We state below the convergence rate of the Epoch-SGD algorithm.

**Lemma 13** (Theorem 5, [28]). *Let $F : \mathcal{X} \to \mathbb{R}$ be $\lambda$-strongly convex with an unbiased stochastic gradient estimator $\hat{g}$ satisfying $\mathbb{E}\|\hat{g}(x)\|^2 \leq O(G^2)$ for all $x \in \mathcal{X}$, and let $x_\star = \text{argmin}_{x \in \mathcal{X}} F(x)$. Epoch-SGD finds an approximate minimizer $x$ that satisfies*

$$
\mathbb{E} F(x) - F(x_\star) \leq O\left( \frac{G^2}{\lambda T} \right)
$$

*using $T$ stochastic gradient queries.*

Applying this lemma with $F = \Gamma_{\epsilon,\lambda}$, $x \in \mathbb{B}_{r_\epsilon}(\bar{x})$ and $T = O\left( \frac{G^2}{\lambda^2 \delta^2} \right)$ immediately gives the following guarantee on the BROO implementation complexity.

**Theorem 1.** *Let each $\ell_j$ satisfy Assumption 1, let $\epsilon, \delta, \lambda > 0$ and let $r_\epsilon = \epsilon/(2G \log M)$. For any query point $\bar{x} \in \mathbb{R}^d$, regularization strength $\lambda \leq O(G/r_\epsilon)$ and accuracy $\delta$, EpochSGD [28, Algorithm 1]) with the gradient estimator (8) outputs a valid $r_\epsilon$-BROO response and has complexity $\mathcal{C}_\lambda(\delta) = O\left( N + \frac{G^2}{\lambda^2 \delta^2} \right)$. Consequently, the complexity of finding an $\epsilon$-suboptimal minimizer of $\mathcal{L}_{\text{g-DRO}}$ (2) with probability at least $\frac{1}{2}$ is*

$$
O\left( N \left( \frac{GR}{\epsilon} \right)^{2/3} \log^{11/3} H + \left( \frac{GR}{\epsilon} \right)^2 \log^2 H \right) \quad \text{where} \quad H := M \frac{GR}{\epsilon}.
$$

*Proof.* We divide the proof into correctness and complexity arguments, addressing the BROO implementation and then the overall algorithm.

**BROO implementation: correctness.** Following Lemma 1 we have that $\Gamma_{\epsilon,\lambda}$ is $\Omega(\lambda)$-strongly convex and Lemma 2 gives $\mathbb{E}\|\hat{g}(x)\|^2 \leq O(G^2)$. Thus, we can directly apply Lemma 13 with $F = \Gamma_{\epsilon,\lambda}$, $\mathcal{X} = \mathbb{B}_{r_\epsilon}(\bar{x})$, the gradient estimator $\hat{g}(x)$ defined in (8) and $T = \frac{2c^2 G^2}{\lambda^2 \delta^2}$ for a constant $c > 0$ for which Epoch-SGD outputs $x$ that satisfies

$$
\mathbb{E} \Gamma_{\epsilon,\lambda}(x) - \Gamma_{\epsilon,\lambda}(x_\star) \leq \frac{cG^2}{\lambda T} \leq \frac{\lambda \delta^2}{2c}. \tag{31}
$$

Following Lemma 1 there is a value of $c$ such that $\mathbb{E}\mathcal{L}_{\text{smax},\epsilon}^{\lambda}(x) - \mathcal{L}_{\text{smax},\epsilon}^{\lambda}(x_\star) \leq c(\mathbb{E}\Gamma_{\epsilon,\lambda}(x) - \Gamma_{\epsilon,\lambda}(x_\star))$ and from (31) we obtain

$$\mathbb{E}\mathcal{L}_{\text{smax},\epsilon}^{\lambda}(x) - \mathcal{L}_{\text{smax},\epsilon}^{\lambda}(x_\star) \leq c(\mathbb{E}\Gamma_{\epsilon,\lambda}(x) - \Gamma_{\epsilon,\lambda}(x_\star)) \leq \frac{\lambda\delta^2}{2}.$$

Therefore, Epoch-SGD outputs a valid $r_\epsilon$-BROO response for $\mathcal{L}_{\text{smax},\epsilon}$.

**BROO implementation: complexity.** For the BROO implementation we run Epoch-SGD with the gradient estimator $\hat{g}(x)$ defined in (8) and computation budget $T = O\left(\frac{G^2}{\lambda^2\delta^2}\right)$. Therefore, we need to evaluate $O\left(\frac{G^2}{\lambda^2\delta^2}\right)$ stochastic gradient estimators with complexity $O(1)$, and our gradient estimator requires additional $N$ functions evaluations for precomputing the sampling probabilities $\{\bar{p}_i\}$. Thus, the total complexity of the BROO implementation is

$$O\left(\frac{G^2}{\lambda^2\delta^2} + N\right). \tag{32}$$

**Minimizing $\mathcal{L}_{\text{g-DRO}}$: correctness.** For any $q \in \Delta^M$ note that $\mathcal{L}_q(x) := \sum_{i\in[M]} q_i\mathcal{L}_i(x) - \epsilon'q_i\log q_i$ is $G$-Lipschitz, since $\mathcal{L}_i$ is $G$-Lipschitz for all $i \in [M]$ and therefore for all $x \in \mathcal{X}$ we have $\|\nabla\mathcal{L}_q(x)\| = \left\|\sum_{i\in[M]} q_i\nabla\mathcal{L}_i(x)\right\| \leq G$. Maximum operations preserve the Lipschitz continuity and therefore $\mathcal{L}_{\text{smax},\epsilon}(x) = \max_{q\in\Delta^M}\mathcal{L}_q(x)$ is also $G$-Lipschitz. Thus, we can use Proposition 1 with $F = \mathcal{L}_{\text{smax},\epsilon}$ and obtain that the output $\bar{x}$ of Algorithm 1 with probability at least $\frac{1}{2}$ will satisfy $\mathcal{L}_{\text{smax},\epsilon}(\bar{x}) - \min_{x_\star\in\mathcal{X}}\mathcal{L}_{\text{smax},\epsilon}(x_\star) \leq \epsilon/2$. In addition, from Lemma 11 for every $x \in \mathcal{X}$ we have that $|\mathcal{L}_{\text{g-DRO}}(x) - \mathcal{L}_{\text{smax},\epsilon}(x)| \leq \epsilon/2$. Therefore, with probability at least $\frac{1}{2}$

$$\mathcal{L}_{\text{g-DRO}}(\bar{x}) - \min_{x_\star\in\mathcal{X}}\mathcal{L}_{\text{g-DRO}}(x_\star) \leq \mathcal{L}_{\text{smax},\epsilon}(\bar{x}) - \min_{x_\star\in\mathcal{X}}\mathcal{L}_{\text{smax},\epsilon}(x_\star) + \epsilon/2 \leq \epsilon.$$

**Minimizing $\mathcal{L}_{\text{g-DRO}}$: complexity.** The complexity of finding an $\epsilon/2$-suboptimal solution for $\mathcal{L}_{\text{smax},\epsilon}$ (and therefore an $\epsilon$-suboptimal solution for $\mathcal{L}_{\text{g-DRO}}$) is bounded by Proposition 1 as:

$$O\left(\left(\frac{R}{r_\epsilon}\right)^{2/3}\left[\left(\sum_{j=0}^{m_\epsilon}\frac{1}{2^j}\mathcal{C}_{\lambda_{\text{m}}}\left(\frac{r_\epsilon}{2^{j/2}m_\epsilon^2}\right)\right)m_\epsilon + (\mathcal{C}_{\lambda_{\text{m}}}(r_\epsilon) + N)m_\epsilon^3\right]\right)$$

where $m_\epsilon = O\left(\log\left(\frac{GR^2}{\epsilon r_\epsilon}\right)\right) = O\left(\log\left(\frac{GR}{\epsilon}\log M\right)\right)$. To obtain the total complexity we evaluate the complexity of running $r_\epsilon$-BROO with accuracy $\delta_j = \frac{r}{2^{j/2}m_\epsilon^2}$ (for the MLMC implementation), and accuracy $\delta_{\text{Bisection}} = \frac{r_\epsilon}{30}$ (for the bisection procedure). Using (32) and noting that $\lambda_{\text{m}} = \frac{\epsilon}{r_\epsilon^{4/3}R^{2/3}}m_\epsilon^2$ we get the following BROO complexities:

1. $\mathcal{C}_{\lambda_{\text{m}}}\left(\frac{r_\epsilon}{m_\epsilon^2 2^{j/2}}\right) = O\left(\frac{G^2 2^j m_\epsilon^4}{\lambda_{\text{m}}^2 r_\epsilon^2} + N\right) = O\left(\frac{\left(\frac{GR}{\epsilon}\right)^{4/3}}{(\log M)^{2/3}}2^j + N\right)$

2. $\mathcal{C}_{\lambda_{\text{m}}}\left(\frac{r_\epsilon}{30}\right) = O\left(\frac{G^2}{\lambda_{\text{m}}^2 r_\epsilon^2} + N\right) = O\left(\frac{\left(\frac{GR}{\epsilon}\right)^{4/3}}{m_\epsilon^4(\log M)^{2/3}} + N\right).$

Therefore

$$O\left(m_\epsilon\sum_{j=0}^{m_\epsilon}\frac{1}{2^j}\mathcal{C}_{\lambda_{\text{m}}}\left(\frac{r_\epsilon}{2^{j/2}m_\epsilon^2}\right)\right) = O\left(m_\epsilon\sum_{j=0}^{m_\epsilon}\frac{1}{2^j}\left(\frac{\left(\frac{GR}{\epsilon}\right)^{4/3}}{(\log M)^{2/3}}2^j + N\right)\right) \leq O\left(m_\epsilon^2\left(\left(\frac{GR}{\epsilon}\right)^{4/3} + N\right)\right)$$

and

$$O\left(m_\epsilon^3(\mathcal{C}_{\lambda_k}(r_\epsilon/30) + N)\right) \leq O\left(\left(\frac{GR}{\epsilon}\right)^{4/3} + Nm_\epsilon^3\right).$$

Substituting the bounds into Proposition 1 with $m_\epsilon = \log\left(\frac{GR}{\epsilon}M\right)$ and $r_\epsilon = \frac{\epsilon}{2G\log M}$, the total complexity is

$$O\left(\left(\frac{R}{r_\epsilon}\right)^{2/3}\left[Nm_\epsilon^3 + m_\epsilon^2\left(\frac{GR}{\epsilon}\right)^{4/3}\right]\right) \leq O\left(\left(\frac{GR}{\epsilon}\right)^{2/3}N\log^{11/3}\left(\frac{GR}{\epsilon}M\right) + \left(\frac{GR}{\epsilon}\right)^2\log^2\left(\frac{GR}{\epsilon}M\right)\right).$$

$\square$

### C.4 SVRG-like estimator properties

We first give a definition of $\Gamma_{\epsilon,\lambda}$ that is more conducive to formulating variance reduction methods:

$$\Gamma_{\epsilon,\lambda}(x) := \sum_{i\in[M]} c_{x',\bar{x}}\, p_i(x')\gamma_i(x,x'),$$

where $\gamma_i(x,x') := \epsilon' e^{\frac{\mathcal{L}_i(x) - \mathcal{L}_i(x') + \frac{\lambda}{2}\|x - \bar{x}\|^2}{\epsilon'}}$, $c_{x',\bar{x}} := \left(\frac{\sum_{j\in[M]} e^{\frac{\mathcal{L}_j(x')}{\epsilon'}}}{\sum_{j\in[M]} e^{\frac{\mathcal{L}_j(\bar{x})}{\epsilon'}}}\right)$ and $p_i(x') :=$

$\frac{e^{\frac{\mathcal{L}_i(x')}{\epsilon'}}}{\sum_{j\in[M]} e^{\frac{\mathcal{L}_j(x')}{\epsilon'}}}$. Therefore, the MLMC estimator of $\gamma_i(x,x')$ is

Draw $J \sim \text{Geom}\left(1 - \frac{1}{\sqrt{8}}\right)$, $S_1,\dots,S_n \overset{\text{iid}}{\sim} w_i$ and let $\widehat{\mathcal{M}}[\gamma_i(x,x')] := \widehat{\gamma}(x,x';S_1) + \frac{\widehat{\mathcal{D}}_{2^J}}{p_J}$

with $\widehat{\mathcal{D}}_n$ defined in (25) and $\widehat{\gamma}(x,x';S_1^n)$ defined in (24).

**Lemma 14.** *The SVRG-like estimator* (9) *is unbiased*

$$\mathbb{E}[\hat{g}_{x'}(x)] = \nabla\Gamma_{\epsilon,\lambda}(x)$$

*Proof.*

$$\mathbb{E}[\hat{g}_{\bar{x}}(x)] = \nabla\Gamma_{\epsilon,\lambda}(x') + \sum_{i\in[M]}\sum_{j\in[N]} p_i(x')w_{ij}\frac{1}{\epsilon'}\left[\mathbb{E}\left(\widehat{\mathcal{M}}[\gamma_i(x,x')]\right)\nabla\ell_j^\lambda(x) - \gamma_i(x',x')\nabla\ell_j^\lambda(x')\right]$$

$$\overset{(i)}{=} \nabla\Gamma_{\epsilon,\lambda}(x') + \sum_{i\in[M]}\sum_{j\in[N]} p_i(x')w_{ij}\frac{1}{\epsilon'}\left[\gamma_i(x,x')\nabla\ell_j^\lambda(x) - \gamma_i(x',x')\nabla\ell_j^\lambda(x')\right]$$

$$= \nabla\Gamma_{\epsilon,\lambda}(x') + \sum_{i\in[M]} p_i(x')[\nabla\gamma_i(x,x') - \nabla\gamma_i(x',x')]$$

$$= \nabla\Gamma_{\epsilon,\lambda}(x)$$

with $(i)$ following from the unbiased property of the MLMC estimator stated in Lemma 2 (that still holds for $\widehat{\mathcal{M}}[\gamma_i(x,x')]$). $\square$

**Lemma 3.** *Let each $\ell_j$ satisfy Assumptions 1 and 2. For any $\lambda \leq \frac{G}{r}$, $r = \frac{\epsilon'}{G}$ and $x,x' \in \mathbb{B}_r(\bar{x})$, the variance of $\hat{g}_{x'}(x)$ is bounded by $\text{Var}(\hat{g}_{x'}(x)) \leq O\left(\left(L + \lambda + \frac{G^2}{\epsilon'}\right)^2\|x - x'\|^2\right)$.*

*Proof.*

$$\text{Var}(\hat{g}_{x'}(x)) = \mathbb{E}\|\hat{g}_{x'}(x) - \mathbb{E}\hat{g}_{x'}(x)\|^2$$

$$= \mathbb{E}\left\|\nabla\Gamma_{\epsilon,\lambda}(x') + \frac{c_{x',\bar{x}}}{\epsilon'}\left[\widehat{\mathcal{M}}[\gamma_i(x,x')]\nabla\ell_j^\lambda(x) - \gamma_i(x',x')\nabla\ell_j^\lambda(x')\right] - \nabla\Gamma_{\epsilon,\lambda}(x)\right\|^2$$

$$\overset{(i)}{\leq} \frac{c_{x',\bar{x}}^2}{\epsilon'^2}\mathbb{E}\left\|\widehat{\mathcal{M}}[\gamma_i(x,x')]\nabla\ell_j^\lambda(x) - \gamma_i(x',x')\nabla\ell_j^\lambda(x')\right\|^2$$

$$= \frac{c_{x',\bar{x}}^2}{\epsilon'^2}\mathbb{E}\left\|\left(\widehat{\mathcal{M}}[\gamma_i(x,x')] - \gamma_i(x',x')\right)\nabla\ell_j^\lambda(x) + \gamma_i(x',x')\left(\nabla\ell_j^\lambda(x) - \nabla\ell_j^\lambda(x')\right)\right\|^2$$

$$\overset{(ii)}{\leq} \frac{c_{x',\bar{x}}^2}{\epsilon'^2}\left[2\mathbb{E}\left\|\left(\widehat{\mathcal{M}}[\gamma_i(x,x')] - \gamma_i(x',x')\right)\nabla\ell_j^\lambda(x)\right\|^2 + 2\mathbb{E}\|\gamma_i(x',x')\left(\nabla\ell_j^\lambda(x) - \nabla\ell_j^\lambda(x')\right)\|^2\right]$$

$$(33)$$

where $(i)$ follows from the inequality $\mathbb{E}[X - \mathbb{E}X]^2 = \mathbb{E}[X^2] - [\mathbb{E}X]^2 \leq \mathbb{E}[X^2]$ and $(ii)$ from the inequality $(a+b)^2 \leq 2a^2 + 2b^2$. Next we bound separately each of the expectation terms. Note that the ball constraint $x \in \mathbb{B}_r(\bar{x})$ with $r = \frac{\epsilon'}{G}$ and $\lambda \leq \frac{G}{r}$ gives:

$$\gamma_i(x', x') = \epsilon' e^{\frac{\lambda}{2\epsilon'}\|x - \bar{x}\|^2} \leq e^{\frac{Gr}{2\epsilon'}} = O(\epsilon')$$

therefore

$$\mathbb{E}\|\gamma_i(x', x')[\nabla \ell_j(x) - \nabla \ell_j(x') + \lambda(x - x')]\|^2 \overset{(i)}{\leq} O(\epsilon'^2)\Big(2\mathbb{E}\|\nabla \ell_j(x) - \nabla \ell_j(x')\|^2 + 2\|\lambda(x - x')\|^2\Big)$$

$$\overset{(ii)}{\leq} O\Big(\epsilon'^2(\lambda^2 + L^2)\|x - x'\|^2\Big)$$

with $(i)$ following from the inequality $(a+b)^2 \leq 2a^2 + 2b^2$ and $(ii)$ from Assumption 2. For the second expectation term we use the fact that each $\ell_j$ is $G$-Lipschitz, $\lambda \leq \frac{G}{r}$ and $\|x - \bar{x}\| \leq r$ and thus $\|\nabla \ell_j(x) + \lambda(x - \bar{x})\| \leq \|\nabla \ell_j(x)\| + \|\lambda(x - \bar{x})\| \leq 2G$. Therefore,

$$\mathbb{E}\left\|\Big(\widehat{\mathcal{M}}[\gamma_i(x, x')] - \gamma_i(x', x')\Big)(\nabla \ell_j(x) + \lambda(x - \bar{x}))\right\|^2 \leq O\Big(G^2\Big(\mathbb{E}\Big|\widehat{\mathcal{M}}[\gamma_i(x, x')] - \gamma_i(x', x')\Big|^2\Big)\Big).$$

From the definition of $\widehat{\mathcal{M}}[\gamma_i(x, x')]$ we get:

$$\mathbb{E}\Big|\widehat{\mathcal{M}}[\gamma_i(x, x')] - \gamma_i(x', x')\Big|^2 \leq 2\mathbb{E}|\widehat{\gamma}(x, x'; S_1) - \gamma_i(x', x')|^2 + 2\sum_{j=1}^{\infty}\Big(1 - \frac{1}{\sqrt{8}}\Big)2^{1.5j}\mathbb{E}\Big|\widehat{\mathcal{D}}_{2^j}\Big|^2$$

$$\overset{(i)}{\leq} 2\mathbb{E}|\widehat{\gamma}(x, x'; S_1) - \gamma_i(x', x')|^2 + O\left(\frac{G^4\|x - x'\|^4}{\epsilon'^2}\right)$$

$$= O\left(\epsilon'^2 e^{\frac{\lambda\|x - \bar{x}\|^2}{\epsilon'}}\mathbb{E}\left(e^{\frac{\ell_{S_1}(x) - \ell_{S_1}(x')}{\epsilon'}} - 1\right)^2 + \left(\frac{G^4\|x - x'\|^4}{\epsilon'^2}\right)\right)$$

$$\overset{(ii)}{\leq} O\left(\epsilon'^2 e^{\frac{\lambda\|x - \bar{x}\|^2}{\epsilon'}}\mathbb{E}\left(\frac{\ell_{S_1}(x) - \ell_{S_1}(x')}{\epsilon'}\right)^4 + \left(\frac{G^4\|x - x'\|^4}{\epsilon'^2}\right)\right)$$

$$\overset{(iii)}{\leq} O\left(\frac{G^4\|x - x'\|^4}{\epsilon'^2}\right)$$

$$\overset{(iv)}{\leq} O\Big(G^2\|x - x'\|^2\Big)$$

with $(i)$ following from Lemma 12, $(ii)$ follows from the inequality $e^x - 1 \leq x + 2x^2 = O(x^2)$ for $x \leq 3$ with $x = \frac{\ell_{S_1}(x) - \ell_{S_1}(x')}{\epsilon'} \leq 2$, $(iii)$ follows since each $\ell_j$ is $G$-Lipschitz and since $e^{\frac{\lambda\|x - \bar{x}\|^2}{\epsilon'}} = O(1)$ and $(iv)$ since $\frac{G^2\|x - x'\|^2}{\epsilon'^2} \leq \frac{G^2 4r^2}{\epsilon'^2} = 4$. Therefore,

$$\mathbb{E}\left\|\Big(\widehat{\mathcal{M}}[\gamma_i(x, x')] - \gamma_i(x', x')\Big)(\nabla \ell_j(x) + \lambda(x - \bar{x}))\right\|^2 \leq O\Big(G^4\|x - x'\|^2\Big).$$

Finally we bound $c_{x', \bar{x}}$ using the ball constraint $x' \in \mathbb{B}_r(\bar{x})$ and the fact that each $\mathcal{L}_i$ is $G$-Lipschitz, therefore

$$c_{x', \bar{x}} = \frac{\sum_{j \in [M]} e^{\mathcal{L}_j(x')/\epsilon'}}{\sum_{j \in [M]} e^{\mathcal{L}_j(\bar{x})/\epsilon'}} = \frac{\sum_{j \in [M]} e^{\frac{\mathcal{L}_j(x') - \mathcal{L}_j(\bar{x})}{\epsilon'}} e^{\frac{\mathcal{L}_j(\bar{x})}{\epsilon'}}}{\sum_{j \in [M]} e^{\frac{\mathcal{L}_j(\bar{x})}{\epsilon'}}} \leq e.$$

Substituting back the bounds on each expectaion term and the bound on $c_{x', \bar{x}}$ into (33) we get

$$\mathrm{Var}(\hat{g}_{\bar{x}}(x)) \leq O\left(L^2 + \lambda^2 + \frac{G^4}{\epsilon'^2}\right)\|x - x'\|^2 \leq O\left(\left(L + \lambda + \frac{G^2}{\epsilon'}\right)^2\|x - x'\|^2\right).$$

$\square$

## C.5 Complexity of the reduced-variance BROO implementation

We first state the complexity bounds of KatyushaX$^s$ [2]

**Lemma 15** ([2, Theorems 1 and 4.3]). *Let F be a $\lambda$-strongly convex function with minimizer $x_\star$ and let $\hat{g}_{x'}(x)$ be a stochastic gradient estimator satisfying the properties*

1. $\mathbb{E}[\hat{g}_{x'}(x)] = \mathbb{E}\nabla F(x)$

2. $\mathbb{E}[\hat{g}_{x'}(x) - \nabla F(x)]^2 \leq \widetilde{L}^2 \|x - \bar{x}\|^2$

3. $\hat{g}_{x'}(\cdot)$ *has evaluation complexity $O(1)$ and preprocessing complexity $O(N)$,*

*then KatyushaX$^s$ with the stochastic gradient estimator $\hat{g}_{x'}$ finds a point $x$ satisfying $\mathbb{E}[F(x) - F(x_\star)] \leq \epsilon$ with complexity*

$$O\left(\left(N + \frac{N^{3/4}\sqrt{\widetilde{L}}}{\sqrt{\lambda}}\right)\log\left(\frac{F(x_0) - F(x_\star)}{\epsilon}\right)\right).$$

Applying Lemma 15 with $\widetilde{L} = O\left(L + \frac{G^2}{\epsilon}\right)$ and accuracy $\frac{\lambda\delta^2}{2}$ gives the following result.

**Theorem 2.** *Let each $\ell_j$ satisfy Assumptions 1 and 2. Let $\epsilon > 0$, $\epsilon' = \epsilon/(2\log M)$ and $r_\epsilon = \epsilon'/G$. For any query point $\bar{x} \in \mathbb{R}^d$, regularization strength $\lambda \leq O(G/r_\epsilon)$ and accuracy $\delta$, KatyushaX$^s$ [2, Algorithm 2] with the gradient estimator (9) outputs a valid $r_\epsilon$-BROO response and has complexity $\mathcal{C}_\lambda(\delta) = O\left(\left(N + \frac{N^{3/4}(G+\sqrt{\epsilon'L})}{\sqrt{\lambda\epsilon'}}\right)\log\left(\frac{Gr_\epsilon}{\lambda\delta^2}\right)\right)$. Consequently, the complexity of finding an $\epsilon$-suboptimal minimizer of $\mathcal{L}_{\text{g-DRO}}$ (2) with probability at least $\frac{1}{2}$ is*

$$O\left(N\left(\frac{GR}{\epsilon}\right)^{2/3}\log^{14/3}H + N^{3/4}\left(\frac{GR}{\epsilon} + \sqrt{\frac{LR^2}{\epsilon}}\right)\log^{7/2}H\right) \quad \text{where} \quad H := M\frac{GR}{\epsilon}.$$

*Proof.* The proof is structured similarly to the proof of Theorem 1.

**BROO implementation: correctness.** From Lemma 1 we have that $\Gamma_{\epsilon,\lambda}$ is $\Omega(\lambda)$-strongly convex, in addition, Lemma 3 and Lemma 14 show that the stochastic gradient estimator defined in (9) is unbiased with $\text{Var}(\hat{g}_{x'}(x)) \leq \widetilde{L}^2\|x - x'\|^2$. Thus, we can directly apply Lemma 15 with $F = \Gamma_{\epsilon,\lambda}$ and accuracy $\frac{\lambda\delta^2}{2}$ and obtain a valid BROO response.

**BROO implementation: complexity.** We use KatyushaX$^s$ [2] for the BROO implementation. Applying Lemma 15 with $F = \Gamma_{\epsilon,\lambda}$, $x_0 = \bar{x}$, $\widetilde{L} = O\left(L + \lambda + \frac{G^2}{\epsilon'}\right) \leq O\left(L + \frac{G^2}{\epsilon'}\right)$ and accuracy $\frac{\lambda\delta^2}{2}$ the complexity of our implementation is

$$O\left(\left(N + N^{3/4}\frac{\sqrt{L\epsilon'} + G}{\sqrt{\epsilon'}}\right)\log\left(\frac{\Gamma_{\epsilon,\lambda}(\bar{x}) - \min_{x_\star\in\mathbb{B}_{r_\epsilon}(\bar{x})}\Gamma_{\epsilon,\lambda}(x_\star)}{\lambda\delta^2}\right)\right) \quad (34)$$

and note that $\Gamma_{\epsilon,\lambda}(\bar{x}) - \min_{x_\star\in\mathbb{B}_{r_\epsilon}(\bar{x})}\Gamma_{\epsilon,\lambda}(x_\star) \leq Gr_\epsilon$, since (from Lemma 1) $\Gamma_{\epsilon,\lambda}$ is $O(G)$-Lipschitz.

**Minimizing $\mathcal{L}_{\text{g-DRO}}$: correctness.** Similarly to the proof of Theorem 1, combining the guarantees of Proposition 1 and Lemma 11, with probability at least $\frac{1}{2}$ the output $\bar{x}$ of Algorithm 1 satisfies $\mathcal{L}_{\text{g-DRO}}(\bar{x}) - \min_{x_\star\in\mathcal{X}}\mathcal{L}_{\text{g-DRO}}(x_\star) \leq \epsilon$.

**Minimizing $\mathcal{L}_{\text{g-DRO}}$: complexity.** The complexity of finding $\epsilon/2$-suboptimal solution for $\mathcal{L}_{\text{smax},\epsilon}$ and therefore an $\epsilon$-suboptimal solution for $\mathcal{L}_{\text{g-DRO}}$, is bounded by Proposition 1 as:

$$O\left(\left(\frac{R}{r_\epsilon}\right)^{2/3}\left[\left(\sum_{j=0}^{m_\epsilon}\frac{1}{2^j}\mathcal{C}_{\lambda_{\text{m}}}\left(\frac{r_\epsilon}{2^{j/2}m_\epsilon^2}\right)\right)m_\epsilon + (\mathcal{C}_{\lambda_{\text{m}}}(r_\epsilon) + N)m_\epsilon^3\right]\right) \quad (35)$$

where $m_\epsilon = O\left(\log\frac{GR^2}{\epsilon r_\epsilon}\right) = \log\left(\frac{GR}{\epsilon}\log M\right)$ and $\lambda_{\text{m}} = O\left(\frac{m_\epsilon^2\epsilon}{r^{4/3}R^{2/3}}\right)$. We first show the complexity of the BROO implementation with $\delta_j = \frac{r_\epsilon}{2^{j/2}m_\epsilon^2}$ (for the MLMC implementation) and with $\delta = \left(\frac{r_\epsilon}{30}\right)$ for the bisection procedure. Using (34) we get:

1. $\mathcal{C}_{\lambda_{\mathrm{m}}}\left(\frac{r_\epsilon}{2^{j/2}m_\epsilon^2}\right) = O\left(\left(N + N^{3/4}\left(\frac{G\sqrt{\log M}}{\sqrt{\epsilon}} + \sqrt{L}\right)\frac{1}{\sqrt{\lambda_{\mathrm{m}}}}\right)\log\left(\frac{\epsilon' 2^j m_\epsilon^4}{\lambda_{\mathrm{m}} r_\epsilon^2}\right)\right)$

2. $\mathcal{C}_{\lambda_{\mathrm{m}}}\left(\frac{r_\epsilon}{30}\right) = O\left(\left(N + N^{3/4}\left(\frac{G\sqrt{\log M}}{\sqrt{\epsilon}} + \sqrt{L}\right)\frac{1}{\sqrt{\lambda_k}}\right)\log\left(\frac{\epsilon'}{\lambda_{\mathrm{m}} r_\epsilon^2}\right)\right)$

From the definitions of $\lambda_{\mathrm{m}}$ and $r_\epsilon$ we have $\frac{\epsilon'}{\lambda_{\mathrm{m}} r_\epsilon^2} = O\left(\left(\frac{GR}{\epsilon}\right)^{2/3}\frac{1}{m_\epsilon^2}\right)$ and $\frac{1}{\sqrt{\lambda_{\mathrm{m}}}} = O\left(\frac{R^{1/3}r_\epsilon^{2/3}}{m_\epsilon\sqrt{\epsilon}}\right)$, therefore,

$$m_\epsilon \sum_{j=0}^{m_\epsilon} \frac{1}{2^j}\mathcal{C}_{\lambda_{\mathrm{m}}}\left(\frac{r_\epsilon}{2^{j/2}m_\epsilon^2}\right) = O\left(m_\epsilon \sum_{j=0}^{m_\epsilon}\frac{1}{2^j}\left(N + N^{3/4}\left(\frac{G\sqrt{\log M} + \sqrt{L\epsilon}}{\sqrt{\epsilon}}\right)\sqrt{\frac{R^{2/3}r_\epsilon^{4/3}}{\epsilon m_\epsilon^2}}\right)\log\left(\frac{GRm_\epsilon^2 2^j}{\epsilon}\right)\right)$$

$$\leq O\left(m_\epsilon^2\left[N + N^{3/4}\left(\frac{GR^{1/3}}{\epsilon} + \sqrt{\frac{LR^{2/3}}{\epsilon}}\right)r_\epsilon^{2/3}\right]\right).$$

Similarly, we have that

$$O\left((\mathcal{C}_{\lambda_{\mathrm{m}}}(r_\epsilon) + N)m_\epsilon^3\right) = O\left(m_\epsilon^3\left(N + N^{3/4}\left(\frac{G\sqrt{\log M} + \sqrt{L}}{\sqrt{\epsilon}}\sqrt{\frac{R^{2/3}r_\epsilon^{4/3}}{\epsilon m_\epsilon^2}}\right)\right)\log\left(\left(\frac{GR}{\epsilon}\right)\frac{1}{m_\epsilon^2}\right)\right)$$

$$\leq O\left(m_\epsilon^4 N + m_\epsilon^{3.5} N^{3/4}\left(\frac{GR^{1/3}}{\epsilon} + \sqrt{\frac{LR^{2/3}}{\epsilon}}\right)r_\epsilon^{2/3}\right).$$

Substituting the bounds into Proposition 1 with $m_\epsilon = \log\left(\frac{GR}{\epsilon}\log M\right)$ and $r_\epsilon = \frac{\epsilon}{2\log M}$ the total complexity is

$$O\left(N\left(\frac{GR}{\epsilon}\right)^{2/3}\log^{14/3}\left(\frac{GR}{\epsilon}\log M\right) + N^{3/4}\left(\frac{GR}{\epsilon} + \sqrt{\frac{LR^2}{\epsilon}}\right)\log^{7/2}\left(\frac{GR}{\epsilon}\log M\right)\right).$$

□

### C.6 Helper lemmas

**Lemma 16.** *Let $Y_1, \ldots, Y_n$ be a sequence of random i.i.d variables such that for every $i \in [n]$ and a constant $c > 0$ we have that $\mathbb{E}[Y_i] = 0$ and $|Y_i| \leq c$ with probability 1. Then*

$$\mathbb{E}\left(\sum_{i=1}^n Y_i\right)^4 \leq O\left(n^2 c^4\right).$$

*Proof.* For $i \neq j$ we have that $\mathbb{E}[Y_i Y_j] = \mathbb{E}[Y_i]\mathbb{E}[Y_j] = 0$, therefore

$$\mathbb{E}\left(\sum_{i=1}^n Y_i\right)^4 = \sum_{i=1}^n \mathbb{E}[Y_i^4] + 3\sum_{i=1}^n\sum_{j\neq i}\mathbb{E}[Y_i^2]\mathbb{E}[Y_j^2]$$

$$\leq nc^4 + 3n\left(\frac{n-1}{2}\right)c^4 = O\left(n^2 c^4\right)$$

□

## D  DRO with $f$-divergence

In this section we provide the proofs for the results in Section 4. In Appendix D.1 we provide the derivation of the dual formulation in (10). In Appendix D.2 we show how to reduce the constrained problem (3) to a regularized problem of the form (10), then in Appendix D.3 we describe the properties of $\mathcal{L}_{\psi,\epsilon}$, the approximation of (10). In Appendix D.4 we provide the proofs for our main technical contribution and give guarantees on the stability of the gradient estimators. Last, in Appendices D.5 and D.6 we give the complexity guarantees of our implementation in the non-smooth and slightly smooth cases.

### D.1 Dual formulation of DRO with $f$-divergence

Here we give the derivation of the objective in (10), also considered in prior work [e.g., 42, 39, 32]. Recall the DRO with $f$-divergence objective:

$$\mathcal{L}_{f\text{-div}}(x) := \max_{q \in \Delta^N : \sum_{i \in [N]} \frac{f(Nq_i)}{N} \leq 1} \sum_{i \in [N]} q_i \ell_i(x).$$

We first show the relation between $\mathcal{L}_{f\text{-div}}$ and its regularized form (10). Using the Lagrange multiplier $\nu$ for the constraint $\sum_{i \in [N]} \frac{f(Nq_i)}{N} \leq 1$ and strong duality we get

$$\mathcal{L}_{f\text{-div}}(x) = \min_{\nu \geq 0} \left\{ \nu + \max_{q \in \Delta^N} \sum_{i \in [N]} \left( q_i \ell_i(x) - \frac{\nu}{N} f(Nq_i) \right) \right\} = \min_{\nu \geq 0} \left\{ \nu + \mathcal{L}_{\nu \cdot f}(x) \right\},$$

where $\mathcal{L}_{\nu \cdot f}(x)$ is the regularized form of $\mathcal{L}_{f\text{-div}}$: writing $\psi(x) = \frac{\nu}{N} f(Nx)$, with slight abuse of notation we have

$$\mathcal{L}_{\nu \cdot f}(x) = \mathcal{L}_\psi(x) = \max_{q \in \Delta^N} \left\{ \sum_{i \in [N]} (q_i \ell_i(x) - \psi(q_i)) \right\}.$$

Adding a Lagrange multiplier $y$ for the constraint that $q \in \Delta^N$ and using strong duality again gives

$$\mathcal{L}_\psi(x) = \max_{q \in \mathbb{R}_+^N} \min_{y \in \mathbb{R}} \left\{ \sum_{i \in [N]} (q_i \ell_i(x) - \psi(q_i) - Gy \cdot q_i) + Gy \right\}$$

$$= \min_{y \in \mathbb{R}} \left\{ \sum_{i \in [N]} \max_{q_i \in \mathbb{R}_+} (q_i \ell_i(x) - \psi(q_i) - Gy \cdot q_i) + Gy \right\}.$$

Finally, using $\psi^*(v) := \max_{t \in \text{dom}(\psi)} \{vt - \psi t\}$ (the Fenchel conjugate of $\psi$), we have

$$\mathcal{L}_\psi(x) = \min_{y \in \mathbb{R}} \left\{ \sum_{i \in [N]} \psi^*(\ell_i(x) - Gy) + Gy \right\},$$

and note that $\psi^*(\ell_i(x) - Gy)$ is equivalent to $\nu f^* \left( \frac{\ell_i(x) - Gy}{\nu} \right)$.

### D.2 Minimizing the constrained objective using the regularized objective

In this section, we show that under the following Assumptions 4 and 5 we can reduce the constrained problem of minimizing (3) to the regularized problem of minimizing (10) by computing a polylogarithmic number of $O(\epsilon)$-accurate minimizers of (10).

**Assumption 4.** *Each loss function $\ell_i$ is bounded, i.e., $\ell_i : \mathcal{X} \to [0, B_\ell]$ for every $i \in [N]$.*

**Assumption 5.** *For any uncertainty set of the form $\mathcal{U} = \{q \in \Delta^N : D_f(q,p) \leq 1\}$, the divergence function $f$ is bounded, i.e., $f : \mathbb{R}_+ \to [0, B_f]$ for some $B_f \geq 1$.*

We note that the above assumptions are weak since the complexity of our approach only depends logarithmically on on $\frac{B_f B_\ell}{\epsilon}$.

We first cite a result on noisy one dimensional bisection, as given by a guarantees on the OneDim-Minimizer algorithm in Cohen et al. [18].

**Lemma 17** (Lemma 33, Cohen et al. [18]). *let $f : \mathbb{R} \to \mathbb{R}$ be a B-Lipschitz convex function defined on the interval $[\ell, u]$, and let $\mathcal{G} : \mathbb{R} \to \mathbb{R}$ be an oracle such that $|\mathcal{G}(y) - f(y)| \leq \widetilde{\epsilon}$ for all y. With $O\left( \log\left( \frac{B(u-\ell)}{\widetilde{\epsilon}} \right) \right)$ calls to $\mathcal{G}$, the algorithm OneDimMinimizer [18, Algorithm 8] outputs $y'$ such that*

$$f(y') - \min_y f(y) \leq 4\widetilde{\epsilon}$$

We specialize Lemma 17 to our settings and provide the complexity guarantees of minimizing (3) to an $\epsilon$-accurate solution using a noisy oracle $\mathcal{G}$.

**Proposition 3.** *Let each $\ell_i$ satisfy Assumption 4 and let $f$ satisfy Assumption 5. Define the function $h(\nu) := \min_{x \in \mathcal{X}} \mathcal{L}_{\nu \cdot f}(x) + \nu$ with $\mathcal{L}_{\nu \cdot f}$ defined in (10) and let $\mathcal{G}$ be an oracle such that $\mathcal{G}(\nu) \geq h(\nu)$ with probability 1 and $\mathcal{G}(\nu) - h(\nu) \leq \frac{\epsilon}{5}$ with probability at least $\frac{1}{2}$. Then applying $\mathrm{OneDimMinimizer}$ [18, Algorithm 8] on the interval $[0, B_\ell]$ outputs $\nu'$ that with probability at leat $\frac{99}{100}$ satisfies*

$$\mathcal{G}(\nu') - \min_{\nu \geq 0} h(\nu) = \mathcal{G}(\nu') - \min_{x \in \mathcal{X}} \mathcal{L}_{\mathrm{f-div}}(x) \leq \epsilon$$

*using $O(\log(H) \log(\log H))$ calls to $\mathcal{G}$, where $H = \frac{B_f B_\ell}{\epsilon}$.*

*Proof.* Let $\hat{h}_{q,x}(\nu) := \sum_{i \in [N]} q_i \ell_i(x) - \nu \left( \frac{1}{N} \sum_{i \in [N]} f(N q_i) - 1 \right)$ and note that for any $q$ and $x$ the function $\hat{h}_{q,x}$ is $B_f$-Lipschitz, since it is linear in $\nu$ and $\left| \frac{1}{N} \sum_{i \in [N]} f(N q_i) - 1 \right| \leq B_f$. Minimization and maximization operations preserve the Lipschitz continuity and therefore the function $h(\nu) = \min_x \mathcal{L}_{\nu \cdot f}(x) = \min_x \max_q \hat{h}_{q,x}(\nu)$ is also $B_f$-Lipschitz continuous. In addition for the $q^\star \in \Delta^N$ that maximizes $\mathcal{L}_{\nu \cdot f}(x)$ we have that

$$\sum_{i \in [N]} \left[ q_i^\star \ell_i(x) - \nu \frac{1}{N} f(N q_i^\star) \right] \geq \sum_{i \in [N]} \frac{1}{N} \ell_i(x)$$

and rearranging gives

$$\frac{1}{N} \sum_{i \in [N]} f(N q_i^\star) \leq \frac{\sum_{i \in [N]} [q_i^\star \ell_i(x) - \frac{1}{N} \ell_i(x)]}{\nu} \leq \frac{B_\ell}{\nu}.$$

Therefore, for all $\nu > B_\ell$ we have $h'(\nu) = 1 - \frac{1}{N} \sum_{i \in [N]} f(N q_i^\star) > 0$ and therefore it suffices to restrict $h(\nu)$ to $[0, B_\ell]$. Next, to turn $\mathcal{G}$ into a high-probability oracle, we call it $\log_2 \left( 100 \log \left( \frac{B_f B_\ell}{\epsilon} \right) \right)$ times and choose the smallest output. Therefore, with probability at least $1 - \left( \frac{1}{2} \right)^{\log \left( 100 \log \left( \frac{B_f B_\ell}{\epsilon} \right) \right)} = 1 - 1 / \left( 100 \log \left( \frac{B_f B_\ell}{\epsilon} \right) \right)$ the result is within $\frac{\epsilon}{5}$ of $h(\nu)$. Since $h$ is $B_f$-Lipschitz and defined on $[0, B_\ell]$ we can use Lemma 17 with $\ell = 0$, $u = B_f$, $\widetilde{\epsilon} = \epsilon/5$, $B = B_f$ and the high-probability version of $\mathcal{G}$. Therefore, using $O\left( \log \left( \frac{B_f B_\ell}{\epsilon} \right) \right)$ calls to the high probability version of $\mathcal{G}$ and applying the union bound, we obtain that with probability at least $\frac{99}{100}$ $\mathrm{OneDimMinimizer}$ outputs $\nu'$ that satisfies

$$h(\nu') - \min_\nu h(v) \leq 4\epsilon/5$$

and therefore

$$\mathcal{G}(\nu') - \min_\nu h(v) = \mathcal{G}(\nu') - h(\nu') + h(\nu') - \min_\nu h(v) \leq \epsilon.$$

$\square$

Finally, in the following corollary we show that finding an $\epsilon$-suboptimal solution for (3) requires a polylogarithmic number of $O(\epsilon)$-accurate minimizers of (10) and a polylogarithmic number of evaluations of (10).

**Corollary 5.** *Let each $\ell_i$ satisfy Assumption 4 and let $f$ satisfy Assumption 5, then minimizing (3) to accuracy $\epsilon$ with probability at least $\frac{99}{100}$ requires $O(\log(H) \log(\log H))$ evaluations of (10) and $O(\log(H) \log(\log H))$ calls to an algorithm that with probability at least $\frac{1}{2}$ returns an $O(\epsilon)$-suboptimal point of (10), where $H = \frac{B_f B_\ell}{\epsilon}$.*

*Proof.* Note that $\mathcal{L}_{\nu \cdot f}$ is defined on (10). Let $\widetilde{\mathcal{G}}(\nu) := \mathcal{L}_{\nu \cdot f}(\widetilde{x}) + \nu$ where $\widetilde{x}$ is the output of an algorithm that with probability at least $\frac{1}{2}$ returns an $\frac{\epsilon}{5}$-suboptimal point of $\mathcal{L}_{\nu \cdot f}$ and let $h(\nu) := \min_{x \in \mathcal{X}} \mathcal{L}_{\nu \cdot f}(x) + \nu$. We have that $\widetilde{\mathcal{G}}(\nu) - h(\nu) \leq \frac{\epsilon}{5}$ with probability at least $\frac{1}{2}$, therefore, we can apply Proposition 3 with $\mathcal{G} = \widetilde{\mathcal{G}}$, and obtain that with $O\left( \log \left( \frac{B_f \cdot B_\ell}{\epsilon} \right) \log \left( \log \frac{B_f \cdot B_\ell}{\epsilon} \right) \right)$ calls

to $\widetilde{\mathcal{G}}$ (i.e., to an algorithm that outputs $\frac{\epsilon}{5}$-suboptimal minimizer of $\mathcal{L}_{\nu \cdot f}$ with probability at least $\frac{1}{2}$), OneDimMinimizer outputs $\nu'$ that satisfies with probability at least $\frac{99}{100}$

$$\widetilde{\mathcal{G}}(\nu') - \min_\nu h(\nu) = \widetilde{\mathcal{G}}(\nu') - \min_x \mathcal{L}_{f\text{-div}}(x) \le \epsilon.$$

Noting that $\mathcal{L}_{f\text{-div}}(x) = \min_{\nu \ge 0} \{\mathcal{L}_{\nu \cdot f}(x) + \nu\}$ we obtain

$$\mathcal{L}_{f\text{-div}}(\widetilde{x}) - \min_x \mathcal{L}_{f\text{-div}}(x) \le \mathcal{L}_{\nu' \cdot f}(\widetilde{x}) + \nu' - \min_x \mathcal{L}_{f\text{-div}}(x) = \widetilde{\mathcal{G}}(\nu') - \min_x \mathcal{L}_{f\text{-div}}(x) \le \epsilon.$$

$\square$

Corollary 5 means that the complexity bounds for approximately minimizing the objective $\mathcal{L}_\psi$ established by Theorems 3 and 4 also apply (with slightly larger logarithmic factors) to approximately minimizing the constrained $f$-divergence objective $\mathcal{L}_{f\text{-div}}$.

### D.3 Properties of $\mathcal{L}_\psi$ and $\mathcal{L}_{\psi,\epsilon}$

**Lemma 18.** *For $\mathcal{L}_\psi$ defined in* (10) *and $\mathcal{L}_{\psi,\epsilon}$ defined in* (12) *we have that*

$$|\mathcal{L}_{\psi,\epsilon}(x) - \mathcal{L}_\psi(x)| \le \frac{\epsilon}{2} \text{ for all } x \in \mathbb{R}^d.$$

*Proof.* Recall that $\epsilon' = \frac{\epsilon}{2 \log N}$ and for $q \in \Delta^N$ we have that $\sum_{i \in N} q_i \log q_i \in [-\log N, 0]$, therefore:

$$|\mathcal{L}_{\psi,\epsilon}(x) - \mathcal{L}_\psi(x)| = \left| \max_{q \in \Delta^N} \left\{ \sum_{i \in [N]} (q_i \ell_i(x) - \psi(q_i) - \epsilon' q_i \log q_i) \right\} - \max_{q \in \Delta^N} \left\{ \sum_{i \in [N]} (q_i \ell_i(x) - \psi(q_i)) \right\} \right|$$

$$\le \left| \epsilon' \sum_{i \in [N]} q_i \log q_i \right| \le \epsilon' \log N = \epsilon/2.$$

$\square$

### D.4 Gradient estimator stability proofs

**Lemma 4.** *For any convex $\psi : \mathbb{R}_+ \to \mathbb{R}$ and $\psi_\epsilon$ defined in* (10)*, $\log(\psi_\epsilon^{*'}(\cdot))$ is $\frac{1}{\epsilon'}$-Lipschitz.*

Note that from the definition of $\psi_\epsilon^{*'}$ we have that $\psi_\epsilon'(0) \to -\infty$, in addition since $q \in \mathbb{R}_+^N$ then $\psi_\epsilon^{*'}$ is non-negative and $\log(\psi_\epsilon^{*'})$ is well-defined. We now give the proof of Lemma 4.

*Proof.* While we write the proof as though the function $\psi$ is differentiable with derivative $\psi'$, one may readily interpret $\psi'$ as an element in the subdifferential of $\psi$ and the proof continues to hold.

Let $\phi_\epsilon = \epsilon' q \log q$ and recall that $\psi_\epsilon(q) = \psi(q) + \phi_\epsilon(q)$. Fix any two numbers $v_1, v_2 \in \mathbb{R}$ and assume without loss of generality that $v_2 > v_1$. For $i = 1, 2$, let

$$q_i := \psi_\epsilon^{*'}(v_i) \text{ and } p_i := \phi_\epsilon^{*'}(v_i) = e^{v_i/\epsilon'-1}. \tag{36}$$

Note that by definition of the Fenchel dual (and strict convexity of $\psi_\epsilon$), $q_i$ is the unique solution to

$$v_i = \psi_\epsilon'(q_i) = \psi'(q_i) + \phi_\epsilon'(q_i)$$

and moreover that $q_2 \ge q_1$ since $\psi_\epsilon^*$ is convex and therefore $\psi_\epsilon^{*'}$ is non-deceasing. Similarly, $p_i$ is the unique solution to

$$v_i = \phi_\epsilon'(p_i)$$

and $p_2 > p_1$. Combining the two equalities yields

$$v_2 - v_1 = \psi'(q_2) + \phi_\epsilon'(q_2) - \psi'(q_1) - \phi_\epsilon'(q_1) = \phi_\epsilon'(p_2) - \phi_\epsilon'(p_1).$$

Rearranging, we find that

$$0 \le \phi_\epsilon'(q_2) - \phi_\epsilon'(q_1) = \phi_\epsilon'(p_2) - \phi_\epsilon'(p_1) - [\psi'(q_2) - \psi'(q_1)] \le \phi_\epsilon'(p_2) - \phi_\epsilon'(p_1).$$

where $\psi'(q_2) - \psi'(q_1) \geq 0$ holds by convexity of $\psi$ and $q_2 \geq q_1$. Recalling that $\phi'_\epsilon(q) = \epsilon' + \epsilon' \log q$, we have

$$0 \leq \log q_2 - \log q_1 \leq \log p_2 - \log p_1 = \frac{1}{\epsilon'}(v_2 - v_1),$$

where the last equality follows by substituting the definition of $p_i$. The proof is complete upon recalling that $q_i = \psi_\epsilon^{*\prime}(v_i)$. $\qquad\square$

**Lemma 5.** *For $G > 0$, $\ell(x) = (\ell_1(x), \ldots, \ell_N(x))$ and $y^\star(x) = \mathrm{argmin}_{y \in \mathbb{R}} \Upsilon_\epsilon(x, y)$, we have $|y^\star(x) - y^\star(x')| \leq \frac{1}{G}\|\ell(x) - \ell(x')\|_\infty$ for all $x, x' \in \mathcal{X}$. Moreover, if each $\ell_i$ is $G$-Lipschitz, we have $|y^\star(x) - y^\star(x')| \leq \|x - x'\|$.*

*Proof.* For $x, x' \in \mathcal{X}$ w.l.o.g. assume that $y^\star(x) \leq y^\star(x')$ and observe that for every $u \in \mathcal{X}$

$$\sum_{i \in [N]} \psi_\epsilon^{*\prime}(\ell_i(u) - Gy^\star(u)) = 1. \tag{37}$$

Let $\widetilde{\ell}_i(x) = \ell_i(x) + \delta$ with $\delta := \|\ell(x') - \ell(x)\|_\infty$ and $\widetilde{y}(x) := \mathrm{argmin}_{y \in \mathbb{R}}\left\{\sum_{i \in [N]} \psi_\epsilon^*\left(\widetilde{\ell}_i(x) - Gy\right) + Gy\right\}$. Then, according to (37)

$$\sum_{i \in [N]} \psi_\epsilon^{*\prime}\left(\widetilde{\ell}_i(x) - G\widetilde{y}(x)\right) = \sum_{i \in [N]} \psi_\epsilon^{*\prime}(\ell_i(x) + \delta - G\widetilde{y}(x)) \stackrel{(i)}{=} \sum_{i \in [N]} \psi_\epsilon^{*\prime}(\ell_i(x) - Gy^\star(x)) = 1$$

and due to the monotonicity of $\psi_\epsilon^{*\prime}$ and $(i)$ we get

$$G\widetilde{y}(x) = Gy^\star(x) + \delta.$$

By convexity, $\psi_\epsilon^{*\prime}$ is monotonically non decreasing, thus

$$\sum_{i \in [N]} \psi_\epsilon^{*\prime}(\ell_i(x') - G\widetilde{y}(x)) \stackrel{(i)}{\leq} \sum_{i \in [N]} \psi_\epsilon^{*\prime}\left(\widetilde{\ell}_i(x) - G\widetilde{y}(x)\right) = \sum_{i \in [N]} \psi_\epsilon^{*\prime}(\ell_i(x') - Gy^\star(x')) = 1 \tag{38}$$

where $(i)$ follows from noting that $\ell_i(x') \leq \ell_i(x) + \max_{i \in [N]}|\ell_i(x') - \ell_i(x)| = \widetilde{\ell}_i(x)$. Therefore, $Gy^\star(x') \leq G\widetilde{y}(x) = Gy^\star(x) + \delta$ giving

$$G|y^\star(x') - y^\star(x)| \leq \|\ell(x') - \ell(x)\|_\infty.$$

In addition, if each $\ell_i$ is $G$-Lipschitz we have

$$G|y^\star(x') - y^\star(x)| \leq \|\ell(x') - \ell(x)\|_\infty \leq G\|x' - x\|.$$

$\qquad\square$

### D.5 Epoch-SGD BROO implementation

In this section we provide the analysis of our algorithm in the non-smooth case, which consists of combining our general BROO acceleration scheme (Algorithm 1) with a variant of Epoch-SGD [28] that we specialize in order to implement a BROO for $\Upsilon_{\epsilon,\lambda}$ (Algorithm 3). We organize this section as follows. First, we prove Lemma 6 showing that our gradient estimators are unbiased with bounded second moment, and therefore can be used in Algorithm 3. Then, in Proposition 4 we give the convergence rate of Algorithm 3. Combining the previous statements with the guarantees of Proposition 1 we prove Theorem 3.

For convenience, we restate the definitions of $\Upsilon_\epsilon$ and our stochastic estimators for $\nabla_x \Upsilon_\epsilon(x, y)$ and $\nabla_y \Upsilon_\epsilon(x, y)$:

$$\Upsilon_\epsilon(x, y) := \sum_{i \in [N]} \psi_\epsilon^*(\ell_i(x) - Gy) + Gy,$$

and

$$\hat{g}^{\mathrm{x}}(x, y) = \frac{\psi_\epsilon^{*\prime}(\ell_i(x) - Gy)}{\bar{p}_i}\nabla\ell_i(x, y) \ , \ \hat{g}^{\mathrm{y}}(x, y) = G\left(1 - \frac{\psi_\epsilon^{*\prime}(\ell_i(x) - Gy)}{\bar{p}_i}\right)$$

where $\bar{p}_i = \psi_\epsilon^{*\prime}(\ell_i(\bar{x}) - G\bar{y})$.

**Lemma 6.** *Let each $\ell_i$ be $G$-Lipschitz, let $\bar{x} \in \mathcal{X}$ and $\bar{y} = \arg\min_{y \in \mathbb{R}} \Upsilon_{\epsilon,\lambda}(\bar{x}, y)$. Let $r_\epsilon = \frac{\epsilon'}{G}$, then for all $x \in \mathbb{B}_{r_\epsilon}(\bar{x})$ and $y \in [\bar{y} - r_\epsilon, \bar{y} + r_\epsilon]$, the gradient estimators $\hat{g}^{\mathrm{x}}$ and $\hat{g}^{\mathrm{y}}$ satisfy the following:*

1. $\mathbb{E}_{i \sim \bar{p}_i}[\hat{g}^{\mathrm{x}}(x, y)] = \nabla_x \Upsilon_\epsilon(x, y)$ *and* $\mathbb{E}_{i \sim \bar{p}_i}[\hat{g}^{\mathrm{y}}(x, y)] = \nabla_y \Upsilon_\epsilon(x, y)$.

2. $\mathbb{E}_{i \sim \bar{p}_i}\|\hat{g}^{\mathrm{x}}(x, y)\|^2 \leq e^4 G^2$ *and* $\mathbb{E}_{i \sim \bar{p}_i}|\hat{g}^{\mathrm{y}}(x, y)|^2 \leq e^4 G^2$.

*Proof.* We first show that the stochastic gradients $\hat{g}^{\mathrm{x}}, \hat{g}^{\mathrm{y}}$ are unbiased

$$\mathbb{E}_{i \sim \bar{p}_i}[\hat{g}^{\mathrm{x}}(x, y)] = \sum_{i \in [N]} \bar{p}_i \cdot \frac{\psi_\epsilon^{*\prime}(\ell_i(x) - Gy)}{\bar{p}_i} \nabla \ell_i(x) = \nabla_x \Upsilon_\epsilon(x, y),$$

and

$$\mathbb{E}_{i \sim \bar{p}_i}[\hat{g}^{\mathrm{y}}(x, y)] = \sum_{i \in [N]} \bar{p}_i \cdot \left( G \left( 1 - \frac{\psi_\epsilon^{*\prime}(\ell_i(x) - Gy)}{\bar{p}_i} \right) \right) = G - G \sum_{i \in [N]} \psi_\epsilon^{*\prime}(\ell_i(x) - Gy) = \nabla_y \Upsilon_\epsilon(x, y).$$

Next, we bound the second moment of the stochastic gradients. For any $i$ we have

$$\|\hat{g}^{\mathrm{x}}(x, y)\| = \frac{\psi_\epsilon^{*\prime}(\ell_i(x) - Gy)}{\psi_\epsilon^{*\prime}(\ell_i(\bar{x}) - G\bar{y})} \|\nabla \ell_i(x)\| \overset{(i)}{\leq} e^{\frac{\ell_i(x) - Gy - (\ell_i(\bar{x}) - G\bar{y})}{\epsilon'}} G \overset{(ii)}{\leq} e^2 G$$

where $(i)$ follows from Lemma 4 and the fact that $\ell_i$ is $G$-Lipschitz and $(ii)$ uses $G$-Lipschitzness again together with $x \in \mathbb{B}_{r_\epsilon}(\bar{x})$ and $y \in [\bar{y} - r_\epsilon, \bar{y} + r_\epsilon]$ to deduce that $\ell_i(x) - Gy - (\ell_i(\bar{x}) - G\bar{y}) \leq 2Gr_\epsilon \leq 2\epsilon'$. Therefore, we have $\mathbb{E}\|\hat{g}^{\mathrm{x}}(x, y)\|^2 \leq e^4 G^2$ as required. The second moment bound on $\hat{g}^{\mathrm{y}}(x, y)$ follows similarly, since

$$|\hat{g}^{\mathrm{y}}(x, y)| \leq G \max \left\{ 1, \frac{\psi_\epsilon^{*\prime}(\ell_i(x) - Gy)}{\psi_\epsilon^{*\prime}(\ell_i(\bar{x}) - G\bar{y})} \right\} \leq e^2 G.$$

$\square$

**Proposition 4.** *Let $\epsilon, \lambda > 0$, $\epsilon' = \frac{\epsilon}{2 \log N}$ and $r_\epsilon = \frac{\epsilon'}{G}$. For any query point $\bar{x}$ let $\bar{y} = \arg\min_{y \in \mathbb{R}} \Upsilon_{\epsilon,\lambda}(\bar{x}, y)$ and let $x_\star, y_\star = \arg\min_{x \in \mathbb{B}_{r_\epsilon}(\bar{x}), y \in [\bar{y} - r_\epsilon, \bar{y} + r_\epsilon]} \Upsilon_{\epsilon,\lambda}(x, y)$. For $\gamma_k = \frac{1}{8\lambda 2^k}$, $T \geq 1$ and threshold $T_{\text{threshold}} = \frac{G^4}{\lambda^2 \epsilon'^2}$ the output $(x, y)$ of Algorithm 3 satisfies*

$$\mathbb{E}\Upsilon_{\epsilon,\lambda}(x, y) - \Upsilon_{\epsilon,\lambda}(x_\star, y_\star) \leq O\left( \frac{G^2}{\lambda T} \right).$$

*Proof.* For convenience let $x_k = x_k^{(0)}$ and $y_k = y_k^{(0)}$, and in addition let $\bar{p}_i = \psi_\epsilon^{\prime *}(\ell_i(\bar{x}) - G\bar{y})$ be the sampling probability from Algorithm 3. We use induction to prove that

$$\mathbb{E}\Upsilon_{\epsilon,\lambda}(x_k, y_k) - \Upsilon_{\epsilon,\lambda}(x_\star, y_\star) \leq \frac{e^4 G^2}{\lambda 2^k}$$

for all $k$. We start with the base case (k=1).

$$\Upsilon_{\epsilon,\lambda}(x_\star, y_\star) - \Upsilon_{\epsilon,\lambda}(x_1, y_1) = \Upsilon_\epsilon(x_\star, y_\star) - \Upsilon_\epsilon(x_1, y_1) + \frac{\lambda}{2}\|x_\star - x_1\|^2$$

$$\overset{(i)}{\geq} \langle \nabla_x \Upsilon_\epsilon(x_1, y_1), x_\star - x_1 \rangle + \langle \nabla_y \Upsilon_\epsilon(x_1, y_1), y_\star - y_1 \rangle + \frac{\lambda}{2}\|x_\star - x_1\|^2$$

$$\overset{(ii)}{\geq} \langle \nabla_x \Upsilon_\epsilon(x_1, y_1), x_\star - x_1 \rangle + \frac{\lambda}{2}\|x_\star - x_1\|^2$$

where $(i)$ follows from convexity of $\Upsilon_\epsilon$ and $(ii)$ since due to optimality conditions we have that $\langle \nabla_y \Upsilon_\epsilon(x_1, y_1), y_\star - y_1 \rangle \geq 0$. Therefore,

$$\Upsilon_{\epsilon,\lambda}(x_1, y_1) - \Upsilon_{\epsilon,\lambda}(x_\star, y_\star) \leq -\langle \nabla_x \Upsilon_\epsilon(x_1, y_1), x_\star - x_1 \rangle - \frac{\lambda}{2}\|x_\star - x_1\|^2$$

$$\leq \max_{x \in \mathbb{B}_{r_\epsilon}(\bar{x})} \left\{ -\langle \nabla_x \Upsilon_\epsilon(x_1, y_1), x - x_1 \rangle - \frac{\lambda}{2}\|x - x_1\|^2 \right\}$$

$$= \frac{\|\nabla_x \Upsilon_\epsilon(x_1, y_1)\|^2}{2\lambda} \leq \frac{e^4 G^2}{2\lambda}$$

**Algorithm 3:** Dual EpochSGD

**Input:** The function $\Upsilon_\epsilon$ defined in (12), ball center $\bar{x}$, ball radius $r_\epsilon$, regularization parameter $\lambda$, smoothing parameter $\epsilon'$ and iteration budget $T$.

**Parameters:** Initial step size $\gamma_1 = 1/(16\lambda)$, epoch length $T_1 = 128$ and threshold

$$T_{\text{threshold}} = \frac{G^4}{\lambda^2 \epsilon'^2}.$$

1 Initialize $x_1^{(0)} = \bar{x}$

2 Initialize $y_1^{(0)} = \bar{y} = \operatorname{argmin}_{y \in \mathbb{R}} \Upsilon_\epsilon(\bar{x}, y)$

3 Precomupte sampling probabilities $\bar{p}_i = \psi_\epsilon'^*(\ell_i(\bar{x}) - G\bar{y})$

4 **for** $k = 1, \ldots, \lceil \log(T/128 + 1) \rceil$ **do**

5     **for** $t = 0, 2, \cdots T_k - 1$ **do**

6         Sample $i \sim \bar{p}_i$

7         Query stochastic gradients $\hat{g}^{\mathrm{x}}\left(x_k^{(t)}, y_k^{(t)}\right)$ and $\hat{g}^{\mathrm{y}}\left(x_k^{(t)}, y_k^{(t)}\right)$ defined in (13)

8         Update $x_k^{(t+1)} = \operatorname{argmin}_{x \in \mathbb{B}_{r_\epsilon}(\bar{x})} \left\{ \gamma_k \left( \langle \hat{g}^{\mathrm{x}}, x \rangle + \frac{\lambda}{2} \|\bar{x} - x\|^2 \right) + \frac{1}{2} \|x_k^{(t)} - x\|^2 \right\}$

9         Update $y_k^{(t+1)} = \operatorname{argmin}_{y \in [\bar{y} - \frac{\epsilon'}{G}, \bar{y} + \frac{\epsilon'}{G}]} \left\{ \gamma_k (\hat{g}^{\mathrm{y}} \cdot y) + \frac{1}{2} \left( y_k^{(t)} - y \right)^2 \right\}$

10     Set $x_{k+1}^{(0)} = \frac{1}{T_k} \sum_{t \in [T_k]} x_k^{(t)}$

11     Set $y_{k+1}^{(0)} = \frac{1}{T_k} \sum_{t \in [T_k]} y_k^{(t)}$

12     Update $T_{k+1} = 2T_k$

13     Update $\gamma_{k+1} = \gamma_k/2$

14     $k \leftarrow k + 1$

15     **if** $T_k \geq T_{\text{threshold}}$ **then**

16         Recompute $y_{k+1}^{(0)} = \operatorname{argmin}_{y \in \mathbb{R}} \Upsilon_\epsilon(x_{k+1}^{(0)}, y)$

17 **return** $x = x_k^{(0)}$

where the last inequality follows from Jensen's inequality and Lemma 6. This gives the base case of our induction. Let $V_x(x') = \frac{1}{2}\|x - x'\|^2$ and $V_y(y') = \frac{1}{2}|y - y'|^2$ and suppose that there is a $k$ such that $\mathbb{E}\Upsilon_{\epsilon,\lambda}(x_k, y_k) - \Upsilon_{\epsilon,\lambda}(x_\star, y_\star) \leq \frac{e^4 G^2}{\lambda 2^k}$. Then, using the mirror descent regret bound [see, e.g., 3, Lemma 3] for $k + 1$ we get

$$
\begin{aligned}
\mathbb{E}\Upsilon_{\epsilon,\lambda}(x_{k+1}, y_{k+1}) - \Upsilon_{\epsilon,\lambda}(x_\star, y_\star) &\leq \frac{\mathbb{E}V_{x_k}(x_\star)}{\gamma_k T_k} + \frac{\mathbb{E}V_{y_k}(y_\star)}{\gamma_k T_k} \\
&\quad + \frac{\gamma_k}{2} \frac{1}{T_k} \sum_{t=1}^{T_k} \mathbb{E}\|\hat{g}^{\mathrm{x}}(x_k, y_k)\|^2 + \frac{\gamma_k}{2} \frac{1}{T_k} \sum_{t=1}^{T_k} \mathbb{E}\|\hat{g}^{\mathrm{y}}(x_{k,k})\|^2 \\
&\overset{(i)}{\leq} \frac{\mathbb{E}V_{x_k}(x_\star)}{\gamma_k T_k} + \frac{\mathbb{E}V_{y_k}(y_\star)}{\gamma_k T_k} + 2e^4 \gamma_k G^2 \\
&\overset{(ii)}{\leq} \frac{\lambda \mathbb{E}V_{x_k}(x_\star)}{8} + \frac{\lambda \mathbb{E}V_{y_k}(y_\star)}{8} + \frac{e^4 G^2}{2\lambda 2^{k+1}}
\end{aligned}
$$

with $(i)$ following from Lemma 6 and $(ii)$ from the choice of $T_k = 64 \cdot 2^k$ and $\gamma_k = \frac{1}{\lambda 2^{k+3}}$. Next, note that the strong convexity of $\Upsilon_\epsilon(x, y)$ in $x$ implies $\lambda \mathbb{E}V_{x_k}(x_\star) \leq \frac{e^4 G^2}{\lambda 2^k}$ since $\lambda \mathbb{E}V_{x_k}(x_\star) \leq \mathbb{E}\lambda\Upsilon_{\epsilon,\lambda}(x_k, y_k) - \mathbb{E}\Upsilon_{\epsilon,\lambda}(x_\star, y_k) \leq \mathbb{E}\lambda\Upsilon_{\epsilon,\lambda}(x_k, y_k) - \Upsilon_{\epsilon,\lambda}(x_\star, y_\star) \leq \frac{e^4 G^2}{\lambda 2^k}$ by the induction hypothesis. In addition from Lemma 5 we have

$$G|y_k - y_\star| \leq G r_\epsilon = \epsilon'$$

by the constraint on $y$. We now bound $\mathbb{E}V_{y_k}(y_\star)$ in each scenario $T_k \leq T_{\text{threshold}} = \frac{G^4}{\lambda^2 \epsilon'^2}$ or $T_k > T_{\text{threshold}}$.

    1. If $T_k \leq T_{\text{threshold}}$ we have that $|y_k - y_\star| \leq \frac{\epsilon'}{G} \leq \frac{G}{\lambda 2^{k/2+2}}$ and thus $V_{y_k}(y_\star) \leq \frac{G^2}{\lambda^2 2^k}$.

2. If $T_k > T_{\text{threshold}}$ Algorithm 3 will recompute the optimal $y_k$ and using Lemma 5 we have $|y_k - y_\star| \le \|x_k - x_\star\|$ and as a result $V_{y_k}(y_\star) \le V_{x_k}(x_\star)$.

Therefore $\mathbb{E}V_{y_k}(y_\star) \le \frac{e^4 G^2}{\lambda^2 2^k}$ and substituting back the bounds on $\mathbb{E}V_{y_k}(y_\star)$ and $\mathbb{E}V_{x_k}(x_\star)$ we obtain

$$\mathbb{E}\Upsilon_{\epsilon,\lambda}(x_{k+1}, y_{k+1}) - \Upsilon_{\epsilon,\lambda}(x_\star, y_\star) \le \frac{e^4 G^2}{\lambda 2^{k+1}}$$

which completes the induction. Let $K$ be the iteration where the algorithm outputs $x = x_K^{(0)}$ and let $y = y_K^{(0)}$. Noting that $T = O(2^K)$, we have

$$\mathbb{E}\Upsilon_{\epsilon,\lambda}(x, y) - \Upsilon_{\epsilon,\lambda}(x_\star, y_\star) \le O\left(\frac{G^2}{\lambda T}\right).$$

$\square$

**Theorem 3.** *Let each $\ell_i$ satisfy Assumption 1. Let $\epsilon, \lambda, \delta > 0$, and $r_\epsilon = \epsilon/(2G \log N)$. For any query point $\bar{x} \in \mathbb{R}^d$, regularization strength $\lambda \le O(G/r_\epsilon)$ and accuracy $\delta < r_\epsilon/2$, Algorithm 3 outputs a valid $r_\epsilon$-BROO response for $\mathcal{L}_{\psi,\epsilon}$ and has complexity $\mathcal{C}_\lambda(\delta) = O\left(\frac{G^2}{\lambda^2\delta^2} + N\log\left(\frac{r_\epsilon}{\delta}\right)\right)$. Consequently, the complexity of finding an $\epsilon$-suboptimal minimizer of $\mathcal{L}_\psi$ (10) with probability at least $\frac{1}{2}$ is*

$$O\left(N\left(\frac{GR}{\epsilon}\right)^{2/3}\log^{11/3} H + \left(\frac{GR}{\epsilon}\right)^2 \log^2 H\right) \quad \text{where} \quad H := N\frac{GR}{\epsilon}.$$

*Proof.* We divide the proof into correctness arguments and complexity bounds.

**BROO implementation: correctness.** We use Algorithm 3 with $T = O\left(\frac{G^2}{\lambda^2\delta^2}\right)$ for the BROO implementation. Applying Proposition 4 the output $(x, y)$ of Algorithm 3 satisfies

$$\mathbb{E}\Upsilon_{\epsilon,\lambda}(x, y) - \Upsilon_{\epsilon,\lambda}(x_\star, y_\star) \le O(\lambda\delta^2).$$

Therefore, there is a constant $c > 0$ for which the output $(x, y)$ of Algorithm 3 with $T = \frac{cG^2}{\lambda^2\delta^2}$ satisfies $\mathbb{E}\Upsilon_{\epsilon,\lambda}(x, y) - \Upsilon_{\epsilon,\lambda}(x_\star, y_\star) \le \frac{\lambda\delta^2}{2}$, and since

$$\mathbb{E}\mathcal{L}_{\psi,\epsilon}(x) - \min_x \mathcal{L}_{\psi,\epsilon}(x) = \mathbb{E}\min_y \Upsilon_\epsilon(x, y) - \Upsilon_\epsilon(x_\star, y_\star) \le \mathbb{E}\Upsilon_{\epsilon,\lambda}(x, y) - \Upsilon_{\epsilon,\lambda}(x_\star, y_\star),$$

Algorithm 3 returns a valid BROO response for $\mathcal{L}_{\psi,\epsilon}$.

**BROO implementation: complexity.** The total number of epochs that Algorithm 3 performs is $K = O\left(\log\left(\frac{G^2}{\delta^2\lambda^2}\right)\right)$. In $O\left(\log\left(\frac{G^2}{\lambda^2\delta^2}\right) - \log\left(\frac{G^2}{\lambda^2 r_\epsilon^2}\right)\right) = O\left(\log\left(\frac{r_\epsilon}{\delta}\right)\right)$ epochs with $T \ge \frac{G^4}{\lambda^2\epsilon'^2} = \frac{G^2}{r_\epsilon^2\lambda^2}$ the algorithm performs $O(N)$ function evaluations to recompute the optimal $y$. In addition, the algorithm performs $O\left(\frac{G^2}{\lambda^2\delta^2}\right)$ stochastic gradient computations (each computation involves a single loss function $\ell_i$ and a single sub-gradient $\nabla\ell_i$ evaluation). Therefore the total complexity of the BROO implementation is

$$O\left(\frac{G^2}{\lambda^2\delta^2} + N\log\left(\frac{r_\epsilon}{\delta}\right)\right). \tag{39}$$

**Minimizing $\mathcal{L}_\psi$: correctness.** For any $q \in \Delta^N$ let $\mathcal{L}_q(x) := \sum_{i\in[N]} q_i\ell_i(x) - \psi_\epsilon(q_i)$ and note that $\mathcal{L}_q$ is $G$-Lipschitz, since for all $x \in \mathcal{X}$ we have $\|\nabla\mathcal{L}_q(x)\| = \left\|\sum_{i\in[N]} q_i\nabla\ell_i(x)\right\| \le G$. Maximum operations preserve the Lipschitz continuity and therefore $\mathcal{L}_{\psi,\epsilon}(x) = \max_{q\in\Delta^N}\mathcal{L}_q(x)$ is also $G$-Lipschitz. Since $\mathcal{L}_{\psi,\epsilon}$ is $G$-Lipschitz, we can use Proposition 1 with $F = \mathcal{L}_{\psi,\epsilon}$ and obtain that with probability at least $\frac{1}{2}$ Algorithm 1 outputs a point $x$ such that $\mathcal{L}_{\psi,\epsilon}(x) - \min_x \mathcal{L}_{\psi,\epsilon}(x) \le \epsilon/2$. In addition, from Lemma 18 we have

$$\mathcal{L}_\psi(x) - \min_x \mathcal{L}_\psi(x) \le \mathcal{L}_{\psi,\epsilon}(x) - \min_x \mathcal{L}_{\psi,\epsilon}(x) + \frac{\epsilon}{2} \le \epsilon.$$

**Minimizing $\mathcal{L}_\psi$: complexity.** We apply Proposition 1 with $F = \Upsilon_{\epsilon,\lambda}$ and $r_\epsilon = \epsilon/(2\log N \cdot G)$, thus the complexity of finding an $\epsilon/2$-suboptimal minimizer of $\mathcal{L}_{\psi,\epsilon}$ (and therefore an $\epsilon$-suboptimal minimizer of $\mathcal{L}_\psi$) is bounded as:

$$O\left(\left(\frac{R}{r_\epsilon}\right)^{2/3}\left[\left(\sum_{j=0}^{m_\epsilon}\frac{1}{2^j}\mathcal{C}_{\lambda_{\mathrm{m}}}\left(\frac{r_\epsilon}{2^{j/2}m_\epsilon^2}\right)\right)m_\epsilon + (\mathcal{C}_{\lambda_{\mathrm{m}}}(r_\epsilon) + N)m_\epsilon^3,\right]\right)$$

where $m_\epsilon = O\left(\log\frac{GR^2}{\epsilon r_\epsilon}\right) = O\left(\log\frac{GR}{\epsilon}\log N\right)$ and $\lambda_{\mathrm{m}} = O\left(\frac{m_\epsilon^2\epsilon}{r_\epsilon^{4/3}R^{2/3}}\right)$. To obtain the total complexity bound, we evaluate the complexity of running $r_\epsilon$-BROO with accuracy $\delta_j = \frac{r_\epsilon}{2^{j/2}m_\epsilon^2}$ and $\delta = \frac{r_\epsilon}{30}$. Using (39) we get

1. $\mathcal{C}_{\lambda_{\mathrm{m}}}\left(\frac{r_\epsilon}{m_\epsilon^2 2^{j/2}}\right) \quad\quad = \quad\quad O\left(\frac{G^2 2^j m_\epsilon^4}{\lambda_{\mathrm{m}}^2 r_\epsilon^2} + N\log\left(m_\epsilon^2 2^{j/2}\right)\right) \quad\quad = $
   $O\left(\frac{\left(\frac{GR}{\epsilon}\right)^{4/3}}{(\log N)^{2/3}}2^j + N\left(m_\epsilon + \log\left(2^{j/2}\right)\right)\right)$

2. $\mathcal{C}_{\lambda_{\mathrm{m}}}\left(\frac{r_\epsilon}{30}\right) = O\left(\frac{G^2}{\lambda_{\mathrm{m}}^2 r_\epsilon^2} + N\right) = O\left(\frac{\left(\frac{GR}{\epsilon}\right)^{4/3}}{m_\epsilon^4(\log N)^{2/3}} + N\right)$.

Thus

$$O\left(m_\epsilon\sum_{j=0}^{m_\epsilon}\frac{1}{2^j}\mathcal{C}_{\lambda_{\mathrm{m}}}\left(\frac{r_\epsilon}{2^{j/2}m_\epsilon^2}\right)\right) \leq O\left(m_\epsilon^2\frac{\left(\frac{GR}{\epsilon}\right)^{4/3}}{(\log N)^{2/3}} + m_\epsilon^2 N\right)$$

and

$$O\left((\mathcal{C}_{\lambda_{\mathrm{m}}}(r_\epsilon) + N)m_\epsilon^3\right) \leq O\left(\frac{\left(\frac{GR}{\epsilon}\right)^{4/3}}{m_\epsilon(\log N)^{2/3}} + Nm_\epsilon^3\right).$$

Substituting the bounds into Proposition 1, the total complexity is bounded as

$$O\left(\left(\frac{R}{r_\epsilon}\right)^{2/3}\left[Nm_\epsilon^3 + \frac{m_\epsilon^2\left(\frac{GR}{\epsilon}\right)^{4/3}}{(\log N)^{2/3}}\right]\right) \leq O\left(N\left(\frac{GR}{\epsilon}\right)^{2/3}\log^{11/3}\left(\frac{GR}{\epsilon}\log N\right) + \left(\frac{GR}{\epsilon}\right)^2\log^2\left(\frac{GR}{\epsilon}\log N\right)\right)$$

$\square$

### D.6  Accelerated variance reduction BROO implementation

In this section we prove the complexity guarantees of our BROO implementation for (potentially only slightly) smooth losses. We first prove Lemma 7, showing that $\Upsilon_{\epsilon,\lambda}$ (the approximation of our objective (10)) is a finite sum of smooth functions, and thus for the BROO implementation we can use a variance reduction method for a finite (weighted) sums. Then, we give Definition 3 of a "valid accelerated variance reduction" (VR) method, and in Lemma 20 we prove the convergence rate of our BROO implementation (Algorithm 4) which is simply a restart scheme utilizing any valid accelerated VR method. We then combine the guarantees of Lemma 20 and Proposition 1 to prove Theorem 4, our final complexity guarantee in the smooth.

To begin, let us restate here the definition of $\Upsilon_{\epsilon,\lambda}$ (that has the form of a weighted finite sum):

$$\Upsilon_{\epsilon,\lambda}(x,y) = \sum_{i\in[N]}\bar{p}_i\upsilon_i(x,y) \text{ where } \upsilon_i(x,y) := \frac{\psi_\epsilon^*(\ell_i(x) - Gy)}{\bar{p}_i} + Gy + \frac{\lambda}{2}\|x - \bar{x}\|^2 \quad (40)$$

and $\bar{p}_i = \psi_\epsilon^{*\prime}(\ell_i(\bar{x}) - G\bar{y})$.

**Lemma 7.** *For any $i \in [N]$, let $\ell_i$ be $G$-Lipschitz and $L$-smooth, let $r_\epsilon = \frac{\epsilon'}{G}$ and $\lambda = O\left(\frac{G}{r_\epsilon}\right)$. The restriction of $\upsilon_i$ to $x \in \mathbb{B}_{r_\epsilon}(\bar{x})$ and $y \in [\bar{y} - r_\epsilon, \bar{y} + r_\epsilon]$ is $O(G)$-Lipschitz and $O\left(L + \frac{G^2}{\epsilon'}\right)$-smooth.*

*Proof.* To show the Lipschitz property we compute the gradient of $v_i(x,y)$ with respect to $x$ and $y$.

$$\nabla_x v_i(x,y) = \frac{\psi_\epsilon^{*\prime}(\ell_i(x) - Gy)}{\bar{p}_i}\nabla\ell_i(x) + \lambda(x - \bar{x})$$

$$\nabla_y v_i(x,y) = G - G\frac{\psi_\epsilon^{*\prime}(\ell_i(x) - Gy)}{\bar{p}_i}$$

Similarly to the proof of Lemma 6, we have that $\frac{\psi_\epsilon^{*\prime}(\ell_i(x) - Gy)}{\bar{p}_i} \leq e^2$ and therefore $\|\nabla_x v_i(x,y)\| \leq e^2 G = O(G)$ and $|\nabla_y v_i(x,y)| \leq e^2 G = O(G)$, giving the first statement. To bound the smoothness of $v_i$, we compute the Hessian of $v_i(x,y)$.

$$\nabla_x^2 v_i(x,y) = \frac{\psi_\epsilon^{*\prime}(\ell_i(x) - Gy)}{\bar{p}_i}\nabla^2\ell_i(x) + \frac{\psi_\epsilon^{*\prime\prime}(\ell_i(x) - Gy)}{\bar{p}_i}\nabla\ell_i(x)\nabla\ell_i(x)^T + \lambda I$$

$$\nabla_y^2 v_i(x,y) = G^2\frac{\psi_\epsilon^{*\prime\prime}(\ell_i(x) - Gy)}{\bar{p}_i}$$

$$\nabla_{xy} v_i(x,y) - G\frac{\psi_\epsilon^{*\prime\prime}(\ell_i(x) - Gy)}{\bar{p}_i}\nabla\ell_i(x).$$

Lemma 4 implies that, for all $v$, $\frac{\psi_\epsilon^{*\prime\prime}(v)}{\psi_\epsilon^{*\prime}(v)} = \left(\log\psi_\epsilon^{*\prime}(v)\right)' \leq \frac{1}{\epsilon'}$ and $\frac{\psi_\epsilon^{*\prime}(\ell_i(x) - Gy)}{\bar{p}_i} \leq e^2$. In addition note that each $\ell_i$ is $L$-smooth and $G$-Lipschitz and $\lambda = O\left(\frac{G}{r_\epsilon}\right) = O\left(\frac{G^2}{\epsilon'}\right)$, therefore

$$\|\nabla_x^2 v_i(x,y)\|_{op} = \left\|\frac{\psi_\epsilon^{*\prime}(\ell_i(x) - Gy)}{\bar{p}_i}\nabla^2\ell_i(x) + \frac{\psi_\epsilon^{*\prime\prime}(\ell_i(x) - Gy)}{\psi_\epsilon^{*\prime}(\ell_i(x) - Gy)}\frac{\psi_\epsilon^{*\prime}(\ell_i(x) - Gy)}{\bar{p}_i}\nabla\ell_i(x)\nabla\ell_i(x)^T + \lambda I\right\|_{op}$$

$$\leq O\left(e^2\left(L + 2\frac{G^2}{\epsilon'}\right)\right)$$

$$\|\nabla_{xy} v_i(x,y)\| = \left\|G\frac{\psi_\epsilon^{*\prime\prime}(\ell_i(x) - Gy)}{\psi_\epsilon^{*\prime}(\ell_i(x) - Gy)}\frac{\psi_\epsilon^{*\prime}(\ell_i(x) - Gy)}{\bar{p}_i}\nabla\ell_i(x)\right\| \leq e^2\frac{G^2}{\epsilon'}$$

$$\nabla_y^2 v_i(x,y) = G^2\frac{\psi_\epsilon^{*\prime\prime}(\ell_i(x) - Gy)}{\psi_\epsilon^{*\prime}(\ell_i(x) - Gy)}\frac{\psi_\epsilon^{*\prime}(\ell_i(x) - Gy)}{\bar{p}_i}\nabla\ell_i(x) \leq e^2\frac{G^2}{\epsilon'}.$$

Applying Lemma 21 with $h = v_i$ we get that

$$\left\|\nabla^2 v_i(x,y)\right\|_{op} \leq e^2\left(L + 2\frac{G^2}{\epsilon'}\right),$$

proving that each $v_i(x,y)$ is $O\left(L + \frac{G^2}{\epsilon'}\right)$-smooth. $\qquad\square$

**Definition 3.** *For a given ball center $\bar{x} \in \mathcal{X}$, let $z_\star \in \mathbb{B}_{r_\epsilon}(\bar{x}) \times \mathbb{R}$ minimize the function $\Upsilon_{\epsilon,\lambda} : \mathbb{B}_{r_\epsilon}(\bar{x}) \times \mathbb{R} \to \mathbb{R}$, and let $\bar{y} = \mathrm{argmin}_{y\in\mathbb{R}}\Upsilon_{\epsilon,\lambda}(\bar{x},y)$. Let VARIANCEREDUCTION be a procedure that takes in $z \in \mathcal{X} \times \mathbb{R}$ and complexity budget $T$, and outputs $z' \in \mathcal{X} \times \mathbb{R}$. We say that VARIANCEREDUCTION is a valid accelerated VR method if it has complexity $T$ and satisfies the following: there a constant $C$ such that for any $\alpha$, input $z$, and*

$$T \geq C\left(N\frac{\Upsilon_{\epsilon,\lambda}(z) - \Upsilon_{\epsilon,\lambda}(z_\star)}{\alpha} + \sqrt{\frac{\widetilde{L}\|z - z_\star\|^2}{\alpha}}\right) \text{ for } \widetilde{L} = L + \frac{G^2}{\epsilon'},$$

*the output $z'$ of VARIANCEREDUCTION$(z;T)$ satisfies*

$$\mathbb{E}\Upsilon_{\epsilon,\lambda}(z') - \Upsilon_{\epsilon,\lambda}(z_\star) \leq \alpha.$$

**Algorithm 4:** Restarting Accelerated Variance Reduction with Optimal Dual Values

---

**Input:** The function $\Upsilon_{\epsilon,\lambda}(x,y) = \sum_{i\in[N]} \bar{p}_i v_i(x,y)$ defined in (40), number of total restarts $K$, and an algorithm VARIANCEREDUCTION that takes in $x, y \in \mathcal{X} \times \mathbb{R}$ and complexity budget $T$, and outputs $x', y' \in \mathcal{X} \times \mathbb{R}$.

**1** $x_0, y_0 = \bar{x}, \bar{y} = \bar{x}, \operatorname{argmin}_{y\in\mathbb{R}} \Upsilon_\epsilon(\bar{x}, y)$

**2 for** $k = 1, \ldots, K$ **do**

**3** $\quad x_k, y'_k = \text{VARIANCEREDUCTION}(x_{k-1}, y_{k-1}; T)$

**4** $\quad y_k = \operatorname{argmin}_{y\in\mathbb{R}} \Upsilon_\epsilon(x_k, y)$

**5 return** $x_K$

---

**Lemma 19.** *Katyusha$^{sf}$ [38] is a valid accelerated VR method for some $C = O(1)$.*

*Proof.* Immediate from [1, Theorem 4.1] and Lemma 7. (We note that the theorem is stated for a finite sum with uniform weights, but the extension of the theorem and the method to non-uniform sampling is standard). $\qquad\square$

The following lemma shows that Algorithm 4, when coupled with any valid accelerated VR method yields a BROO implementation for

$$\mathcal{L}_{\psi,\epsilon} = \max_{q\in\Delta^N} \sum_{i\in[N]} (q_i \ell_i(x) - \psi_\epsilon(q_i)) = \min_{y\in\mathbb{R}} \Upsilon_\epsilon(x,y),$$

i.e., it outputs an approximate minimizer of

$$\mathcal{L}_{\psi,\epsilon,\lambda}(x) := \mathcal{L}_{\psi,\epsilon}(x) + \frac{\lambda}{2}\|x - \bar{x}\|^2 = \min_{y\in\mathbb{R}} \Upsilon_{\epsilon,\lambda}(x,y)$$

in $\mathbb{B}_{r_\epsilon}(\bar{x})$.

**Lemma 20.** *Let Assumptions 1 and 3 hold, and suppose Algorithm 4 uses a valid accelerated VR method with constant $C$ (defined above). Then, for $\widetilde{L} = L + G^2/\epsilon'$ and $T \geq 2C(N + \sqrt{N\widetilde{L}/\lambda})$, for any $K \geq 0$ the output $x$ of Algorithm 4 satisfies*

$$\mathbb{E}\mathcal{L}_{\psi,\epsilon,\lambda}(x) - \min_{x_\star\in\mathbb{B}_{r_\epsilon}(\bar{x})} \mathbb{E}\mathcal{L}_{\psi,\epsilon,\lambda}(x_\star) \leq 2^{-K}\left(\mathcal{L}_{\psi,\epsilon,\lambda}(\bar{x}) - \min_{x_\star\in\mathbb{B}_{r_\epsilon}(\bar{x})} \mathbb{E}\mathcal{L}_{\psi,\epsilon,\lambda}(x_\star)\right).$$

*Moreover, the complexity of Algorithm 4 is $K(N + T) = O\left(K(N + \sqrt{N\widetilde{L}/\lambda})\right)$*

*Proof.* Let $z_\star = (x_\star, y_\star)$ minimize $\Upsilon_{\epsilon,\lambda}$ in $\mathbb{B}_{r_\epsilon}(\bar{x}) \times \mathbb{R}$, so that $x_\star = \operatorname{argmin}_{x\in\mathbb{B}_{r_\epsilon}(\bar{x})} \mathcal{L}_{\psi,\epsilon,\lambda}(x)$ as well. Note that for all of the outer loop iterations $z_k = (x_k, y_k)$ we have, by the optimality of $y_k$ and Lemma 5,

$$\|z_k - z_\star\|^2 = \|x_k - x_\star\|^2 + (y_k - y_\star)^2 \leq 2\|x_k - x_\star\|^2.$$

Moreover, the $\lambda$-strong convexity of $\mathcal{L}_{\psi,\epsilon,\lambda}$ implies that

$$\|x_k - x_\star\|^2 \leq \frac{2(\mathcal{L}_{\psi,\epsilon,\lambda}(x_k) - \mathcal{L}_{\psi,\epsilon,\lambda}(x_\star))}{\lambda}.$$

Furthermore, note that $\Upsilon_{\epsilon,\lambda}(z_k) = \min_{y\in\mathbb{R}} \Upsilon_{\epsilon,\lambda}(x_k, y) = \mathcal{L}_{\psi,\epsilon,\lambda}(x_k)$. Recalling that by Lemma 5 restricting the domain of $y$ to $[\bar{y} - r_\epsilon, \bar{y} + r_\epsilon]$ does not change the optimal $y$, we conclude that for VARIANCEREDUCTION to guarantee $\Upsilon_{\epsilon,\lambda}(x_{k+1}, y'_{k+1}) - \Upsilon_{\epsilon,\lambda}(z_\star) \leq \alpha$ it suffices to choose

$$T \geq C\left(N\frac{\mathcal{L}_{\psi,\epsilon,\lambda}(x_k) - \mathcal{L}_{\psi,\epsilon,\lambda}(x_\star)}{\alpha} + \sqrt{\frac{4\widetilde{L}(\mathcal{L}_{\psi,\epsilon,\lambda}(x_k) - \mathcal{L}_{\psi,\epsilon,\lambda}(x_\star))}{\lambda\alpha}}\right)$$

In particular, we see that $T \geq 2C\left(N + \sqrt{N\widetilde{L}/\lambda}\right)$ suffices for $\alpha = \frac{\mathcal{L}_{\psi,\epsilon,\lambda}(x_k) - \mathcal{L}_{\psi,\epsilon,\lambda}(x_\star)}{2}$, from which we conclude that

$$\mathcal{L}_{\psi,\epsilon,\lambda}(x_{k+1}) - \mathcal{L}_{\psi,\epsilon,\lambda}(x_\star) = \Upsilon_{\epsilon,\lambda}(x_{k+1}, y_{k+1}) - \Upsilon_{\epsilon,\lambda}(z_\star)$$
$$\leq \Upsilon_{\epsilon,\lambda}(x_{k+1}, y'_{k+1}) - \Upsilon_{\epsilon,\lambda}(z_\star) \leq \frac{\mathcal{L}_{\psi,\epsilon,\lambda}(x_k) - \mathcal{L}_{\psi,\epsilon,\lambda}(x_\star)}{2},$$

giving the claimed optimality bound. Finally, the complexity of the method is clearly $K(T + N)$ since we make $K$ calls to VarianceReduction with complexity $T$ and $K$ exact minimizations over $y$ with complexity $N$. $\qquad\square$

To efficiently implement the BROO , we repeatedly apply an accelerated variance reduction scheme that does not require strong convexity, such as Katyusha$^{ns}$ [2], each time with complexity budget $\widetilde{O}(N + \sqrt{NL'/\lambda})$, where $L' = L + G^2/\epsilon'$. We start each repetition by the $x$ variable output by the previous Katyusha$^{ns}$ call, and with $y = \arg\min_{y \in \mathbb{R}} \Upsilon_{\epsilon,\lambda}(y, x)$ for that $x$. Using Lemma 5 in lieu of strong-convexity in $y$, we show that error halves after each restart, and therefore a logarithmic number of restarts suffices. We arrive at the following complexity bound.

**Theorem 4.** *Let each $\ell_i$ satisfy Assumptions 1 and 3, let $\epsilon, \lambda, \delta > 0$, and $r_\epsilon = \frac{\epsilon}{2G \log N}$. For any query point $\bar{x} \in \mathbb{R}^d$, regularization strength $\lambda \leq O(\frac{G}{r_\epsilon})$ and accuracy $\delta$, Algorithm 4 outputs a valid $r_\epsilon$-BROO response for $\mathcal{L}_{\psi,\epsilon}$ and has complexity $\mathcal{C}_\lambda(\delta) = O\left(\left(N + \frac{\sqrt{N}(G + \sqrt{\epsilon'L})}{\sqrt{\lambda\epsilon'}}\right) \log \frac{Gr_\epsilon}{\lambda\delta^2}\right)$. Consequently, the complexity of finding an $\epsilon$-suboptimal minimizer of $\mathcal{L}_\psi$ (10) with probability at least $\frac{1}{2}$ is*

$$O\left(N\left(\frac{GR}{\epsilon}\right)^{2/3} \log^{14/3} H + \sqrt{N}\left(\frac{GR}{\epsilon} + \sqrt{\frac{LR^2}{\epsilon}}\right) \log^{5/2} H\right) \quad where \quad H := N\frac{GR}{\epsilon}.$$

*Proof.* We first show correctness and complexity of the BROO implementation and then argue the same points for the overall method.

**BROO implementation: correctness and complexity.** Combining Lemmas 19 and 20 and setting $K = \log_2 \frac{\Upsilon_\epsilon(x_0, y_0) - \arg\min_{x \in \mathbb{B}_{r_\epsilon}(\bar{x}), y \in \mathbb{R}} \Upsilon_\epsilon(x,y)}{\lambda\delta^2}$, we obain a valid BROO implementation with complexity

$$O\left(\left(N + \frac{\sqrt{N}\left(\sqrt{L\epsilon'} + G\right)}{\sqrt{\epsilon'\lambda}}\right) \log\left(\frac{\Upsilon_\epsilon(x_0, y_0) - \arg\min_{x \in \mathbb{B}_{r_\epsilon}(\bar{x}), y \in \mathbb{R}} \Upsilon_\epsilon(x,y)}{\lambda\delta^2}\right)\right) \quad (41)$$

Furthermore, we note that $\Upsilon_\epsilon$ is $O(G)$-Lipschitz and therefore $\Upsilon_\epsilon(x_0, y_0) - \arg\min_{x \in \mathbb{B}_{r_\epsilon}(x), y \in \mathbb{R}} \Upsilon_\epsilon(x, y) \leq O(Gr_\epsilon)$.

**Minimizing $\mathcal{L}_\psi$: correctness.** Similarly to the proof of Theorem 3, we note that $\mathcal{L}_{\psi,\epsilon}$ is $G$-Lipschitz, and therefore we can apply Proposition 1 with $F = \mathcal{L}_{\psi,\epsilon}$ and obtain that with probability at least $\frac{1}{2}$ Algorithm 1 outputs $x$ such that $\mathcal{L}_{\psi,\epsilon}(x) - \min_x \mathcal{L}_{\psi,\epsilon}(x) \leq \epsilon/2$ and Lemma 18 gives

$$\mathcal{L}_\psi(x) - \min_x \mathcal{L}_\psi(x) \leq \mathcal{L}_{\psi,\epsilon}(x) - \min_x \mathcal{L}_{\psi,\epsilon}(x) + \frac{\epsilon}{2} \leq \epsilon.$$

**Minimizing $\mathcal{L}_\psi$: complexity.** Applying Proposition 1 with $F = \Upsilon_\epsilon$ and $r_\epsilon = \frac{\epsilon}{2G \log N}$, the complexity of finding an $\epsilon$-suboptimal minimizer of $\mathcal{L}_\psi$ is

$$O\left(\left(\frac{R}{r_\epsilon}\right)^{2/3}\left[\left(\sum_{j=0}^{m_\epsilon} \frac{1}{2^j} \mathcal{C}_{\lambda_m}\left(\frac{r_\epsilon}{2^{j/2} m_\epsilon^2}\right)\right) m_\epsilon + (\mathcal{C}_{\lambda_m}(r_\epsilon) + N) m_\epsilon^3\right]\right) \quad (42)$$

where $m_\epsilon = O\left(\log \frac{GR^2}{\epsilon r_\epsilon}\right) = \log\left(\frac{GR}{\epsilon} \log N\right)$ and $\lambda_m = O\left(\frac{m_\epsilon^2 \epsilon}{r^{4/3} R^{2/3}}\right)$. Using similar calculations to the proof of Theorem 2 we obtain that

1. $\mathcal{C}_{\lambda_\mathrm{m}}\left(\frac{r_\epsilon}{2^{j/2}m_\epsilon^2}\right) = O\left(\left(N + N^{1/2}\left(\frac{G\sqrt{\log N}}{\sqrt{\epsilon}} + \sqrt{L}\right)\frac{1}{\sqrt{\lambda_\mathrm{m}}}\right)\log\left(\frac{\epsilon' 2^j m_\epsilon^4}{\lambda_\mathrm{m}r_\epsilon^2}\right)\right)$

2. $\mathcal{C}_{\lambda_\mathrm{m}}\left(\frac{r_\epsilon}{30}\right) = O\left(\left(N + N^{1/2}\left(\frac{G\sqrt{\log N}}{\sqrt{\epsilon}} + \sqrt{L}\right)\frac{1}{\sqrt{\lambda_\mathrm{m}}}\right)\log\left(\frac{\epsilon'}{\lambda_\mathrm{m}r_\epsilon^2}\right)\right)$

with $\frac{\epsilon'}{\lambda_\mathrm{m}r_\epsilon^2} = O\left(\left(\frac{GR}{\epsilon}\right)^{2/3}\frac{1}{m_\epsilon^2}\right)$ and $\frac{1}{\sqrt{\lambda_\mathrm{m}}} = O\left(\frac{R^{1/3}r_\epsilon^{2/3}}{m_\epsilon\sqrt{\epsilon}}\right)$. Therefore,

$$m_\epsilon \sum_{j=0}^{\infty}\frac{1}{2^j}\mathcal{C}_{\lambda_\mathrm{m}}\left(\frac{r_\epsilon}{2^{j/2}m_\epsilon^2}\right) \leq O\left(m_\epsilon^2\left[N + N^{1/2}\left(\frac{GR^{1/3}}{\epsilon} + \sqrt{\frac{LR^{2/3}}{\epsilon}}\right)r_\epsilon^{2/3}\right]\right).$$

and

$$O\left((\mathcal{C}_{\lambda_\mathrm{m}}(r_\epsilon) + N)m_\epsilon^3\right) \leq O\left(m_\epsilon^4 N + m_\epsilon^{3.5}N^{1/2}\left(\frac{GR^{1/3}}{\epsilon} + \sqrt{\frac{LR^{2/3}}{\epsilon}}\right)r_\epsilon^{2/3}\right).$$

Substituting the bounds into Proposition 1 with $m_\epsilon = \log\left(\frac{GR}{\epsilon}\log N\right)$ and $r_\epsilon = \frac{\epsilon}{2\log N}$ the total complexity is

$$O\left(N\left(\frac{GR}{\epsilon}\right)^{2/3}\log^{14/3}\left(\frac{GR}{\epsilon}\log N\right) + N^{1/2}\left(\frac{GR}{\epsilon} + \sqrt{\frac{LR^2}{\epsilon}}\right)\log^{7/2}\left(\frac{GR}{\epsilon}\log N\right)\right).$$

$\square$

## D.7 Helper lemmas

**Lemma 21.** *Let $x \in \mathbb{R}^d$, $y \in \mathbb{R}$, then for any $h : \mathbb{R}^d \times \mathbb{R} \to \mathbb{R}$ we have*

$$\left\|\nabla^2 h(x,y)\right\|_{\mathrm{op}} \leq \max\left\{\nabla_y^2 h(x,y) + \|\nabla_{xy}h(x,y)\|, \left\|\nabla_x^2 h(x,y)\right\|_{\mathrm{op}} + \|\nabla_{xy}h(x,y)\|\right\}$$

*Proof.*

$$\left\|\nabla^2 h(x,y)\right\|_{\mathrm{op}} = \sup_{\|v\|^2+u^2=1} v^T\nabla_x^2 h(x,y)v + 2u(\nabla_{xy}h(x,y))^T v + u^2\nabla_y^2 h(x,y)$$

$$\overset{(i)}{\leq} \|v\|^2\left\|\nabla_x^2 h(x,y)\right\|_{\mathrm{op}} + 2u\|\nabla_{xy}h(x,y)\|\|v\| + u^2\nabla_y^2 h(x,y)$$

$$\overset{(ii)}{\leq} \|v\|^2\left\|\nabla_x^2 h(x,y)\right\|_{\mathrm{op}} + \left(u^2 + \|v\|^2\right)\|\nabla_{xy}h(x,y)\| + u^2\nabla_y^2 h(x,y)$$

$$= u^2\left(\nabla_y^2 h(x,y) + \|\nabla_{xy}h(x,y)\|\right) + (1-u^2)\left(\left\|\nabla_x^2 h(x,y)\right\|_{\mathrm{op}} + \|\nabla_{xy}h(x,y)\|\right)$$

$$\leq \max\left\{\nabla_y^2 h(x,y) + \|\nabla_{xy}h(x,y)\|, \left\|\nabla_x^2 h(x,y)\right\|_{\mathrm{op}} + \|\nabla_{xy}h(x,y)\|\right\}$$

with $(i)$ following due to Hölder's inequality and $(ii)$ follows from the inequality $2ab \leq a^2 + b^2$. $\square$