# OpenReview forum: "Distributionally Robust Optimization via Ball Oracle Acceleration"
_NeurIPS.cc/2022/Conference — NeurIPS 2022 Accept_

### Official Review · Reviewer_ZWwH · 2022-07-05

**Rating:** 3
**Confidence:** 3
**Soundness:** 3 good
**Presentation:** 2 fair
**Contribution:** 1 poor

**Summary:**

The paper deals with distributionally robust optimization, in two specific settings : group DRO and f-divergence DRO. Distributionally robust optimization can minimizes a supremum over distributions in a set of test distributions, which is finite in the case of group DRO and a ball of finite radius for a certain similarity measure in the case of f-divergence DRO. This supremum makes the optimization harder : one cannot readily apply available techniques for minimisation of convex losses.

The contributions of the paper are :

1. To present methods to compute ball oracles in a fast way in both cases for potentially non smooth function
2. To present methods to compute ball oracles in an accelerated way when dealing with a certain degree of smoothness
3. To use these ball oracles to design an optimization with low complexity in terms of gradient evaluations, relying on a MLMC methods.
Point 1 and 2 are crucially involve the introduction of a way of computing a gradient estimator which differs from previous work (and which also relies one a MLMC method).

**Questions:**

- Can you provide any kind of experiment ? case where BROO has been use even in a non DRO setting
- Could you provide more information about working (or non-working) DRO techniques
- NB : the fact that regularisation transforms the dual to a smooth thing is not new at all (see Nesterov books for instance)


**Limitations:**

For me, the clear limitation is the apparent complexity of the algorithms (there are a lot of logarithmic bricks but which add up) and no experiment to support their applicability.

**Strengths And Weaknesses:**

For me, the methods are in general not new even if the gradient estimate seems new.

I believe the related works are adequately cited for the technical part, and it is pretty clear the this work is essentially an application of a set of technique (MLMC + BROO optimization) to the problem of DRO (even if that application calls for the introduction of some “new” machinery”)

I believe the work on DRO could be more cited, in particular the more applicable work, as this work does not contain any sort of simulation.

    * Quality:

I did not read the entire proofs but a large part of them, and they are correct.

For me, while these theoretical results are interesting, as the lower bounds are known, the goal is to reach them with an implementable algorithm (at least in a conference like NeurIPS). The complete lack of experiments is disturbing, especially as the proposed methods seem to have a very high constant hidden.

Moreover, the papers on which this relies are also of a theoretical nature, and very few experiments convince me of the quality of the method. While this drawback is underlined in the paper, I believe it to be a very crucial lack of the MCML methods.



    * Clarity:

In terms of organisation, the paper is clear

In terms of claims, I believe the references are not clear enough : algorithm 1 mentioned in the main paper has to be found manually, and no detail is given about it. Also, the bounds involve many complex terms. I believe that part of this complexity should be moved to the appendix, to better present the results (as this paper is completely concentrated in the appendices in any case), especially for part 2.


Technically, the paper is involved for a novice in the field, and cannot be followed easily without looking into the references. I believe doing sketches of proofs or algorithm in a more self contained way (for example the lambda bisection is completely skipped) would help understanding.



    * Significance:

Without any kind of applications, even on toy problems, it is very hard for me to evaluate the significance of the results. For me, for NeurIPS, the lack of ANY kind of non-theoretical work is disturbing.

---

> ### Author Response · Authors · 2022-08-02
> **Author response (part 2)**
>
> > In terms of claims, I believe the references are not clear enough : algorithm 1 mentioned in the main paper has to be found manually, and no detail is given about it. Also, the bounds involve many complex terms. I believe that part of this complexity should be moved to the appendix, to better present the results (as this paper is completely concentrated in the appendices in any case), especially for part 2.
>
> We do not consider Algorithm 1, Proposition 1, and its proof, to be a main contribution of the paper. Rather, in the Proposition 1 re-states results of prior with minor modifications, in a way that facilitates arriving at our final complexity bounds given bounds on $\mathcal{C}_{\lambda}(\delta)$ that we subsequently prove and that constitute our main results. If the reviewer believes it would help, we can also include a diagram of the various algorithmic components in the paper, highlighting the parts we contribute and the parts that are mainly based on prior work.
>
> > Technically, the paper is involved for a novice in the field, and cannot be followed easily without looking into the references. I believe doing sketches of proofs or algorithm in a more self contained way would help understanding.
>
> Thank you for the writing suggestion - we will take it into account when revising the paper. In particular, we will try to provide additional proof sketches and intuition.
>
> > Without any kind of applications, even on toy problems, it is very hard for me to evaluate the significance of the results. For me, for NeurIPS, the lack of ANY kind of non-theoretical work is disturbing.
>
> We believe that our paper is significant because it shows new and improved complexity bounds on a class of important DRO problems. No experiment is needed to verify this claim.
>
> > Can you provide any kind of experiment ? case where BROO has been use even in a non DRO setting
>
> We can provide an experiment demonstrating that BROO acceleration is effective for CVaR problems given an *exact BROO*. Please see additional details in a dedicated reply.
>
> > NB : the fact that regularisation transforms the dual to a smooth thing is not new at all (see Nesterov books for instance)
>
> We cite Nesterov’s “Smooth minimization of non-smooth functions” in Table 1 next to “AGD on softmax.” The idea of smoothing by regularizing the dual (specifically using entropy) is indeed old, and goes back at least to Bersekas’s “Nondifferentiable optimization via approximation” from 1975. We will include this and additional citations in the revised manuscript.
>
>
> However, to the best of our knowledge the particular smoothing property we provide in Lemma 4 *is* new. If the reviewer believes otherwise, we would appreciate a precise reference.

---

> ### Author Response · Authors · 2022-08-02
> **Author response (part 1)**
>
> We thank the reviewer for the feedback, which we address in detail below. We hope that the reviewer will re-evaluate our paper based on these replies.
>
> > For me, the methods are in general not new even if the gradient estimate seems new.
>
> We believe that our results are significant because they provide new complexity bound on two large classes of well-studied DRO problems. On the technical level, while we rely on many advanced techniques from recent years, our results are by no means easy corollaries of prior work. In particular, the gradient estimators that the reviewer recognizes as new constitutes the main focus and innovation of our paper. Below, we list three ideas in our paper that (to the best of our knowledge) do not appear in prior works:
>
> 1. Our variance-reduced gradient estimator in Eq. (9) and its variance bound do not follow directly from the SVRG template. Instead, we must carefully apply MLMC within the estimator and conduct a tailored variance analysis (Lemma 3).
>
> 2. Lemma 4 is a basic observation about the multiplicative stability inherent in smoothing Fenchel duals of arbitrary functions using entropy. Except for the special case $\psi=0$, we could not find this lemma anywhere in the literature. We believe that the simplicity and generality of this result makes it likely to be useful in future work.
>
> 3. For Section 4.3, we modify the standard Epoch SGD method to handle the fact that our objective function is not strongly convex in all of its variables. To do so, we separately optimize the non-strongly-convex coordinate after each epoch, and leverage its specific structure via Lemma 5 (which is also new). See lines 297-307 for additional discussion.
>
> > I believe the work on DRO could be more cited, in particular the more applicable work, as this work does not contain any sort of simulation.
>
> > Could you provide more information about working (or non-working) DRO techniques
>
> We cite several application-focused DRO papers [40, 54, 29, 12, 47,21, 49], as well as algorithm-focused papers with experiments [26, 46, 35, 14], and a review paper about DRO [33]. We are happy to take suggestions for additional papers to cite.
>
>
> > For me, while these theoretical results are interesting, as the lower bounds are known, the goal is to reach them with an implementable algorithm (at least in a conference like NeurIPS). The complete lack of experiments is disturbing
>
> We strongly disagree that the goal of every paper published in NeurIPS is to propose implementable algorithms. We believe that strong theoretical foundations are crucial for any science-based discipline - in particular, we believe that improving theoretical bounds on the complexity of important classes of optimization problems is a worthy contribution to the conference.
>
> Even if the reviewer disagrees with these statements, and would prefer NeurIPS to be an empirical-only conference, in reality the conference has a strong record of accepting and even celebrating theory-only papers.  Citations [3,9,51] in our paper have appeared in NeurIPS in prior years and do not contain experiments. Moreover, in NeurIPS 2021, 3 out of 12 oral presentation in sessions whose title contained the word “optimization” had no experiments; and in NeurIPS 2020 at least 7 out of 43 papers in the optimization themed orals and spotlights tracks (tracks 21, 30 and 32) did not not include experiments. A standard of requiring every NeurIPS optimization paper to contain empirical evaluation does not exist - it would be unfair to review this paper assuming otherwise.
>
> > Moreover, the papers on which this relies are also of a theoretical nature, and very few experiments convince me of the quality of the method. While this drawback is underlined in the paper, I believe it to be a very crucial lack of the MCML methods.
>
> Even if MLMC-based methods have limited performance in practice, they are still very useful for proving theoretical complexity bounds. These bounds in turn tell us and the rest of the community what to aim for with designing alternative, more practical methods. Some of the theoretical MLMC papers we rely on have previously appeared in NeurIPS, which again goes to show that a substantial portion of the NeurIPS community cares about theoretical complexity bounds even when they do not immediately translate to better performance in experiments.

---

### Official Review · Reviewer_MgTq · 2022-07-11

**Rating:** 6
**Confidence:** 3
**Soundness:** 3 good
**Presentation:** 2 fair
**Contribution:** 3 good

**Summary:**

The paper proposed new algorithms for two formulations of DRO, i.e. group DRO and f-divergence DRO. The new algorithms achieve better regret bound than existing ones. The key idea is to implement ball optimization oracle and plug it into state of the art optimization algorithms to solve DRO efficiently.

**Questions:**

I think this is a solid paper, I do not have much questions. One suggestion is to move less important content into Appendix and add more discussions in the main paper.

**Limitations:**

Limitations are well-discussed.

**Strengths And Weaknesses:**

Strengths are the solid analyses and improved convergence rate. The weaknesses are that these algorithms are purely theoretical and not tested in practice as mentioned by authors. Also the presentation of the paper could be improved as currently there is too much content filled in the main paper.

---

> ### Author Response · Authors · 2022-08-02
> **Author response**
>
> We thank the reviewer for the comments and suggestions. Below, we address them in detail.
>
> > Strengths are the solid analyses and improved convergence rate.
>
> Thank for recognizing the quality of our analysis and the fact that it improves over previously known complexity bounds.
>
> > The weaknesses are that these algorithms are purely theoretical and not tested in practice as mentioned by authors.
>
> We agree that our algorithms in their current form are impractical and that this is a limitation of our paper. Nevertheless, we believe that the ideas we develop can eventually lead to improved algorithms. In a dedicated reply addressed to all reviewers, we present a preliminary proof of concept experiment showing that our ball oracle acceleration framework significantly outperforms simply iterating a ball oracle, *when the ball oracle is implemented to high accuracy*.
>
> > I think this is a solid paper, I do not have much questions. One suggestion is to move less important content into Appendix and add more discussions in the main paper.
>
> Thank you for the suggestion. When revising the paper we will include additional discussion and move less important content to the appendix.

---

### Official Review · Reviewer_zx8s · 2022-07-12

**Rating:** 7
**Confidence:** 3
**Soundness:** 3 good
**Presentation:** 3 good
**Contribution:** 3 good

**Summary:**

This paper proposes accelerated algorithms for distributionally-robust optimization based on group-structured and bounded f-divergence sets uncertainty. The algorithm uses ball oracle technique, a multi-level monte-carlo technique and exploits specific structures associated with these uncertainty sets. These ingredients lead to improved convergence guarantees both in terms of accuracy of the iterate and the number of iterations.


**Questions:**

--
Can this be extended for Wasserstein metric which is prevalent in a lot of ML literature?

**Limitations:**

--
Lack of numerical experiments demonstrating that the algorithm works.

**Strengths And Weaknesses:**

-- Strength

Nice mixture of existing ideas for uncertainty sets are practically applicable. Further, results are improved for smooth, Lipschitz functions.

-- Weakness:
- The results are tailored to very specific uncertainty sets. The other ingredients or their combination doesn't seem to be new.
- Lack of numerical experiments to validate. This is crucial especially for this paper since the claims themselves are about reducing computation relative to state-of-the-art!

---

> ### Author Response · Authors · 2022-08-02
> **Author response**
>
> We thank the reviewer for the feedback and questions. Below, we address each point in detail.
>
> > Nice mixture of existing ideas for uncertainty sets are practically applicable. Further, results are improved for smooth, Lipschitz functions.
>
> Thank you for recognizing  that our complexity bounds improve over existing theoretical guarantees for relevant DRO forumation
>
> > The results are tailored to very specific uncertainty sets.
>
> Group DRO and $f$ divergence DRO are some of the two of the most well-studied DRO uncertainty sets, see references [50, 25, 42, 40, 32, 14, 53, 12, 47] in our paper. Therefore, while our results are indeed tailored to them, we believe they are still significant.
>
> > The other ingredients or their combination doesn't seem to be new.
>
> The main novelty in our paper is obtaining the complexity bounds described in Table 1. While the way we arrive at these bounds builds extensively on prior work, the bounds are by no means a trivial or even easy conclusion of these papers. Below, we list three specific ideas in our paper that (to the best of our knowledge) do not appear in prior works:
>
> 1. Our variance-reduced gradient estimator in Eq. (9) and its variance bound do not follow directly from the SVRG template. Instead, we must carefully apply MLMC within the estimator and conduct a tailored variance analysis (Lemma 3).
>
> 2. Lemma 4 is a basic observation about the multiplicative stability inherent in smoothing Fenchel duals of arbitrary functions using entropy. Except for the special case $\psi=0$, we could not find this lemma anywhere in the literature. We believe that the simplicity and generality of this result makes it likely to be useful in future work.
>
> 3. For Section 4.3, we modify the standard Epoch SGD method to handle the fact that our objective function is not strongly convex in all of its variables. To do so, we separately optimize the non-strongly-convex coordinate after each epoch, and leverage its specific structure via Lemma 5 (which is also new). See lines 297-307 for additional discussion.
>
> > Lack of numerical experiments to validate. This is crucial especially for this paper since the claims themselves are about reducing computation relative to state-of-the-art!
>
>
> Our paper is about theoretical oracle complexity bounds for DRO. We do not make any claims about empirically improving runtime compared to other algorithms and furthermore discuss in the practical limitations of the algorithms we study (see lines 106-111). Since we do not make any claims that require validation by numerical experiments, we do not believe such experiments are crucial.
>
> More broadly, having empirical claims is not a requirement for acceptance to Neurips. Citations [3,9,51] in our paper have appeared in NeurIPS in prior years and do not contain experiments. Moreover, completely theoretical optimization papers are often awarded spotlight and oral presentations: in NeurIPS 2021, 3 out of 12 oral presentation in sessions whose title contained the word “optimization” had no experiments; and in NeurIPS 2020 at least 7 out of 43 papers in the optimization themed orals and spotlights tracks (tracks 21, 30 and 32) did not not include experiments. A standard of requiring every NeurIPS optimization paper to contain empirical evaluation does not exist.
>
> That being said, we do believe that the techniques we describe in the paper can eventually (after significant additional work) lead to runtime improvements in practice. As a preliminary proof of concept, we conducted an experiment showing that our ball oracle acceleration framework significantly outperforms simply iterating a ball oracle, *when the ball oracle is implemented to high accuracy* - we provide additional details in a dedicated reply addressed to all reviewers.
>
> > Can this be extended for Wasserstein metric which is prevalent in a lot of ML literature?
>
> Wasserstein DRO problems often admit unbiased gradient estimation via a simple Lagrange dual form (see [44, Proposition 1]) that may obviate the need for much of the machinery we use. However, in cases where maximizing the loss over the sample is computationally expensive, our approach might be useful for reducing the number of such maximization operations. This is an interesting topic for future work.

---

### Official Review · Reviewer_uy4v · 2022-07-13

**Rating:** 7
**Confidence:** 3
**Soundness:** 4 excellent
**Presentation:** 2 fair
**Contribution:** 3 good

**Summary:**

This paper proposes efficient algorithms for distributionally robust optimization (DRO). The objective in DRO is to minimize the worst case expected loss over a family of distributions over the population. A simple example of this problem is group DRO, where we have $N$ separate loss functions, and $m$ distributions, where each distribution assigns a weight to each of the loss functions; the objective in this case becomes to minimize the maximum over m weighted linear combinations of the loss functions.
A more challenging setting is DRO with $f$-divergence, where the family of distributions is all distributions that are close to some fixed distribution where the distance between distributions is measured by a divergence.

Assuming that each of the loss functions is Lipschitz, and that the domain has bounded diameter, the authors present an algorithm for group DRO / f-divergence DRO that can compute a point that achieves a loss at most $\epsilon$ larger than the minimum possible in roughly $N\epsilon^{-2/3} + \epsilon^{-2}$ gradient evaluations of individual loss functions. Directly applying previously known algorithms results in a bound of $N\epsilon^{-2}$ gradient evaluations.

The approach is similar to [11], and designs a "ball-optimization oracle" for regularized versions of the problems, in order to be able to apply accelerated optimization algorithm from [9]. This paper establishes the ball optimization oracle by using SGD and building unbiased gradient estimators for the the group DRO and a regularized version of $f$-divergence DRO.

[9] Y. Carmon, A. Jambulapati, Q. Jiang, Y. Jin, Y. T. Lee, A. Sidford, and K. Tian. Acceleration with a ball optimization oracle. In Advances in Neural Information Processing Systems (NeurIPS), 2020.

[11] Y. Carmon, A. Jambulapati, Y. Jin, and A. Sidford. Thinking inside the ball: Near-optimal minimization of the maximal loss. In Conference on Learning Theory (COLT), 2021.

**Questions:**

My main suggestion would be to work at making the key ideas in the paper more accessible. The paper gets really terse and difficult to follow a few pages in.
As a concrete suggestion, shorter theorem statements highlight the most significant parts of the results would be helpful.

**Limitations:**

As the authors already identify and suggest, it seems likely that these bounds should extend to the case of arbitrary subsets of the simplex, but the current algorithms depend highly on the structure of the DRO problems. Obtaining such an extension would be quite impressive.

**Strengths And Weaknesses:**

**Strengths**:
- DRO is a well-studied problem from robust optimization. It is somewhat surprising that the authors are able to obtain essentially the same running time complexity as SGD for small $N,$ and the same as in the case of minimizing the maximum of $N$ loss functions
- The bound on gradient evaluations achieved by this paper matches known lower bounds even in the case of minimizing the maximum over $N$ distinct loss functions. Thus, in some parameter regimes, the presented algorithm is essentially optimal.

**Weaknesses**:
- The approach is a somewhat too similar to [11]. The main difference is the construction of gradient estimators. The estimator for group DRO is pretty straightforward.
- Even the main body of the paper is very very notation heavy, and it is difficult to parse at many places to discern the key technical ideas

---

> ### Author Response · Authors · 2022-08-02
> **Author response**
>
> We thank the reviewer for the helpful feedback and for recognizing the novelty and strength of our theoretical results. Below, we address each issue raised in the review.
>
> > It is somewhat surprising that the authors are able to obtain essentially the same running time complexity as SGD for small $N$ and the same as in the case of minimizing the maximum of $N$ loss functions
>
> We were also pleasantly surprised by this finding.
>
> > The approach is a somewhat too similar to [11]
>
> While our approach substantially builds on [3, 9, 11], we would like to emphasize that our results (improved rates for Group DRO  and DRO with $f$-divergence) are completely new - as we explain in lines 91-105, just using the algorithms in [11] does not not suffice to obtain them. We are confident that the (admittedly quite technical) developments in our paper are necessary for obtaining our new bounds. Below, we list three ideas in our paper that (to the best of our knowledge) do not appear in prior works:
>
> 1. Our variance-reduced gradient estimator in Eq. (9) and its variance bound do not follow directly from the SVRG template. Instead, we must carefully apply MLMC within the estimator and conduct a tailored variance analysis (Lemma 3).
>
> 2. Lemma 4 is a basic observation about the multiplicative stability inherent in smoothing Fenchel duals of arbitrary functions using entropy. Except for the special case $\psi=0$, we could not find this lemma anywhere in the literature. We believe that the simplicity and generality of this result makes it likely to be useful in future work.
>
> 3. For Section 4.3, we modify the standard Epoch SGD method to handle the fact that our objective function is not strongly convex in all of its variables. To do so, we separately optimize the non-strongly-convex coordinate after each epoch, and leverage its specific structure via Lemma 5 (which is also new). See lines 297-307 for additional discussion.
>
> > The main difference is the construction of gradient estimators. The estimator for group DRO is pretty straightforward.
>
> The estimator for group DRO requires application of MLMC - we are not aware of an efficient estimator that doesn’t use it. While the way we apply MLMC in Section 3.2 is perhaps straightforward, we would argue that MLMC itself is still a nontrivial piece of machinery, with most prior applications to stochastic optimization being fairly recent. Consequently, the overall estimator we develop does have some novelty.
>
> Moreover, as we point out in item 1 above, our variance reduction scheme in the group DRO (Section 3.3) setting is significantly less straightforward.
>
> > Even the main body of the paper is very very notation heavy, and it is difficult to parse at many places to discern the key technical ideas
>
> We will revise the paper to streamline notation and  to make it clearer. We hope the list of technical ideas provided above also helps.
>
> > My main suggestion would be to work at making the key ideas in the paper more accessible. The paper gets really terse and difficult to follow a few pages in. As a concrete suggestion, shorter theorem statements highlight the most significant parts of the results would be helpful.
>
> We thank the reviewer for the writing suggestion - we will revise the paper to make our main result statements more concise and use the additional space to provide more intuition behind our results.

---

> > ### Comment · Reviewer_uy4v · 2022-08-09
> > **Thank you for your clarifications**
> >
> > Thank you for your clarifications. I understand better the challenges of achieving your results.
> >
> > I support your paper, and will keep my score.

---

### Author Response · Authors · 2022-08-02
**A preliminary experiment**

A number of reviewers have asked about the lack of empirical evaluation and practical potential of our method. We acknowledge that - despite their significant theoretical implications -  the algorithms in our paper are, in their current form, impractical. We nevertheless believe that the ball optimization approach has the potential to eventually yield powerful practical technique. Below, we describe a preliminary experiment in support of this view. The experiment is meant to answer the following specific question by reviewer ZWwH:

> Can you provide any kind of experiment ? case where BROO has been use even in a non DRO setting

We can provide an experiment demonstrating that BROO (ball-regularized optimization oracle) acceleration is effective for a DRO setting given an *exact BROO*. By “exact BROO” we mean an oracle $\mathcal{O}_r: \mathcal{X}\to\mathcal{X}$ such that $\mathcal{O}_r(x)$ is the (almost) exact minimizer of the objective function $F$ at a ball of radius $r$ around $x$. Given such oracle, we compare two algorithms: the first is simply iterating the oracle $x _ {t+1} = \mathcal{O}(x_t)$, and the second is the improved, practical ball Monteiro-Svaiter scheme from the recent paper [A].

The specific objective $F$ we consider is the CVaR at level $\alpha=0.1$ of logistic classification losses, with smoothing parameter $\epsilon’=10^{-3}$. We use synthetically-generated Gaussian mixture data. We implement $\mathcal{O}_r(\bar{x})$ by applying L-BFGS-B to minimize $\Upsilon _ {\epsilon}(x,y) + \frac{\lambda}{2}\Vert x-\bar{x}\Vert ^2$ over $x$ and $y$, where $\Upsilon _ {\epsilon}$ is as defined in Eq. (12) in our paper and $\lambda$ is gradually decreased until chosen so that $\Vert \mathcal{O} _ r(\bar{x}) - \bar{x}\Vert = r$.


The results of the experiment are summarized in a figure found in the file `rebuttal_experiment.pdf` included in a revised supplementary materials file that we uploaded.
The figure shows the clear advantage of the accelerated method and the validity of the theoretically-predicted $r^{-2/3}$ scaling of the convergence rate. In particular, we see that as $r$ becomes smaller, the advantage of acceleration over oracle iterations becomes more pronounced.


However, we would like to emphasize that these experiments *do not* take into account the computational cost of BROO implementation. Instead, they demonstrate the potential gains from accelerating BROO’s. Adapting the gradient estimation techniques we propose in order to obtain a practical and efficient BROO implementation for DRO problems is still a work in progress.

In our opinion, the preliminary experiment reported here is outside the scope of the submitted paper. However, if the reviewers believe it would strongly improve the submitted paper, we are open to adding it.

[A] Y. Carmon, D. Hausler, A. Jambulapati, Y. Jin and A. Sidford, "Optimal and Adaptive Monteiro-Svaiter Acceleration." arXiv preprint arXiv:2205.15371 (2022).

---

### Meta-Review · Area_Chair_Ro6D · 2022-08-22

**Recommendation:** Accept
**Confidence:** Certain

**Metareview:**

The paper provides new complexity guarantees for distributionally robust optimization (DRO) in two main settings: (i) when the ambiguity set is discrete and consists of a finite number of possible distributions (group DRO) and (ii) when the ambiguity set is based on f-divergence. The algorithmic techniques build on the recent results on ball oracle acceleration by Carmon et al., (2021). The key technical contribution is building a novel unbiased gradient oracle estimator and combining it with SGD & ball oracle acceleration to obtain tight gradient oracle complexity guarantees. The same results cannot be recovered by simply applying existing algorithms. The improvement over existing algorithms in terms of the gradient oracle complexity is reducing it from $N/\epsilon^2$ to $N/\epsilon^{2/3} + 1/\epsilon^2$. The obtained improvements seem to be primarily of theoretical interest at this point, as they require an exact minimizer over a ball of a given radius (i.e., an exact ball oracle); however, this contribution is within the scope of the conference. While numerical experiments are not   a requirement for NeurIPS papers, if the authors do decide to add them, they are advised to ensure that any comparison to other algorithms they make is a fair apple-to-apple comparison. The paper would also benefit from being made more accessible to novice readers who may not be familiar with prior closely related results on ball oracle acceleration.

**Award:**

No

---

### Decision · Program_Chairs · 2022-09-14

Accept